# A lysosomal surveillance response to stress extends healthspan

Terytty Yang Li [1,2,5] ✉, Arwen W. Gao [2,3,5], Rendan Yang [1,5], Yu Sun [1,5], Yuxuan Lei[1], Xiaoxu Li [2], Lin Chen[1], Yasmine J. Liu[2], Rachel N. Arey[4], Kimberly Morales[4], Raya B. Liu[1], Wenzheng Wang [1], Ang Zhou[1], Tong-jin Zhao [1], Weisha Li[3], Amélia Lalou[2], Qi Wang [2], Tanes Lima[2], Riekelt H. Houtkooper [3] & Johan Auwerx [2] ✉

Lysosomes are cytoplasmic organelles central for the degradation of macromolecules to maintain cellular homoeostasis and health. However, how lysosomal activity can be boosted to counteract ageing and ageing-related diseases remains elusive. Here we reveal that silencing specific vacuolar H⁺-ATPase subunits (for example, *vha-6*), which are essential for intestinal lumen acidification in *Caenorhabditis elegans*, extends lifespan by ~60%. This longevity phenotype can be explained by an adaptive transcriptional response typified by induction of a set of transcripts involved in lysosomal function and proteolysis, which we termed the lysosomal surveillance response (LySR). LySR activation is characterized by boosted lysosomal activity and enhanced clearance of protein aggregates in worm models of Alzheimer's disease, Huntington's disease and amyotrophic lateral sclerosis, thereby improving fitness. The GATA transcription factor ELT-2 governs the LySR programme and its associated beneficial effects. Activating the LySR pathway may therefore represent an attractive mechanism to reduce proteotoxicity and, as such, potentially extend healthspan.

Lysosomes are crucial cytoplasmic organelles for degradation and recycling of building blocks and control multiple cellular signalling and metabolic pathways[1–4]. A variety of substrates are degraded in the lysosomes, ranging from macromolecules (including proteins, glycans, lipids and nucleic acids) to organelles and pathogens, which reach the lysosomes either through the endocytic, phagocytic or autophagic routes[5–7]. The catabolic function of the lysosome is accomplished by a wide repertoire of proteases, lipases, nucleases, sulfatases and other hydrolytic enzymes that usually require an optimal acidic pH of 4.5–5.0,

regulating many processes such as the turnover of cellular components, downregulation of surface receptors, inactivation of pathogenic organisms, antigen presentation and bone remodelling[2–5,7].

Dysfunction of lysosomes has been historically associated with lysosomal storage disorders, commonly caused by impaired degradation of lysosomal substrates due to mutations in acidic hydrolases as well as non-enzymatic lysosomal proteins[8,9]. Comprising more than 70 individual rare pathologies, the lysosomal storage disorders have a combined incidence of 1 in 5,000 live births and typically manifest

[1]State Key Laboratory of Genetics and Development of Complex Phenotypes, Shanghai Key Laboratory of Metabolic Remodeling and Health, Laboratory of Longevity and Metabolic Adaptations, Institute of Metabolism and Integrative Biology, Fudan University, Shanghai, China. [2]Laboratory of Integrative Systems Physiology, Interfaculty Institute of Bioengineering, École Polytechnique Fédérale de Lausanne, Lausanne, Switzerland. [3]Laboratory Genetic Metabolic Diseases, Amsterdam Gastroenterology, Endocrinology, and Metabolism, Amsterdam UMC, University of Amsterdam, Amsterdam, the Netherlands. [4]Department of Molecular and Cellular Biology and Center for Precision Environmental Health, Baylor College of Medicine, Houston, TX, USA. [5]These authors contributed equally: Terytty Yang Li, Arwen W. Gao, Rendan Yang, Yu Sun. ✉e-mail: teryttyliyang@fudan.edu.cn; admin.auwerx@epfl.ch

progressive neurodegeneration symptoms since infancy or childhood[8]. The accumulation of misfolded and aggregated proteins caused by impaired lysosomal function and proteostasis facilitates the ageing process[9–14], as well as the onset and progression of proteotoxic degenerative diseases including Alzheimer's disease, Parkinson's disease, Huntington's disease and amyotrophic lateral sclerosis (ALS)[1,9,10].

The vacuolar H⁺-ATPase (v-ATPase), which consists of more than 20 subunits, is a highly conserved large complex proton pump essential for the acidification of lysosomes[15,16]. Furthermore, growing evidence suggested a role of v-ATPase in the acidification of other intracellular and extracellular compartments, such as the intestinal lumen in *Caenorhabditis elegans*[17,18]. The expression of many v-ATPase subunit transcripts decreases with age[19]. In addition to its role as a proton pump, v-ATPase has been shown to be crucial for the sensing and integrating of multiple signalling pathways, including the mechanistic target of rapamycin complex 1 (mTORC1)[20], adenosine monophosphate-activated protein kinase (AMPK)-metformin[21] and Janus kinase 2 (JAK2)-signal transducer and activator of transcription-3 (STAT3) signalling[22], allowing the modulation of key cellular processes such as nutrient sensing, energy metabolism and immune response.

Here we demonstrate that RNA interference (RNAi) of v-ATPase subunits (for example, *vha-6*, *vha-8*, *vha-14* and *vha-20*), which are essential for intestinal lumen acidification, extends *C. elegans* lifespan by ~60%, whereas knocking down of some other v-ATPase subunits (for example, *vha-16* and *vha-19*), which are key for lysosomal acidification, shortens worm lifespan. Transcriptomic analysis revealed an upregulation of 760 genes, enriched for 'lysosome/proteolysis', 'metabolic pathways' and 'innate immune response', specifically in the long-lived *vha-6* RNAi worms. We termed this longevity-linked transcriptional response as the 'lysosomal surveillance response (LySR)', which aims to surveil/maintain or even boost lysosomal function. Indeed, short-lived *vha-16/vha-19* RNAi worms demonstrated disrupted lysosomal activity, while boosted lysosomal activity was detected in the long-lived *vha-6* RNAi worms. A motif prediction analysis of the LySR targets identified ELT-2 as the major regulator of the LySR programme and LySR-linked longevity. Dietary restriction (DR) partially hijacks the LySR pathway to promote longevity. Moreover, in worm models of neurodegenerative diseases and of normal ageing, *vha-6* RNAi-mediated LySR activation enhances proteostasis, reduces protein aggregates and improves animal health. Collectively, these findings reveal a previously uncharacterized longevity mechanism to boost lysosomal function, reduce proteotoxicity and protect against neurodegenerative diseases and normal ageing.

## Results

### Specific v-ATPase RNAi extends lifespan and activates LySR

In light of the fact that an adaptive anti-ageing mitochondrial stress response is activated by RNAi of *cco-1*, a gene that encodes a mitochondrial respiratory chain complex IV subunit[23,24], while the endoplasmic reticulum (ER) stress response is induced by RNAi of an ER chaperone gene, *hsp-3* (ref. 25), we asked whether a lysosomal protective transcriptional response could be activated by knocking down specific v-ATPase subunits and explored its potential association with fitness and longevity. By measuring the lifespan of *C. elegans* exposed to RNAi against each of the major v-ATPase subunits (Fig. 1a–g, Extended Data Fig. 1a–m and Supplementary Table 1), we found that *vha-6* RNAi extended lifespan by almost 70%, while less pronounced lifespan extensions were also detected in worms fed with *vha-8*, *vha-14*, *vha-15* or *vha-20* RNAi (Fig. 1a–e). On the contrary, RNAi targeting *vha-1*, *vha-4*, *vha-5*, *vha-16* or *vha-19* shortened lifespan[26] (Fig. 1f,g and Extended Data Fig. 1a,d,e). By examining six different transgenic strains expressing mCherry-tagged VHA-6 or green fluorescent protein (GFP)-tagged VHA-14, VHA-15, VHA-16, VHA-20 and VHA-1, we confirmed that different VHA RNAi, all reliably reduced the expression of the corresponding VHA subunits (Extended Data Fig. 1n–s).

To determine the transcriptional footprints underlying the extended or shortened lifespan conferred by knocking down different v-ATPase subunits, we compared the transcripts of *C. elegans* exposed to *vha-6* (extended lifespan) and *vha-16* or *vha-19* (reduced lifespan) RNAi (Fig. 1h and Supplementary Table 2). Knockdown of each of the three v-ATPase subunits induced the expression of 4,391–4,900 genes, and the majority (3,322 genes) of them were shared and related to pathways such as 'Integral component of membrane', confirming a key role of v-ATPase in cellular membrane dynamics[15] (Fig. 1i and Extended Data Fig. 1t). In particular, 760 genes were exclusively upregulated in the long-lived *vha-6* RNAi model but not in the short-lived *vha-16* or *vha-19* RNAi model. These 760 genes were enriched for 'lysosome/proteolysis', 'metabolic pathways' and 'innate immune response' (Fig. 1i–k). To focus our future study on these 760 genes that probably contribute to the longevity phenotype, we named this unique lysosome- and longevity-linked transcriptional response, the 'lysosomal surveillance response (LySR)', which can be triggered by knocking down specific v-ATPase subunits (for example, *vha-6*) and typified by the strong induction of a large panel of genes related to the lysosome and proteolysis, such as *cpr-5* and *cpr-8*, two worm orthologues of human cathepsin B[27]. Of note, the LySR programme covered a variety of endopeptidase types including the cysteine type (for example, *cpr-5*), serine type (for example, *ctsa-1.2*), aspartic type (for example, *asp-1*), metallo type (for example, *nep-17*) and dipeptidyl type (for example, *pcp-1*), as well as amidohydrolase (for example, *asah-1*) (Fig. 1k). Interestingly, while RNAi of *vha-6*, *vha-16* and *vha-19* all induced the expression of some autophagy-related transcripts, *vha-16* or *vha-19* RNAi worms had even higher levels of autophagy genes as compared with that in *vha-6* RNAi worms (Fig. 1k). Notably, the GFP intensity of *cpr-5p::gfp* worms fed with RNAi against different v-ATPase subunits strongly correlated with the changes in their mean lifespans (Pearson's $r$, $P = 2.91 \times 10^{-7}$) (Fig. 1l–n and Extended Data Fig. 1u), indicating that the transcriptional level of the lysosomal protease, CPR-5, is probably predictive for the longevity of v-ATPase RNAi worms. Finally, *vha-6* RNAi extended worm lifespan even when the RNAi treatment started since the larval stage 4 (L4)/young adult stage (Extended Data Fig. 1v).

### LySR is a novel stress response and longevity mechanism

As expected, the expression of lysosome/proteolysis-related transcripts, including *cpr-5*, *cpr-8*, *ctsa-1* and *asp-10*, robustly increased in the long-lived (for example, *vha-6*, *vha-8*, *vha-14*, *vha-15* and *vha-20* RNAi) worms but not in the short-lived (for example, *vha-16* and *vha-19* RNAi) worms (Fig. 2a). The impact of *vha-6* RNAi on GFP induction in *cpr-5p::gfp* worms and lifespan extension was reliably reproduced when worms were exposed to different amounts of *vha-6* RNAi (Fig. 2b,c and Extended Data Fig. 2a). By contrast, different amounts of *vha-16* RNAi did not affect or shortened lifespan (Extended Data Fig. 2b–e). Moreover, another two RNAi clones (*vha-6*_RNAi_2 and *vha-6*_RNAi_3) targeting different regions of the *vha-6* messenger RNA, as compared with that used in the RNAi screening (*vha-6*_RNAi_1) (Fig. 2d), consistently induced GFP–CPR-5 expression and extended lifespan (Fig. 2e,f).

To understand whether LySR activation upon *vha-6* RNAi is a result of the induction of other classical stress responses, we used multiple different stress reporter strains and stresses. We found that *vha-6* RNAi partially alleviated the activation of mitochondrial, ER and the oxidative stress responses but not of the heat-shock response (Extended Data Fig. 2f–m), despite that the effect of *vha-6* RNAi on these stress responses might not be as impactful as shown due to the developmental stage differences. Importantly, all these stress reporter strains tested were barely activated upon single *vha-6* RNAi (Extended Data Fig. 2f–m), suggesting that the transcriptional response in reaction to *vha-6* RNAi is a novel stress response specifically related to lysosomes. Of note, different developmental stages of *vha-6* RNAi-treated worms all displayed much higher *cpr-5p::gfp* induction compared with worms given control RNAi (Extended Data Fig. 2n), indicating that the LySR

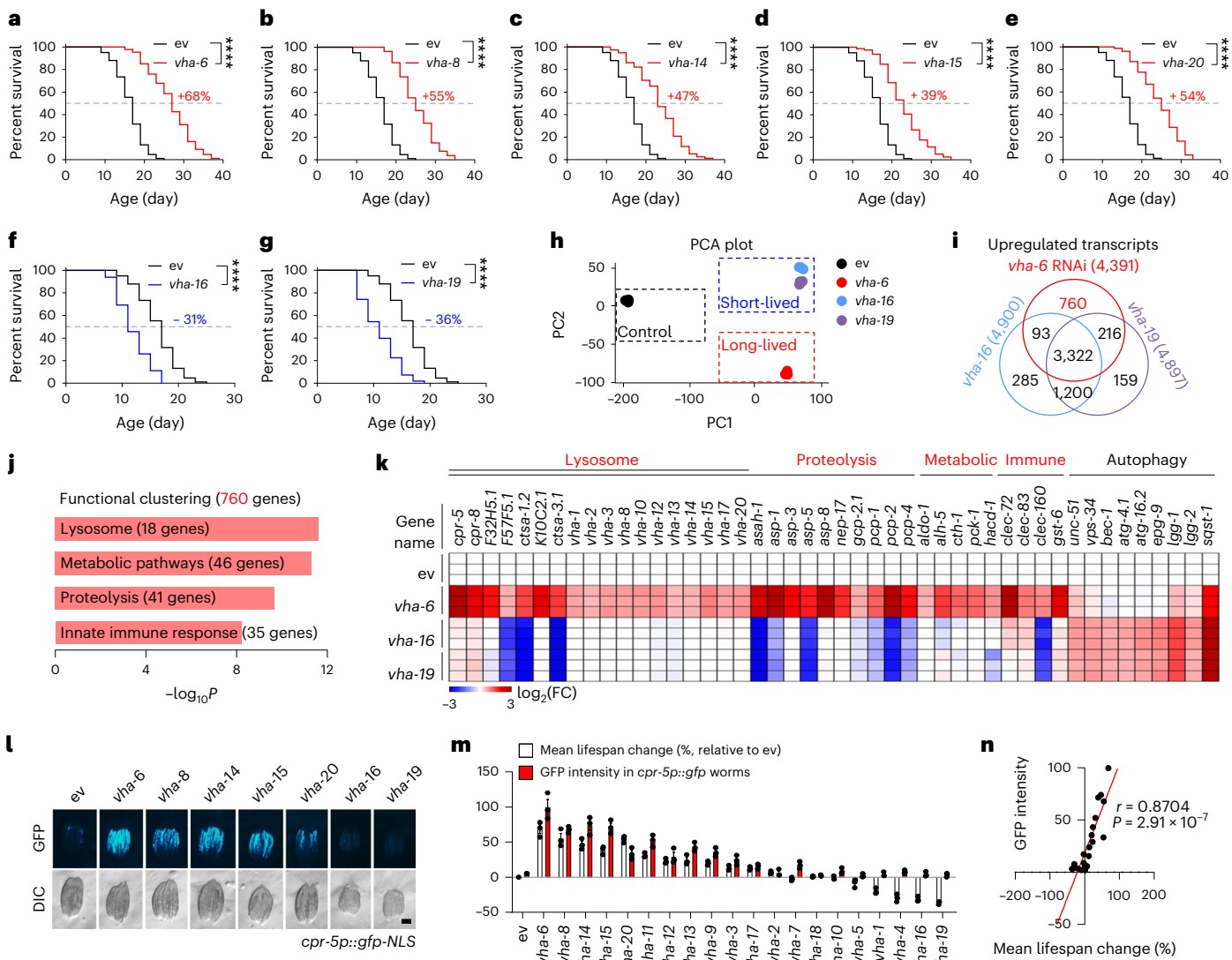

**Fig. 1 | Knockdown of specific v-ATPase subunits extends *C. elegans* lifespan and activates an adaptive lysosomal surveillance response. a–g**, The survival of worms treated with control (ev) or RNAi targeting *vha-6* (**a**), *vha-8* (**b**), *vha-14* (**c**), *vha-15* (**d**), *vha-20* (**e**), *vha-16* (**f**) and *vha-19* (**g**). Each v-ATPase RNAi occupied 40%, except for *vha-6*, *vha-16* and *vha-20* RNAi, which occupied 20% (****P < 0.0001). The control RNAi was used to supply to a final 100% of RNAi for all conditions. The percentages indicate the mean lifespan changes relative to control. **h**, A principal component analysis (PCA) plot of the RNA-seq results of the worms treated with control, *vha-6* (long-lived), *vha-16* and *vha-19* (short-lived) RNAi. PC, principal component. **i**, Venn diagram of the upregulated differentially expressed genes (DEGs) in response to *vha-6*, *vha-16* and *vha-19* RNAi. **j**, The functional clustering of the 760 DEGs as indicated in **i**. The *P* value was derived from DAVID (a one-sided Fisher's exact test). **k**, A heat map of the relative expression levels of representative DEGs in response to *vha-6*, *vha-16* and *vha-19* RNAi. The colour represents the gene expression differences in log$_2$FC relative to the control RNAi condition. FC, fold change. **l**, The GFP expression levels of *cpr-5::gfp* worms treated with RNAi targeting different v-ATPase subunits. DIC, differential interference contrast; NLS, nuclear localization signal. Scale bar, 0.3 mm. **m**, Percentages of the mean lifespan change (relative to the ev condition) and GFP intensity of *cpr-5p::gfp* worms treated with control or RNAi targeting v-ATPase subunits (*n* = 3 independent experiments). **n**, The GFP intensity of *cpr-5p::gfp* worms positively correlates with worm lifespan change. Pearson's correlation coefficient (*r*) was calculated with the mean lifespan change values (*x* axis) and the GFP intensity of *cpr-5p::gfp* worms (*y* axis) as indicated in **m** (two-sided *P* value). The error bars denote the standard error of the mean. The statistical analysis was performed by a log-rank test in **a**–**g**. The statistical data for lifespan can be found in Supplementary Table 1.

response and body size can be decoupled. RNAi of *vha-6* furthermore extends the reproductive span of *C. elegans* (Extended Data Fig. 2o), although the total progeny number was reduced. Of note, the overall retarded reproductive span and reduced total egg output may partially be due to the developmental delay, as also seen in mitochondrial stressed worms[24].

To test if any of the canonical longevity pathways contribute to *vha-6* RNAi-induced lifespan extension, we knocked down *vha-6* in worms carrying null mutations in insulin/IGF-1 signalling[28], mTOR signalling[29], AMPK signalling[30], caloric restriction[31] and mitochondrial stress signalling[32]. *vha-6* RNAi extended the lifespan of *daf-2*,

*daf-16*, *raga-1*, *aak-2*, *eat-2* and *atfs-1* mutants (Fig. 2g–l), suggesting that *vha-6* regulates longevity independently of insulin/IGF-1 (*daf-2/daf-16*), mTOR/AMPK signalling (*raga-1/aak-2*), caloric restriction (*eat-2*) and mitochondrial stress response (*atfs-1*) pathways. Strikingly, the mean lifespan of *vha-6* RNAi-treated *daf-2(e1370)* was extended to 65 days (Figs. 2g), 1.6-fold and 3.8-fold greater than that of control RNAi *daf-2(e1370)* (~40 days) and wild-type (~17 days) worms, respectively.

## Transcription factor ELT-2 governs LySR activation
To identify which transcription factor dominates the LySR activation, we analysed the promoters of the 760 genes upregulated only upon

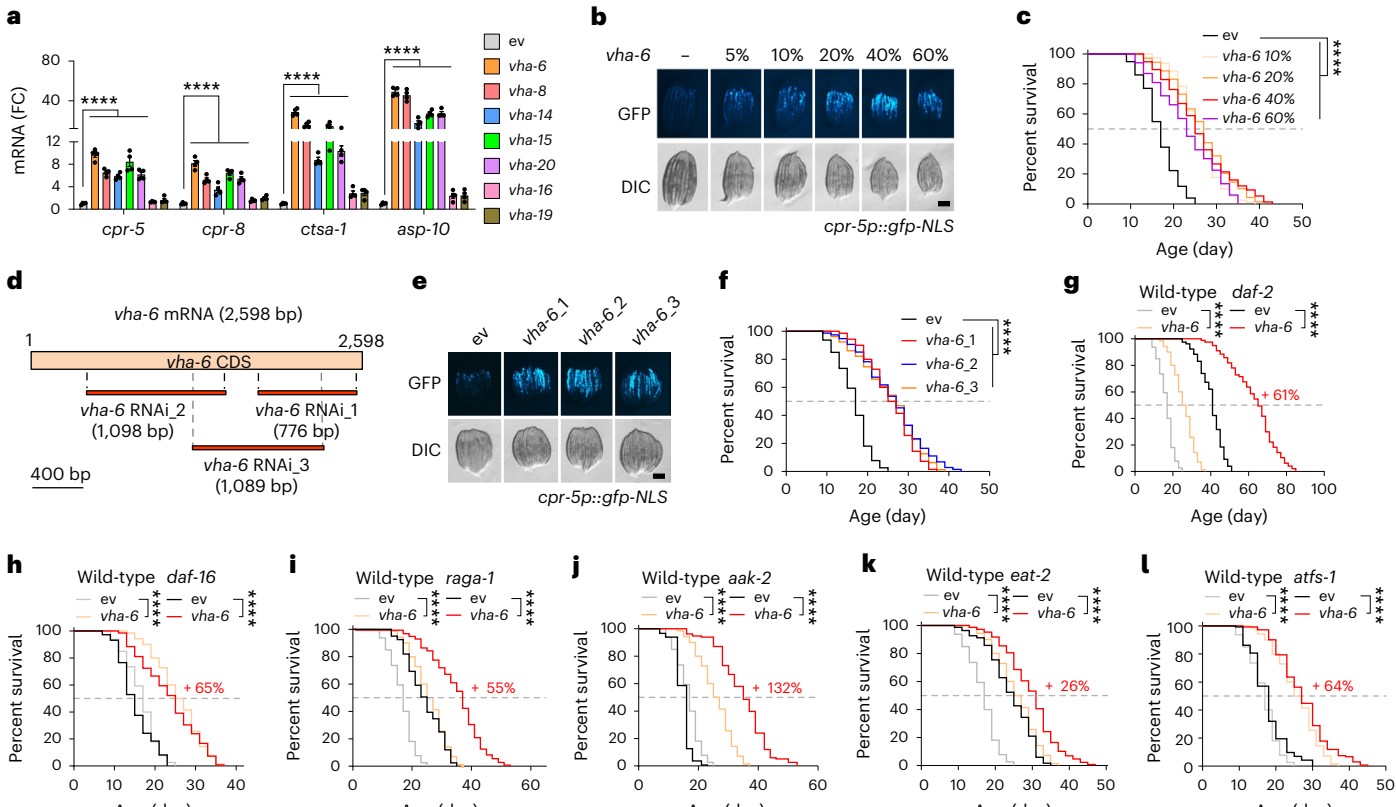

**Fig. 2 | Impact of *vha-6*, *vha-8*, *vha-14*, *vha-15*, *vha-20*, *vha-16* and *vha-19* RNAi on gene expression and lifespan of *C. elegans*. a**, A qRT–PCR analysis of transcripts (*n* = 4 biologically independent samples) in worms treated with control (ev) or RNAi targeting v-ATPase subunits (****$P$ < 0.0001). **b,c**, The GFP–CPR-5 expression level (**b**) and survival (**c**) of worms treated with control or 10–60% *vha-6* RNAi. The control RNAi was used to supply to a final 100% of RNAi for all conditions (****$P$ < 0.0001). **d**, A schematic diagram showing the regions on mRNA targeted by the three *vha-6* RNAi obtained from either the Vidal (*vha-6_1*) or Ahringer (*vha-6_2*, *vha-6_3*) library. CDS, coding sequence.

**e,f**, The GFP–CPR-5 expression level (**e**) and survival (**f**) of worms treated with control or the *vha-6* (20%) RNAi as indicated in **d** (****$P$ < 0.0001). **g–l**, *vha-6* RNAi extends the lifespan of *daf-2(e1370)* (**g**), *daf-16(mu86)* (**h**), *raga-1(ok386)* (**i**), *aak-2(ok524)* (**j**), *eat-2(ad465)* (**k**) and *atfs-1(gk3094)* (**l**) mutants by 61%, 65%, 55%, 132%, 26% and 64%, respectively (****$P$ < 0.0001). Scale bars, 0.3 mm. The error bars denote the standard error of the mean. The statistical analysis was performed by ANOVA followed by Tukey's post hoc test in **a** or a log-rank test in **c** and **f–l**. The statistical data for lifespan can be found in Supplementary Table 1.

*vha-6* RNAi but not upon *vha-16* or *vha-19* RNAi and identified a 10-bp ACTGATAAGA motif (hereafter defined as 'LySR motif') highly enriched in this set of promoters (253 genes out of 760, $P = 1 \times 10^{-31}$) (Fig. 3a, Extended Data Fig. 3a and Supplementary Table 2) and was located ~100 bp upstream of the transcription start site (TSS) for both the 760 '*vha-6* only' genes and all the other *C. elegans* genes, with the '*vha-6* only' genes more enriched according to the similarity scores calculated on the basis of the position weight matrix (Fig. 3b). We next asked which transcription factors may bind to the LySR motif. After comparing this motif with the putative binding motifs of all known transcription factors in *C. elegans*, eight GATA transcription factors were found among the top ten hits (Extended Data Fig. 3b), in line with the presence of a 'GATA' sequence at the centre of the 10-bp LySR motif (Fig. 3a).

By testing RNAi's targeting all 14 known GATA transcription factor genes in *C. elegans*[33,34], we discovered that *elt-2* RNAi but not other GATA family members almost completely blocked the GFP induction in *cpr-5p::gfp* worms in response to *vha-6* silencing (Fig. 3c,d and Extended Data Fig. 3c). In addition, RNAi of *hlh-30*, the worm orthologue of the key lysosomal gene regulator transcription factor EB (TFEB)[35,36], the essential mitochondrial unfolded protein response (UPR^mt) transcription factor *atfs-1* (ref. 32) or *pqm-1*, which encodes a transcription factor that binds to a GATA-like DAF-16 associated element (DAE) motif[37], did not affect *vha-6* RNAi-induced GFP expression in *cpr-5p:gfp* worms (Fig. 3c and Extended Data Fig. 3c). Admittedly, RNAi of these transcription factor candidates may not ensure sufficient

knockdown, which could lead to false negative results. We thus further tested knockout lines of *hlh-30* and *atfs-1* and similar results were acquired (Extended Data Fig. 3d–f). Knockdown of *elt-2* furthermore abrogated the induction of lysosome/proteolysis-related transcripts, including *cpr-5*, *cpr-8*, *ctsa-1* and *asp-10*, upon RNAi of *vha-6*, *vha-8*, *vha-14*, *vha-15* or *vha-20* individually, although with different sensitivity (Fig. 3e,f). Of note, the basal levels of *cpr-5* and *cpr-8* but not *ctsa-1* and *asp-10* were also reduced in *elt-2* RNAi worms (Fig. 3e,f), suggesting that ELT-2 serves as a constitutive regulator for some of the lysosomal protease genes. By contrast, the *vha-6* RNAi-induced expression of these lysosomal proteases was not affected in autophagy-defective mutants (Extended Data Fig. 3g).

We then performed another RNA sequencing (RNA-seq) experiment, this time using the alternative *vha-6* RNAi_2, which targets a different and broader region of the *vha-6* mRNA, as compared with the *vha-6* RNAi_1 previously used in Fig. 1h (Fig. 2d). A total of 3,201 transcripts were commonly induced by *vha-6* RNAi_1 and *vha-6* RNAi_2, among which 488 genes overlapped with the 760 LySR genes (Fig. 3g and Supplementary Table 3). Importantly, within these 488 transcripts, up to 65.2% (318) transcripts required ELT-2 for induction ($P < 2.2 \times 10^{-16}$, two-sided Fisher's exact test), while only 21.9% (1,229) transcripts rely on ELT-2 among all the 5,617 *vha-6* RNAi_2-induced transcripts in general (Fig. 3g). These ELT-2-dependent transcripts were highly enriched for lysosome/proteolysis, innate immune response[38,39] and metabolic pathways (Fig. 3h,i).

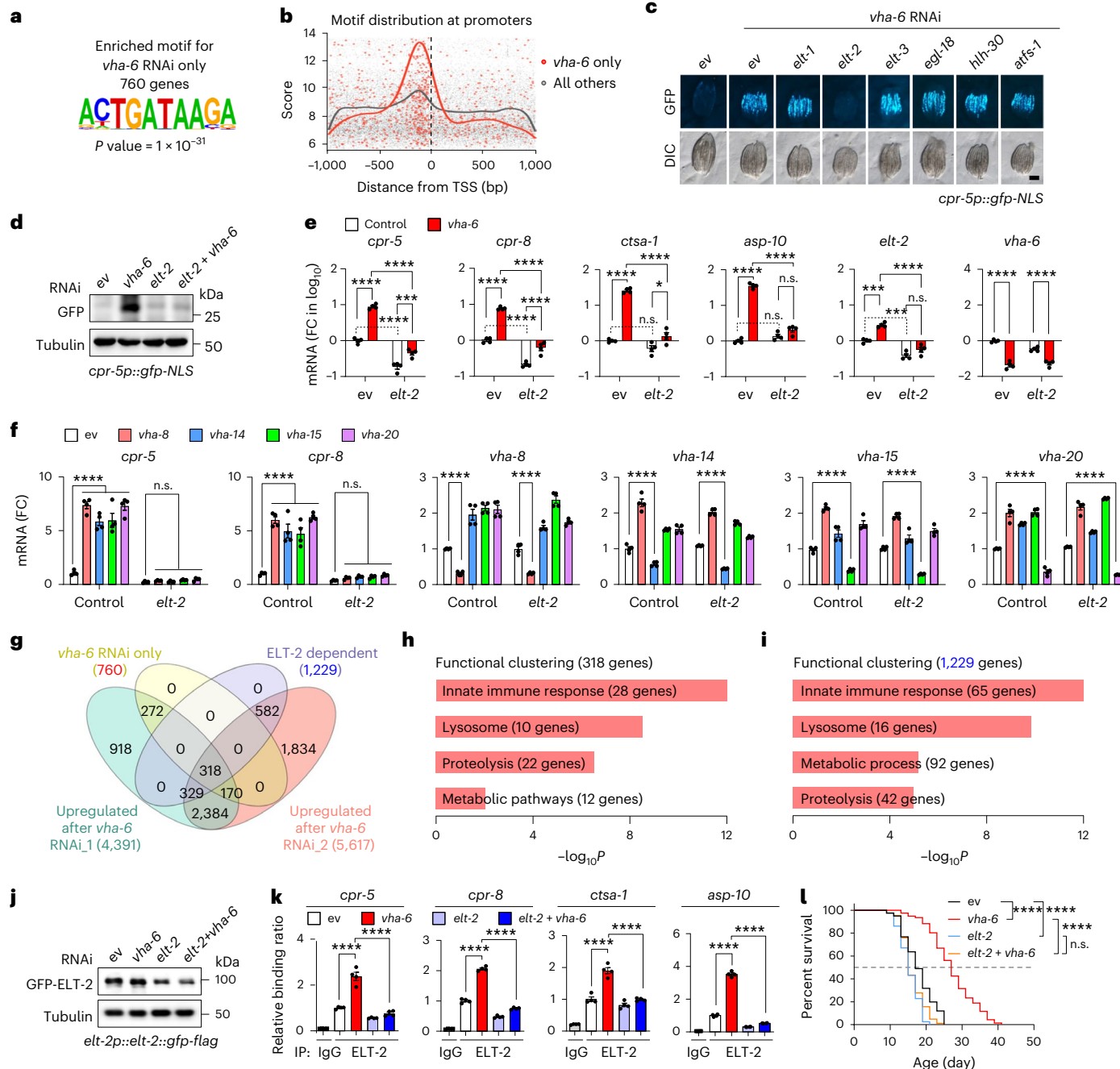

**Fig. 3 | ELT-2 regulates LySR activation and LySR-associated lifespan extension.** **a**, The most enriched binding motif in the promoters of *vha-6* RNAi only 760 genes. The *P* value was derived from HOMER (a one-sided hypergeometric test). **b**, The genomic distribution of the motif hits at promoters of *vha-6* RNAi only and all other genes. **c**, RNAi of *elt-2* attenuated GFP expression of *cpr-5p::gfp* worms upon *vha-6* RNAi. Scale bar, 0.3 mm. **d,e**, The western blots (**d**) and qRT–PCR analysis (**e**) (*n* = 4 biologically independent samples) of *cpr-5p::gfp* worms treated with control, *vha-6* and/or *elt-2* RNAi (****P < 0.0001; for *cpr-5*, ***P = 0.0006 (*elt-2* versus *elt-2* + *vha-6*); for *ctsa-1*, P = 0.1563 (not significant (n.s.), control (ev) versus *elt-2*), *P = 0.0136 (*elt-2* versus *elt-2* + *vha-6*); for *asp-10*, P = 0.2298 (n.s., ev versus *elt-2*), P = 0.0798 (n.s., *elt-2* versus *elt-2* + *vha-6*); for *elt-2*, ***P = 0.0002 (ev versus *vha-6*), ***P = 0.0003 (ev versus *elt-2*), P = 0.1278 (n.s., *elt-2* versus *elt-2* + *vha-6*)). **f**, A qRT–PCR analysis (*n* = 4 biologically independent samples) of worms treated with indicated RNAi (****P < 0.0001; for

*cpr-5*, P > 0.9999 (n.s., *elt-2* versus *elt-2* + *vha-8/vha-14/vha-15*), P = 0.9999 (n.s., *elt-2* versus *elt-2* + *vha-20*); for *cpr-8*, P > 0.9999 (n.s., *elt-2* versus *elt-2* + *vha-8*), P = 0.9971 (n.s., *elt-2* versus *elt-2* + *vha-14*), P = 0.9977 (n.s., *elt-2* versus *elt-2* + *vha-15*), P = 0.9706 (n.s., *elt-2* versus *elt-2* + *vha-20*)). **g**, A Venn diagram of DEGs with indicated conditions. **h,i**, A functional clustering of the 318 (**h**) and 1,229 (**i**) DEGs in **g**. The *P* value was derived from DAVID (a one-sided Fisher's exact test). **j,k**, Western blots (**j**) and ChIP–qPCR (**k**) (*n* = 4 biologically independent samples) of *elt-2p::elt-2::gfp-flag* worms treated with indicated RNAi (****P < 0.0001). **l**, The survival of worms treated with indicated RNAi (****P < 0.0001, P = 0.0541 (n.s., *elt-2* versus *elt-2* + *vha-6*)). The error bars denote the standard error of the mean. The statistical analysis was performed by ANOVA followed by Tukey's post hoc test in **e**, **f** and **k** or a log-rank test in **l**. The statistical data for lifespan can be found in Supplementary Table 1.

Despite an increase in *elt-2* mRNA level in response to *vha-6* RNAi (Fig. 3e), the total protein level of ELT-2 was largely unaffected by *vha-6* RNAi (Fig. 3j). However, by using chromatin immunoprecipitation

coupled with quantitative PCR (ChIP–qPCR) analysis, robust enrichments of ELT-2 were detected at the promoters of lysosomal proteases including *cpr-5*, *cpr-8*, *ctsa-1* and *asp-10* in response to *vha-6* silencing

(Fig. 3k), suggesting that ELT-2 directly binds to the promoters of these LySR genes upon *vha-6* RNAi. As negative controls, ELT-2 does not bind to the promoters of stress-responsive genes including *hsp-4*, *hsp-3*, *sod-3* and *hsp-16.2*, as well as the housekeeping genes *act-1* and *act-3* (Extended Data Fig. 3h). By contrast, ELT-2 strongly binds to the promoters of intestine-enriched genes *ges-1* and *elo-6* (Extended Data Fig. 3h), as reported previously[34,40]. However, these promoter interactions are barely affected by *vha-6* RNAi treatment (Extended Data Fig. 3h). Importantly, RNAi of *elt-2* completely abolished the lifespan extension induced by *vha-6* RNAi (Fig. 3l). Consistent with previous results[34,41], *elt-2* RNAi alone shortened lifespan (Fig. 3l), which may be due to the decreased basal expression of some of the LySR genes, including *cpr-5* and *cpr-8* (Fig. 3e,f). Thus, the GATA transcription factor, ELT-2, is a key regulator of LySR and LySR-associated lifespan extension in *C. elegans*.

## Auxin-inducible degradation-mediated depletion of ELT-2 phenocopies *elt-2* RNAi

As an alternative approach to disrupt *elt-2* function, we tested the auxin-inducible degradation (AID) system for ELT-2 degradation[42]. We generated *elt-2::degron::mNeonGreen* knockin strains using the CRISPR–Cas9 technology[43] (Extended Data Fig. 4a,b) and then combined with the somatic *eft-3* promoter-driven TIR1-mRuby, followed by exposure to the natural auxin indole-3-acetic acid (IAA). IAA applied at 0.1 mM was sufficient to reduce the protein expression of ELT-2 (Extended Data Fig. 4c). Higher concentrations of IAA not only have similar impact on ELT-2 expression but also reduce worm body size (Extended Data Fig. 4c). IAA at 0.1 mM almost completely blocked *vha-6* RNAi-induced lysosomal protease gene expression and lifespan extension (Extended Data Fig. 4d,e), in line with the *elt-2* RNAi results (Fig. 3e,l). Together, these results confirmed a determining role of ELT-2 in LySR activation and LySR-associated lifespan extension.

## Attempts to degrade VHA-6 with the AID/AID2 system

We also tested the AID system for VHA-6 degradation. By using a similar strategy as applied for *elt-2*, we successfully generated two *vha-6::degron::mNeonGreen* knockin strains (Extended Data Fig. 5a–c). However, 1–10 mM of IAA treatment since egg stage barely reduced Degron-mNeonGreen-VHA-6 expression in *vha-6::Degron::mNeonGreen; eft-3p::TIR1::mRuby* worms (Extended Data Fig. 5d). As positive controls, *vha-6* RNAi strongly decreased VHA-6 protein, and 1 mM of IAA effectively degraded Degron-mNeoGreen-DAF-16 (ref. 44) (Extended Data Fig. 5d,e). Likewise, 1-naphthaleneacetic acid (NAA), a synthetic auxin[45], also failed to reduce VHA-6 expression (Extended Data Fig. 5f). Finally, similar negative results were also acquired using the AID2 system for VHA-6 degradation (Extended Data Fig. 5g), by combining AtTIR1(F79G) with the 5-phenyl-indole-3-acetic acid (5-Ph-IAA) ligand[46]. These results suggest that the AID/AID2 system is not an effective method for VHA-6 degradation in *C. elegans*. It is plausible that the apical membrane localization and the large v-ATPase complex in which VHA-6 is assembled may interfere with its proteasome-dependent degradation in AID. Consistently, AID system has been shown to be ineffective in the degradation of DLG-1, a protein localized at the apical junctions of intestinal cells[47].

## Acetyltransferase CBP-1 links VHA-6 loss to LySR activation

Overexpression of *elt-2* with its own promoter has also been shown to extend worm lifespan by ~20% (refs. 34,41). However, such extent of lifespan extension is much less than that of the effect of *vha-6*, *vha-8* or *vha-20* RNAi, which is ~60% extension (Fig. 1a–e). Consistently, by reanalysing an extant RNA-seq dataset (GSE69263)[34], we found that the LySR genes are only sporadically induced in both young and aged *elt-2* overexpression worms, as compared with that in control worms (Extended Data Fig. 6a), suggesting that *elt-2* overexpression alone is insufficient to induce LySR activation. By checking the distribution of GFP-tagged ELT-2 in control and *vha-6* RNAi worms, we observed that ELT-2–GFP constitutively colocalized with the blue-fluorescent DNA

stain, 4′,6-diamidino-2-phenylindole (DAPI) in the intestinal nucleus under both basal and *vha-6* RNAi conditions (Extended Data Fig. 6b,c). Interestingly, the DAPI signal appeared to be more scattered in response to *vha-6* RNAi (Extended Data Fig. 6b). A quantification of the four fractions, divided on the basis of the DAPI staining intensity within each of the nucleus, confirmed an overall higher dispersal and probably less-compacted chromatin/DNA distribution in *vha-6* RNAi-treated intestinal nuclei (Extended Data Fig. 6b). These results suggest that some reorganization of the chromatin/DNA may facilitate the binding of ELT-2 to the promoters of LySR genes upon *vha-6* RNAi.

We thus asked if any type of epigenetic modifications may contribute to LySR activation. As our initial search in this direction, we focused on histone acetylation, a classical epigenetic modification that plays a vital role in chromatin reorganization and transcriptional regulation[48]. By performing an RNAi screen with RNAi targeting all 13 putative lysine acetyltransferases (KATs) in *C. elegans*[49] (Fig. 4a), we found that only RNAi of *cbp-1* (ref. 50), the orthologue of human *CBP/p300*, blocked the GFP induction in *cpr-5p::gfp* worms in response to *vha-6* RNAi to a similar extent as the silencing of *elt-2* (Fig. 4b). Moreover, two distinct *cbp-1* RNAi clones suppressed *vha-6* RNAi-induced expression of typical LySR/ELT-2-target genes, including *cpr-5*, *cpr-8*, *ctsa-1* and *asp-10* (Fig. 4c,d). Similar to the effect of *elt-2* silencing (Fig. 3e,f), *cbp-1* RNAi suppressed the basal expression of *cpr-5* and *cpr-8* but not *ctsa-1* and *asp-10* (Fig. 4d). Notably, CBP-1-dependent histone 3 acetylation at K27 (H3K27Ac) was increased 2.2-fold in worms exposed to *vha-6* RNAi (Fig. 4e). By contrast, neither *cbp-1* nor *vha-6* silencing affects H3K9Ac and H3K4Ac levels (Fig. 4e). It has been known that H3K27Ac, which transforms condensed chromatin into a more relaxed structure, typically facilitates the binding of transcription factors to the promoter regions[48,49]. Consistently, *cbp-1* RNAi attenuated ELT-2 binding to the promoters of LySR genes in response to *vha-6* RNAi (Fig. 4f). Finally, *vha-6* RNAi-induced lifespan extension was strongly blocked by *cbp-1* silencing (Fig. 4g). Together, these results highlight that acetyltransferase CBP-1 is another essential downstream factor that connects VHA-6 loss to ELT-2-mediated LySR activation and lifespan extension.

## Solid dietary restriction partially hijacks the LySR pathway to promote longevity

The acidic environment of the intestinal lumen appeared to be critical for nutrient absorption in worms, as evidenced by reduced dipeptide uptake and fasting-like gene expression pattern in worms exposed to *vha-6* or *vha-20* RNAi[17,18]. Thus, it is likely that the LySR is activated as an adaptive mechanism to improve intracellular energy homoeostasis by boosting lysosome-dependent protein degradation and recycling of building blocks. We found that solid dietary restriction (sDR) of adult worms by feeding them with serially diluted bacteria on solid plates or by complete removal of bacteria[51,52], strongly induced the expression of LySR marker gene *cpr-5* by more than tenfold (Fig. 5a,b). Meanwhile, the expression of *vha-6* was reduced (Fig. 5b). Treatment with *elt-2* RNAi for 24 h (L4 to adult day 1 stage) was sufficient to block the expression of sDR-induced LySR activation and lifespan extension (Fig. 5c,d). In line with this result, RNAi of *elt-2* strongly blocked the lifespan extension of the genetic DR model *eat-2* mutant (Fig. 5e). These results suggest that classical DR models were able to partially hijack the LySR pathway to promote longevity. However, we have also noticed that for several other *vha-6* RNAi-induced lysosomal cathepsin genes such as *ctsa-1* and *asp-10* (Figs. 2a and 3e), their expression is not induced but rather decreased in response to sDR (Fig. 5b). Interestingly, among the 1,020 genes upregulated upon sDR, only 64 (6.3%) genes overlapped with the 760 LySR genes (Fig. 5f and Supplementary Table 4). In addition, *vha-6* RNAi further extends the lifespan of *eat-2* mutant by 26% (Fig. 2k). Moreover, AAK-2 and DAF-16, which are essential for sDR-induced lifespan extension[51], are not required for *vha-6* RNAi-induced longevity (Fig. 2h,j). Thus, *vha-6* RNAi-induced disruption of the intestinal lumen pH and subsequent adaptive response is only partially explained by sDR.

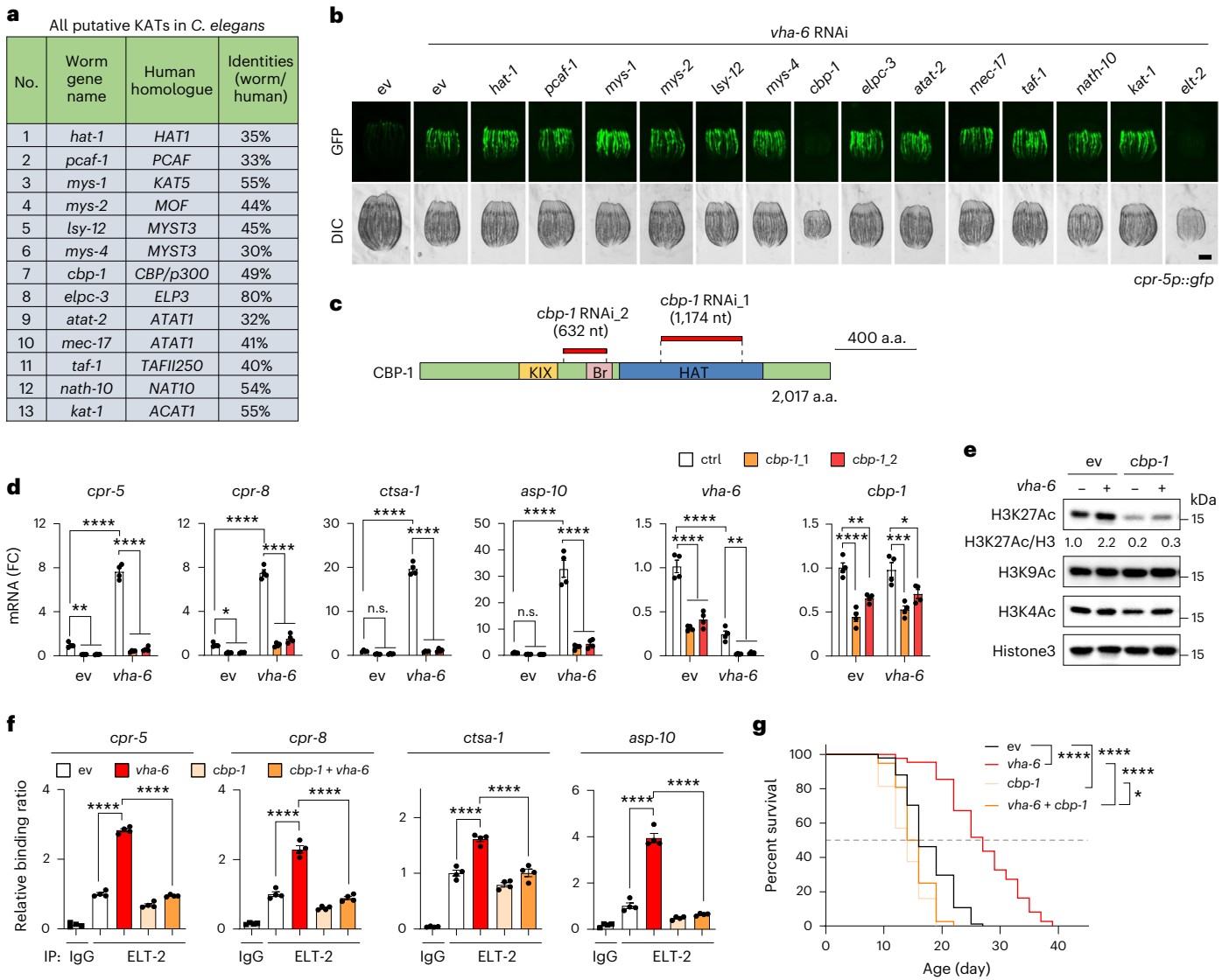

**Fig. 4 | CBP-1 links VHA-6 loss to LySR activation and longevity. a**, All the KATs in *C. elegans* and their human homologues. **b**, The identification of CBP-1 as an essential gene for LySR activation. *cpr-5p::gfp* worms were treated with control (ev) or *vha-6* (25%) RNAi in combination with RNAi targeting different KATs (75%). Scale bar, 0.3 mm. **c**, A shematic diagram showing the different regions targeted by the two different *cbp-1* RNAi clones. KIX, kinase-inducible domain interacting domain; Br, bromodomain; HAT, histone acetyltransferase domain; a.a., amino acids; nt, nucleotides. **d**, A qRT–PCR analysis (*n* = 4 biologically independent samples) of worms treated with control, *cbp-1* and/or *vha-6* RNAi (****$P$ < 0.0001; for *cpr-5*, **$P$ = 0.0091 (ev versus *cbp-1_1*), **$P$ = 0.0092 (ev versus *cbp-1_2*); for *cpr-8*, *$P$ = 0.0315 (ev versus *cbp-1_1*), *$P$ = 0.0368 (ev versus *cbp-1_2*); for *ctsa-1*, $P$ = 0.6009 (not significant (n.s.), ev versus *cbp-1_1*), $P$ = 0.6595 (n.s., ev versus *cbp-1_2*); for *asp-10*, $P$ = 0.9997 (n.s., ev versus *cbp-1_1*), $P$ = 0.9995 (n.s., ev versus

*cbp-1_2*); for *vha-6*, **$P$ = 0.0066 (*vha-6* versus *vha-6* + *cbp-1_1*), **$P$ = 0.0100 (*vha-6* versus *vha-6* + *cbp-1_2*); for *cbp-1*, **$P$ = 0.0025 (ev versus *cbp-1_2*), ***$P$ = 0.0001 (*vha-6* versus *vha-6* + *cbp-1_1*), *$P$ = 0.0206 (*vha-6* versus *vha-6* + *cbp-1_2*)). **e**, H3K27 acetylation increases in a CBP-1-dependent manner during LySR activation induced by *vha-6* RNAi. The worms were treated with control, *cbp-1* RNAi_1 and/or *vha-6* RNAi. **f**, A ChIP–qPCR (*n* = 4 biologically independent samples) of *elt-2p::elt-2::gfp-flag* worms treated with control, *vha-6* and/or *cbp-1* RNAi_1 (****$P$ < 0.0001). IP, immunoprecipitation. **g**, The survival of worms treated with control, *vha-6* (20%), and/or *cbp-1* (50%) RNAi_1 (****$P$ < 0.0001, *$P$ = 0.0102 (*cbp-1* versus *vha-6* + *cbp-1*)). The error bars denote the standard error of the mean. The statistical analysis was performed by ANOVA followed by Tukey's post hoc test in **d** and **f** or a log-rank test in **g**. The statistical data for lifespan can be found in Supplementary Table 1.

## LySR activation boosts lysosomal activity

We noticed that the expression patterns of VHA-6 and VHA-16 proteins are different. Indeed, VHA-6 expression is mostly confined to the apical membrane of the intestine cells[17] (Fig. 6a, Extended Data Fig. 7a and Supplementary Video 1), while VHA-16 is widely expressed in various tissues including pharynx, excretory cells, hypodermis and vulva[16] (Fig. 6b). Some intestinal apical membrane localization was also found for VHA-14, VHA-15 and VHA-20 (ref. 18) (Extended Data Fig. 7b–d). It has been reported that VHA-6 almost exclusively acts at the intestinal apical membrane and contributes to the acidification of intestine lumen, rather than the acidification of intracellular organelles[17].

By taking advantage of a pH sensitive dye, Oregon Green-dextran 488 (refs. 17,53), which yields a brighter green signal at a higher pH with a pKa of 4.8, we found that RNAi of *vha-6*, *vha-8*, *vha-14*, *vha-15* and *vha-20* lead to a brighter signal of Oregon Green, suggesting disrupted intestinal lumen acidification, while much less impact of *vha-16* and *vha-19* RNAi was detected (Extended Data Fig. 7e,f). Such differences nicely separate the long-lived and short-lived v-ATPase RNAi conditions, suggesting a close link between intestinal lumen acidification disruption in LySR activation and worm longevity.

To reveal the impact of different v-ATPase RNAi and LySR activation on lysosomal function, we first examined the lysosomal pH

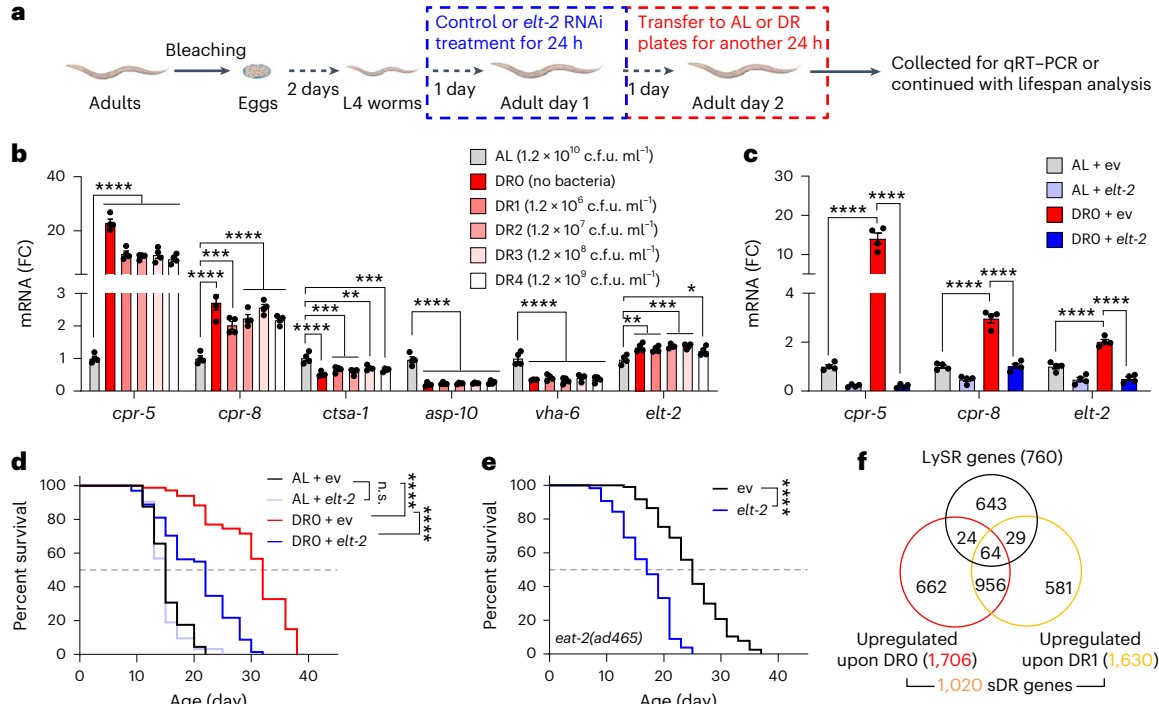

**Fig. 5 | DR partially hijacks the LySR pathway to promote longevity.**
**a**, A schematic diagram for the DR of worms since adult day 1 stage. **b**, DR mimics *vha-6* RNAi and induces LySR activation in *C. elegans*. A qRT–PCR analysis (*n* = 4 biologically independent samples) of adult day 2 worms after feeding with ad libitum (AL) (-1.2 × 10¹⁰ c.f.u. ml⁻¹), serially diluted HT115 bacteria or no bacteria for 1 day since adult day 1 stage (****P < 0.0001; for *cpr-8*, ***P = 0.0005 (AL versus DR1); for *ctsa-1*, ***P = 0.0010 (AL versus DR1), ***P = 0.0001 (AL versus DR2), **P = 0.0039 (AL versus DR3), ***P = 0.0006 (AL versus DR4); for *elt-2*, **P = 0.0017 (AL versus DR0), **P = 0.0069 (AL versus DR1), ***P = 0.0007 (AL versus DR2/DR3), *P = 0.0336 (AL versus DR4)). **c,d**, A qRT–PCR analysis (*n* = 4 biologically independent samples) (**c**) or the survival (**d**) of worms treated with control or *elt-2* RNAi between L4 to adult day 1 and then transferred to plates with AL (-1.2 × 10¹⁰ c.f.u. ml⁻¹) or no bacteria (DR0). Adult day 2 worms were analysed for **c** (****P < 0.0001) (in **d**, P = 0.4262 (not significant (n.s.), AL + control (ev) versus AL + *elt-2*)). **e**, The survival of *eat-2(ad465)* worms treated with control or *elt-2* RNAi (****P < 0.0001). **f**, A Venn diagram of the upregulated DEGs in response to DR0 and DR1 as indicated in **b** and the 760 LySR genes. 1,020 genes that commonly upregulated upon both DR0 and DR1 were considered as genes upregulated upon sDR. The error bars denote the standard error of the mean. The statistical analysis was performed by ANOVA in **b** and **c** or a log-rank test in **d** and **e**. The statistical data for lifespan can be found in Supplementary Table 1.

by costaining the worms with LysoSensor and LysoTracker[19,54]. The LysoTracker is less sensitive to increased acidity than LysoSensor and is used as a control for normalizing the dye intake[19,54]. A strong signal of LysoSensor Green and LysoTracker Red (note the yellow signal in merged images) was found in both the control and the long-lived worms with *vha-6*, *vha-8*, *vha-14*, *vha-15* and *vha-20* RNAi, indicating overall normal lysosomal acidification in these worms (Fig. 6c,d and Extended Data Fig. 7g,h). *vha-6* RNAi worms demonstrated even a more intense yellow/green signal compared with that in control worms (Fig. 6c,d), suggesting boosted lysosomal activity upon *vha-6* RNAi treatment. By contrast, the LysoSensor Green signal was attenuated in the short-lived *vha-16* and *vha-19* RNAi-treated worms, while the LysoTracker staining was largely unaffected (Fig. 6c,d), indicating the attenuation of lysosomal acidification in these short-lived worms. Of note, the remaining LysoTracker signal in these *vha-16* and *vha-19* RNAi worms could be dissipated by the pretreatment of v-ATPase inhibitor bafilomycin A1 (BafA1)[55] (Extended Data Fig. 7i,j). These data suggest that a certain level of residual VHA-16/VHA-19 probably exist in these *vha-16/vha-19* RNAi conditions, whereby only 20% of the bacterial diet consisted of *vha-16/vha-19* RNAi bacteria, with the remaining 80% being control RNAi bacteria (as specified in the figure legends), resulting in lysosomes with minimal function that could still be labelled by the LysoTracker, but not the LysoSensor. In support of this hypothesis, a higher dose (60%) of *vha-16/vha-19* RNAi disrupted the LysoTracker signal even without BafA1 pretreatment (Extended Data Fig. 7i,j), in line with even smaller size and further shortened lifespan as the *vha-16* RNAi dosage increases (Extended Data Fig. 2b,c).

To verify the impact of different v-ATPase RNAi on lysosomal acidity, we utilized a worm strain *hsp-16.2p::nuc-1::pHTomato*[19], which expresses the lysosomal marker protein NUC-1 tagged with a pH sensitive Tomato variant pHTomato under the control of a heat-shock promoter. pHTomato has a pKa ~7.8 and exhibits increased fluorescence when the pH increases[56]. We found that the average fluorescence intensity of NUC-1::pHTomato in each lysosome was significantly lower in worms treated with *vha-6* RNAi than in control (Extended Data Fig. 7k,l). By contrast, worms exposed to *vha-16* or *vha-19* RNAi demonstrated increased fluorescence intensity of NUC-1::pHTomato. As a positive control, RNAi of the lysosomal Ca²⁺ channel *cup-5* disrupts lysosome acidity and increases NUC-1::pHTomato fluorescence[19,57] (Extended Data Fig. 7k,l). Collectively, these data validate that lysosomal acidification is boosted in the long-lived *vha-6* RNAi worms and attenuated in the short-lived *vha-16/vha-19* RNAi worms.

Next, by taking advantage of the autophagy-indicator strain *lgg-1p::lgg-1::gfp*[58], we found that both the GFP–LGG-1 and the processed GFP signal were robustly increased in worms exposed to *vha-16* and *vha-19* RNAi (Fig. 6e,f), confirming a disruption of lysosomal activity-dependent autophagic degradation in these worms[26]. Meanwhile, a trend of reduced GFP–LGG-1 and increased processed GFP was detected in *vha-6* RNAi worms (Fig. 6e,f), indicating intact lysosomal function and some increased autophagic-lysosomal activity in response to *vha-6* RNAi. As an alternative approach to check lysosomal activity, we examined the maturation of cathepsin L (CPL-1 in *C. elegans*), which is synthesized as an inactive pro form and converted to the active mature form through proteolytic removal of the prodomain in

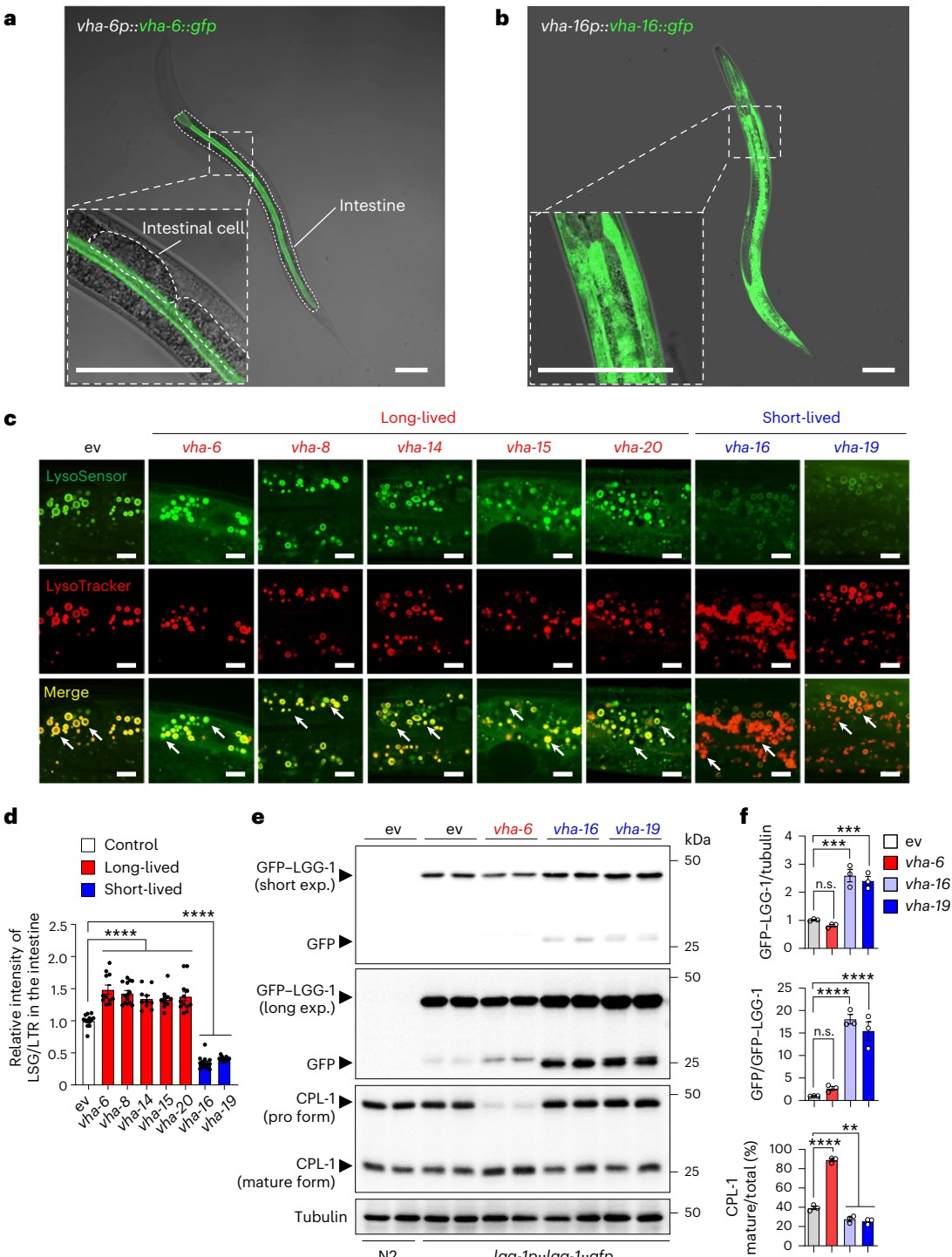

**Fig. 6 | LySR activation is featured by boosted lysosomal activity. a**,**b**, The expression and localization of GFP-tagged VHA-6 (**a**) and GFP-tagged VHA-16 (**b**) in transgenic worms. Scale bars, 0.1 mm. **c**, Confocal fluorescence images of the intestine of worms treated with RNAi targeting different v-ATPase subunits as indicated and then stained by LysoSensor Green (LSG) DND-189 and LysoTracker Red (LTR) DND-99. Each VHA RNAi occupies 20%; control RNAi was used to supply to a final 100% of RNAi for all conditions. The pictures in the same channels were taken at the same settings. Scale bars, 10 μm. **d**, The relative intensity of LSG/LTR in worms treated with RNAi as indicated in **c** was quantified (*n* = 13 worms for ev, *n* = 11 worms for *vha-6/vha-19*, *n* = 14 worms for *vha-8/vha-16*, *n* = 10 worms for *vha-14/vha-15*, *n* = 12 worms for *vha-20*) (\*\*\*\**P* < 0.0001).

**e**, The wild-type N2 or *lgg-1p::lgg-1::gfp* worms were treated with control, *vha-6*, *vha-16* or *vha-19* RNAi and analysed by western blots. exp., exposure. **f**, Statistical analyses (*n* = 3 independent experiments) of the relative GFP–LGG-1 expression versus tubulin, GFP versus GFP–LGG-1 and the percentage of mature form of CPL-1 as compared with the total CPL-1 in conditions as shown in **e** (\*\*\*\**P* < 0.0001; for GFP–LGG-1/tubulin, *P* = 0.6346 (not significant (n.s.), ev versus *vha-6*), \*\*\**P* = 0.0001 (ev versus *vha-16*), \*\*\**P* = 0.0003 (ev versus *vha-19*); for GFP/GFP– LGG-1, *P* = 0.6482 (n.s., ev versus *vha-6*); for CPL-1 mature/total (%), \*\**P* = 0.0048 (ev versus *vha-16*), \*\**P* = 0.0016 (ev versus *vha-19*)). The error bars denote the standard error of the mean. The statistical analysis was performed by ANOVA followed by Tukey's post hoc test.

lysosomes[19,59]. We found that CPL-1 maturation was robustly enhanced in *vha-6* RNAi worms (Fig. 6e,f). Together, these results suggest that *vha-6* RNAi boosts lysosomal activity in *C. elegans*.

### LySR reduces protein aggregates and extends healthspan

Lysosomal proteases, including the cathepsins[27], are central enzymes that are involved in the proteolytic degradation of misfolded and aggregation-prone proteins, such as amyloid-β (Aβ) and polyglutamine (polyQ)-expanded huntingtin (HTT), the contributing factors in the pathogenesis of Alzheimer's disease and Huntington's disease[60,61], respectively. We thus asked whether LySR activation could reduce Aβ proteotoxicity in vivo. The GMC101 strain is a worm Alzheimer's disease model that expresses the human Aβ1–42 peptide in the body-wall muscle cells[62]. GMC101 adults develop age-progressive paralysis and exacerbated amyloid deposition after a temperature shift from 20 °C to 25 °C. In response to the temperature shift, transcript levels of multiple LySR-associated lysosomal proteases increased in GMC101 worms (Extended Data Fig. 8a), suggesting that the LySR is induced concomitantly with proteotoxicity. Strikingly, RNAi of *vha-6* further increased the mRNA level of these proteases by more than tenfold (Fig. 7a) and reduced Aβ aggregates in GMC101 worms to an almost undetectable level (close to that in the non-Aβ expressing CL2122 control worms) at both 20 °C and 25 °C (Extended Data Fig. 8b), an effect that was abrogated by *elt-2* RNAi (Fig. 7b). Treatment with high-dose (5 mM) of lysosomal inhibitor chloroquine (CQ)[63], blunted *vha-6* RNAi-induced reduction of Aβ aggregates as well as lifespan extension (Extended Data Fig. 8c,d), confirming a lysosome-dependent regulatory mechanism. Interestingly, CQ treatment at 1 mM blocked aggregation clearance but not lifespan extension (Extended Data Fig. 8c,d), suggesting that longevity extension seems to be possible even when proteostasis is compromised. Importantly, the prototypical ageing-associated decline in movement and exacerbation in paralysis of GMC101 worms was also fully normalized by *vha-6* RNAi in an ELT-2-dependent manner (Fig. 7c).

Likewise, *vha-6* RNAi reduced the ageing-associated formation of polyQ and mutant superoxide dismutase 1 (SOD1) aggregates in *C. elegans* models of Huntington's disease and ALS[64,65], respectively (Fig. 7d,e and Extended Data Fig. 8e,f). These beneficial effects were furthermore attenuated by CQ in a dose-dependent manner (Extended Data Fig. 8g). For yet unknown mechanisms, CQ partially rescues the small size phenotype of *vha-6* RNAi animals, suggesting that the small size might correlate with worm health in this specific context (Extended Data Fig. 8g). Moreover, fitness, evaluated by alleviated paralysis and increased movement, was also improved by *vha-6* RNAi in the polyQ or ALS animal models (Extended Data Fig. 8h). Furthermore, in a pan-neuronal human Aβ1-42 expressing worm strain GRU102, which displays age-dependent neuromuscular behaviour/memory impairments similar to Alzheimer's disease pathogenesis[66], *vha-6* RNAi improved the intermediate-term memory[67] of GRU102 worms to a level close to that in the non-Aβ expressing control worms (Fig. 7f). Interestingly, *vha-6* RNAi even had a modest but statistically significant improvement on memory in control animals (Fig. 7f).

To further reveal which lysosomal proteases are responsible for *vha-6* RNAi-induced beneficial effects, we used RNAi targeting each of the eight typical *vha-6* RNAi-induced cathepsin proteases (that is, *cpr-5*, *cpr-8*, *F57F5.1*, *F32F5.1*, *ctsa-1.2*, *K10C2.1*, *ctsa-3.1* and *ctsa-1*), as well as three aspartic-type endopeptidases (that is, *asp-1*, *asp-8* and *asp-10*) (Fig. 1k and Extended Data Fig. 9a–c), and tested their impact on *vha-6* RNAi-induced Aβ aggregate clearance. The results revealed that silencing of *cpr-5* but not other proteases or endopeptidases partially blocked *vha-6* RNAi-induced beneficial effects on Aβ aggregates clearance (Fig. 7g). In line with this result, RNAi of *cpr-5* attenuated *vha-6* RNAi-induced lifespan extension, as well as polyQ and SOD1 aggregation clearance (Fig. 7h and Extended Data Fig. 9d–g). Together, these results pinpointed the lysosomal cathepsin protease, CPR-5, as a major executing factor for *vha-6* RNAi-induced beneficial effect on

aggregation clearance in multiple neurodegenerative disease models, as well as lifespan extension.

The identified LySR transcription factor ELT-2 was found almost exclusively expressed in intestine cells (Extended Data Fig. 9h). Thus, the question arises: How does CPR-5, a cathepsin B-like cysteine protease that is also specifically expressed in the intestine[68], mediate the systemic function of LySR in aggregation clearance? We hence checked whether CPR-5 can be secreted and affect other tissues. In *C. elegans*, coelomocytes are scavenger cells that take up secreted materials from the body cavity and serve as a monitor of secreted proteins[69]. We generated a transgenic strain expressing an intestine-specific polycistronic transcript encoding both CPR-5–Discosoma striata red (DsRed) fluorescent fusion protein and GFP, such that GFP indicates cells expressing *cpr-5* and DsRed directly labels the CPR-5 protein. Without tagging any proteins, GFP was detected only within intestinal cells (Extended Data Fig. 9i). CPR-5–DsRed fusion, on the other hand, was detected within both intestinal cells and also in coelomocytes (Extended Data Fig. 9i), indicating CPR-5 secretion from the intestine into the body cavity. This secretion of CPR-5-DsRed is furthermore enhanced in the long-lived *vha-6* RNAi animals but not in the short-lived *vha-16* or *vha-19* RNAi worms (Extended Data Fig. 9j).

In line with previous studies[19,33,34], the transcripts of the lysosomal protease genes, including *cpr-5*, *cpr-8*, *ctsa-1* and *asp-10*, as well as *vha-6* and *elt-2*, were progressively downregulated with age in control RNAi worms (Fig. 7i,j). By contrast, we revealed that, in the long-lived *vha-6* RNAi worms, the expression levels of these LySR-related transcripts, as well as movement, were increased and generally sustained later in adult life (at least untill adult day 8), a phenomenon entirely blunted by *elt-2* RNAi (Fig. 7i–k). Collectively, these results highlight that activation of LySR by *vha-6* RNAi reduces protein aggregates and extends organismal healthspan.

### HLH-30 and PHA-4 in LySR activation and lifespan extension

HLH-30, the worm orthologue of TFEB[35], has been shown to regulate autophagy and lysosomal homoeostasis by targeting lysosomal genes including *cpr-1*, *ctsa-3.2* and *asp-1* (ref. 36). In line with the results using the *cpr-5p::gfp* reporter (Extended Data Fig. 3d), knockdown of *hlh-30*, did not affect *vha-6* RNAi-induced upregulation of typical LySR targets including *cpr-5* and *cpr-8*, in the GMC101 Aβ-expressing worms (Extended Data Fig. 10a). Meanwhile *hlh-30* knockdown also failed to block the *vha-6* RNAi-induced reduction of Aβ, polyQ and SOD1 aggregations (Extended Data Fig. 10b–f), although we cannot fully rule out the possibility that the residual *hlh-30* may still be functional. Interestingly, *vha-6* RNAi-induced lifespan extension was strongly attenuated in the *hlh-30* mutant (Extended Data Fig. 10g). One possible explanation is that HLH-30 may act downstream of the LySR to regulate longevity. Alternatively, HLH-30 may also act in parallel with ELT-2 but mainly executes part of the LySR response that is required for the longevity response. In support of the alternative explanation, *hlh-30* mRNA increases by ~120% in response to *vha-6* RNAi (Extended Data Fig. 10a), a process that is largely not affected by *elt-2* RNAi (Supplementary Table 3). By contrast, inactivation of the FoxA orthologue *pha-4*, which is required for DR-induced longevity[70], barely affects the lifespan extension induced by *vha-6* RNAi (Extended Dpcata Fig. 10h).

## Discussion

Extensive work has described and characterized key transcriptional responses to promote the homoeostasis of mitochondria and ER and regulate ageing in multiple organisms[23,25,71–73]. By contrast, little is known about the pathways to surveil and boost the function of lysosomes to counteract ageing and ageing-associated diseases[74]. Here, we reveal a longevity-linked lysosomal surveillance response (LySR) that can be activated by RNAi of specific intestinal apical membrane-localized v-ATPase subunits (for example, *vha-6* RNAi) in *C. elegans* (Fig. 7l). A major function of the intestinal apical membrane-localized v-ATPase

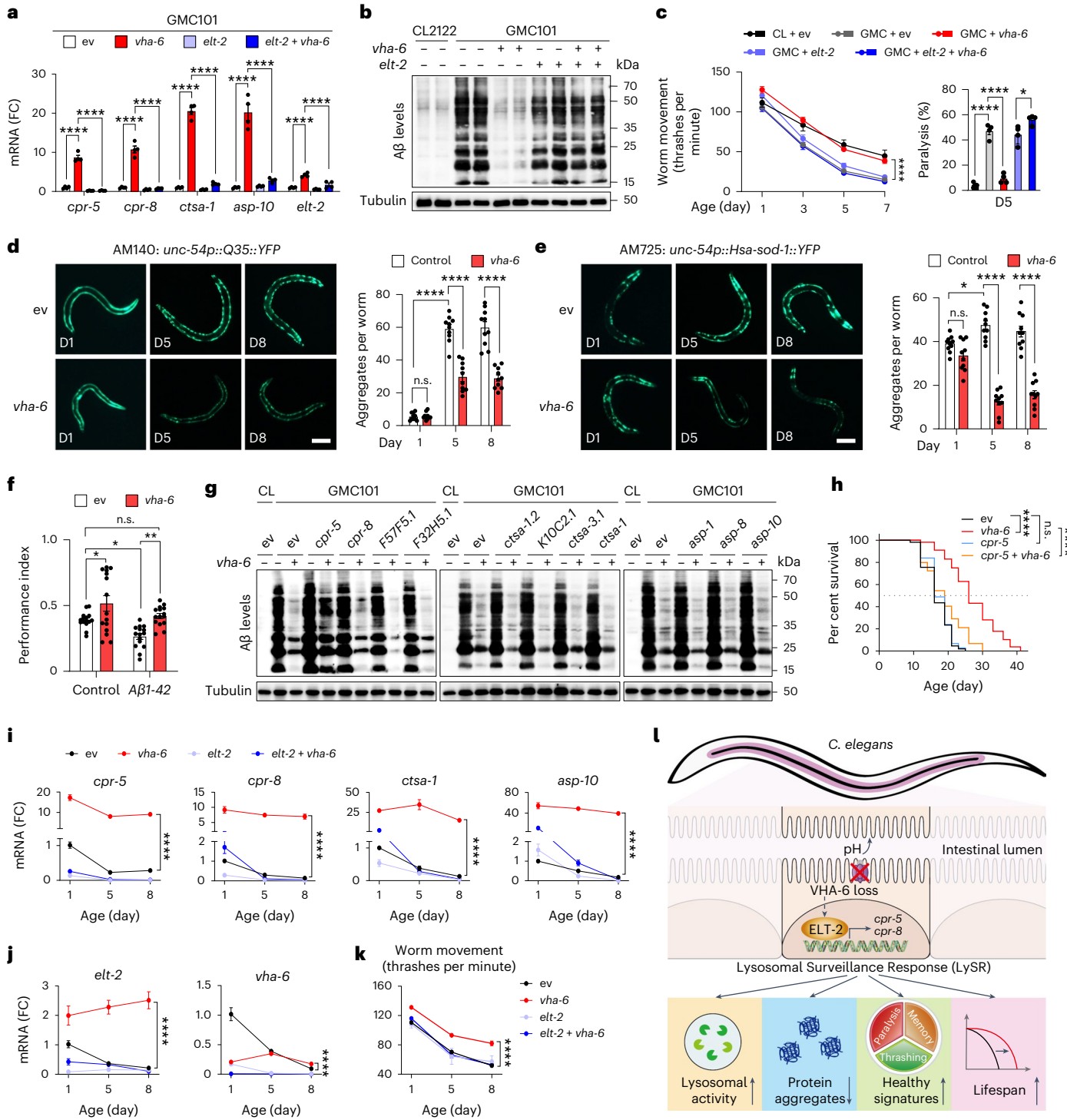

**Fig. 7 | Activation of LySR reduces protein aggregates and extends healthspan.**
**a–c**, A qRT–PCR analysis (n = 4 biologically independent samples) (**a**), western blots (**b**), movement (n = 12 individual worms for each condition) and paralysis (n = 4 independent experiments) (**c**) of CL2122 or GMC101 worms treated with control, *vha-6* and/or *elt-2* RNAi (****P < 0.0001; in **c**, *P = 0.0240 (GMC + *elt-2* versus GMC + *elt-2* + *vha-6*)). **d,e**, RNAi of *vha-6* (20%) reduces the aggregate formation in *unc-54p::Q35::YFP* (polyQ model) (**d**) and *unc-54p::Hsa-sod-1::YFP* (ALS model) worms (n = 10 individual worms for each condition) (****P < 0.0001; in **d**, P > 0.9999 (not significant (n.s.), day 1 (D1) ev versus D1 *vha-6*); in **e**, P = 0.3577 (n.s., D1 ev versus D1 *vha-6*), *P = 0.0358 (D1 ev versus D5 ev)). Scale bars, 0.2 mm. **f**, *vha-6* RNAi improves intermediate-term memory in the worm Alzheimer's disease model GRU102 (*unc-119p::Aβ1-42*) strain with constitutive neuronal Aβ1-42 expression, analysed at D4 adulthood (n = 15 chemotaxis assays of 50–100 worms for each condition) (*P = 0.0476

(ev versus *vha-6*), *P = 0.0498 (control versus *Aβ1-42*), P = 0.9153 (n.s., control + ev versus *Aβ1-42* + *vha-6*), **P = 0.0089 (*Aβ1-42* + ev versus *Aβ1-42* + *vha-6*)).
**g**, Western blots of CL2122 or GMC101 worms treated with control or *vha-6* (20%) RNAi combined with RNAi targeting lysosomal protease genes (80%).
**h**, The lifespan of N2 worms treated with control, *cpr-5* and/or *vha-6* RNAi (****P < 0.0001, P = 0.2547 (n.s., ev versus *cpr-5*)). **i–k**, The mRNA levels of indicated genes (n = 4 biologically independent samples) (**i,j**) and movement (n = 12 individual worms for each condition) (**k**) of N2 worms treated with control or *vha-6* RNAi, in combination with *elt-2* RNAi, collected at different ages (****P < 0.0001). **l**, The proposed model for LySR activation and regulation. The error bars denote the standard error of the mean. The statistical analysis was performed by ANOVA followed by Tukey post hoc test in **a**, **c–f** and **i–k** or a log-rank test in **h**. The statistical data for lifespan can be found in Supplementary Table 1.

subunits appears to be the acidification of the intestinal lumen but not that of the lysosomes. Typified by the induction of a large panel of lysosome/proteolysis-related genes (for example, *cpr-5*) and regulated by the GATA transcription factor ELT-2, LySR activation improves proteostasis, reduces protein aggregates and extends healthspan, as well as lifespan in several *C. elegans* models of neurodegenerative diseases and of normal ageing. Importantly, the beneficial effects of *vha-6* RNAi strongly depend on the intact function of lysosomes (Extended Data Fig. 8c,d,g), as well as the lysosomal protease CPR-5 (Fig. 7g,h and Extended Data Fig. 9d–g), confirming a key role of the LySR targeting lysosome/proteolysis genes in healthspan determination.

How the RNAi of different v-ATPase subunits, which, in theory, are equally important for the function of the giant v-ATPase complex[16], leads to distinct changes in lifespan, lysosomal and intestinal lumen pH, and gene expression remains an interesting topic for future work. For example, *vha-6*, *vha-16* and *vha-19* all encode one of the subunits of the v0 domain of v-ATPase, but only *vha-6* RNAi extends animal lifespan (Fig. 1a,f,g). One plausible explanation is that different v-ATPase subunits are expressed in different cell types or separate membrane compartments and that some subunits are functionally similar to each other[16,75]. Indeed, based on the GFP reporter expression patterns of v-ATPase subunits, most v-ATPase subunits have distinct tissue-specific expression in *C. elegans*; some are primarily expressed in the H-shaped excretory cell (for example, *vha-1*, *vha-2*, *vha-4*, *vha-5*, *vha-11* and *vha-17*)[16,75–77]; *vha-6* is almost exclusively expressed in the intestine[17,75] (Fig. 6a and Extended Data Fig. 7a); *vha-7* is enriched in the hypodermis, uterus and spermatheca[75]; *vha-8* is highly expressed in the hypodermis, intestine and excretory cells[78]; *vha-15* is expressed in the muscle, intestine and neurons[79] (Extended Data Fig. 7c); *vha-20* is expressed in the intestine, excretory cell and amphid neurons[18] (Extended Data Fig. 7d); and *vha-16* and *vha-19* are widely present in tissues including the excretory cell, hypodermis, pharynx and vulva[80,81] (Fig. 6b). Thus, distinct v-ATPase complexes consisting of different v-ATPase subunits, at least in the case of the four highly similar 'a' subunits VHA-5, VHA-6, VHA-7 and NUC-32 (ref. 75), are probably assembled for the acidification of cell-specific intracellular and extracellular compartments in vivo.

Although initially discovered as an adaptive response to *vha-6* RNAi, the LySR pathway is activated or suppressed at least in three relevant physiological contexts. First, it is activated concomitantly with proteotoxicity in the Alzheimer's disease worm model GMC101 (Extended Data Fig. 8a), suggesting that organisms/cells may utilize this pathway against toxic protein aggregations. Second, its activity progressively downregulates with ageing (Fig. 7i,j), highlighting the loss of proteostasis as one of the hallmarks and driving factors for ageing[82]. Finally, DR partially hijacks the LySR pathway to promote longevity, as evidenced by the induction of LySR targeting gene *cpr-5*/*cpr-8* and ELT-2-dependent lifespan extension (Fig. 5b–e).

Despite the partial activation of LySR downstream of DR, our continued investigations do not support that *vha-6* RNAi or intestinal lumen pH disruption represents another form of DR. In fact, among the 1,020 genes upregulated upon sDR, only 6.3% (64) genes overlapped with the 760 LySR genes (Fig. 5f and Supplementary Table 4), suggesting two largely different transcriptional responses. Moreover, AAK-2 and DAF-16, two essential regulators for sDR-induced lifespan extension[51], are not required for *vha-6* RNAi-induced longevity (Fig. 2h,j). Thus, even though AMPK and DAF-16 are not required for all longevity-induced DR methods (for example, liquid bacterial DR)[83], we conclude that DR only partially explains the LySR activation and subsequent lifespan extension in response to *vha-6* RNAi or intestinal lumen pH disruption. Indeed, with multiple acid-sensing ion channels existing in the intestine of *C. elegans*[84], the acid environment by itself—in the form of protons—may directly serve as signalling factors that mediate the activation of pathways in both neuronal and non-neuronal cells[85,86].

Similar to the UPR^mt induced by *cco-1* RNAi[23], the LySR activated by *vha-6* RNAi is also probably cell-non-autonomous in *C. elegans*, especially considering that *vha-6* is predominantly expressed in the intestine while the aggregation-prone proteins in the Alzheimer's disease, Huntington's disease and ALS worm models applied in the current study were all expressed under the muscle-specific *unc-54* promoter[16,62,64,65]. In support of this model, CPR-5, whose expression is considered to be intestinally restricted, was also detected in other tissues such as coelomocytes (Extended Data Fig. 9i); this effect is furthermore enhanced in response to *vha-6* RNAi (Extended Data Fig. 9j). Consistently, in mammalian systems, many cathepsins have been shown to be secreted into the extracellular space or serum to mediate tissue-to-tissue crosstalks and controlling a wide range of physiological processes[27,87]. Therefore, peripheral cathepsin secretion may function as an evolutionarily conserved mechanism to facilitate intertissue communication and promote protein aggregate clearance cell-non-autonomously in the context of LySR activation.

Admittedly, an exact mechanism linking VHA-6 loss to ELT-2 activation and subsequent aggregation clearance in peripheral tissues is still lacking in our current study. One potential explanation is that extracellular cathepsins could reach muscle-specific aggregates via endocytosis of the secreted protease followed by autophagic degradation. However, autophagosomes typically fuse with endogenous lysosomes, which may already contain abundant proteases, including cathepsins, probably at concentrations exceeding those contributed by internalized extracellular cathepsins. Another plausible scenario is that VHA-6 loss could trigger systematic metabolic adaptations facilitating intertissue coordination. In support of such a hypothesis, gene sets related to 'innate immune response' and 'metabolic pathways' were significantly enriched following *vha-6* RNAi treatment, in addition to genes involved in 'lysosome/proteolysis' (Fig. 1j). Intriguingly, a recent study demonstrated that enhanced lysosomal lipolysis via intestinal *lipl-4* overexpression activates a neuropeptide signalling pathway in the nervous system to promote longevity[69]. Consistent with this, our findings indicate that *vha-6* silencing markedly elevates the expression of multiple lysosomal lipases, including *lipl-1*, *lipl-2*, *lipl-4* and *lipl-6* (Supplementary Table 3). Thus, the neuropeptide signalling pathway may represent another mechanism mediating intertissue communication upon *vha-6* RNAi. Finally, transcription factors such as MXL-3, HLH-30 and DAF-16, which orchestrate adaptive responses to nutritional status by regulating lysosomal lipolysis[88,89], may furthermore coordinate with ELT-2 to determine organismal healthspan in the context of LySR activation.

Key components in this LySR pathway are well conserved in mammals, suggesting that a similar mechanism may also exist in mammalian cells. As a case in point, cathepsin B, one of the crucial enzymes involved in the degradation of neurotoxic proteins in Alzheimer's disease, Huntington's disease and ALS mouse models[90–92], belongs to the LySR network. Further investigation is therefore warranted to explore whether targeting the LySR pathway to boost lysosomal function and reduce proteotoxicity may also provide protection against normal ageing and neurodegenerative diseases in other organisms in vivo.

## Online content

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

## Methods

### *C. elegans* strains

The N2 (Bristol) strain was employed as the wild-type strain. *IA123 (ijIs10[cpr-5::GFP-NLS::lacZ + unc-76( + )])*, *CB1370 [daf-2(e1370)]*, *CF1038 [daf-16(mu86)]*, *VC222 [raga-1(ok386)]*, *RB754 [aak-2(ok524)]*, *DA465 [eat-2(ad465)]*, *VC3201 [atfs-1(gk3094)]*, *OP56 (gaEx290 [elt-2::TY1::EGFP::3xFLAG(92C12) + unc-119(+)])*, *CL2122 (dvIs15 [pPD30.38] unc-54(vector) + (pCL26) mtl-2::GFP)*, *GMC101 (dvIs100 [unc-54p::A-beta-1-42::unc-54 3'-UTR + mtl-2p::GFP])*, *AM140 (rmIs132 [unc-54p::Q35::YFP])*, *AM725 (rmIs290 [unc-54p::Hsa-sod-1(127X)::YFP])*, *DA2123 (adIs2122 [lgg-1p::GFP::lgg-1 + rol-6(su1006)])*, *GRU101(gnaIs1[myo-2p::yfp])*, *CA1200 (ieSi57 [eft-3p::TIR1::mRuby::unc-54 3'UTR + Cbr-unc-119(+)] II)*, *JIN1375 [hlh-30(tm1978) IV]*, *atfs-1(tm4525) V*, *SM190 [pha-4(zu225); smg-1(cc546ts)]*, *GRU102 (gnaIs1[myo-2p::yfp + unc-119p::Aβ_{1-42}])*, *HZ1683 [atg-2(bp576)]*, *HZ1684[atg-3(bp412)]*, *HZ1687[atg-9(bp564)]* and *HZ1688 [atg-13(bp414)]* were provided by the *Caenorhabditis* Genetics Center (CGC, University of Minnesota) or the National Bioresource Project (NBRP). The *XW19180 [hsp-16.2p::nuc-1::pHTomato]* strain was a kind gift from Professor Xiaochen Wang (SUSTech). The *MQD2491[daf-16(hq389[daf-16::gfp::degron]) I; ieSi57[eft-3p::TIR1::mRuby::unc-54 3'UTR + Cbr-unc-119(+)] II; unc-119(ed3) III; daf-2(e1370ts) III]* strain was a kind gift from Professor Meng-Qiu Dong (NIBS).

For generation of the strains with GFP-tag of *vha-1*, *vha-14*, *vha-15*, *vha-16* and *vha-20*, the constructs (*vha-1*, clone: 9473457628999774 E12; *vha-14*, clone: 8859124759762056 C08; *vha-15*, clone: 3304493 055384826 B08; *vha-16*, 2491680425634929 G12; *vha-20*, clone: 5745981749165295 F12) were obtained from Professor Mihail Sarov, as part of the TransgeneOme project (https://transgeneome.mpi-cbg. de/). The constructs were injected at 10–60 ng μl⁻¹ along with a coinjection marker pRF4 (*rol-6*) at 40 ng μl⁻¹ to generate transgenic lines. These strains were made by the SunyBiotech.

A new ultraviolet-integrated N2 background *cpr-5* reporter strain *TYL001 (cpr-5p::gfp + rol-6)* for optimal LySR activation detection was also constructed, which is available upon request. To construct this strain, the 1018 bp *cpr-5* promoter was amplified with the following primers: 5'- GAATTGACATGCACTCCGGC-3' and 5'-AAGAATAGCGGAGAGCTTCC-3' and ligated in frame with eGFP in a pPD95.75 expression vector (Addgene, #184130). The construct was then injected at 50 ng μl⁻¹ along with a coinjection marker pRF4 (*rol-6*) at 40 ng μl⁻¹. The extrachromosomal arrays were integrated using ultraviolet irradiation and backcrossed two times to N2, non-roller worms were maintained afterwards.

For generation of the *TYL002 (ges-1p::cpr-5-DsRed::SL2::GFP + rol-6)* worm strain, the *ges-1* promoter (amplified from the pJL3 plasmid, Addgene #184131), the *cpr-5* protein coding sequence (amplified from worm total complementary DNA (cDNA)) and DsRed sequence (amplified from the pJL6 plasmid, Addgene #184134), were ligated into a pPD95.77_SL2 vector backbone (Addgene #184129), between SphI and XmaI restriction sites. The construct was then injected at 25 ng μl⁻¹ along with a coinjection marker pRF4 (*rol-6*) at 40 ng μl⁻¹ to generate transgenic lines.

For generation of the knockin worm strains with endogenously GFP/Degron-mNG tagged VHA-6 [*TYL003 (vha-6p::vha-6::gfp)* and *TYL004 (vha-6::Degron::mNG)*] and ELT-2 [*TYL005 (elt-2::Degron::mNG)*], the CRISPR–Cas9 engineering was performed by microinjection using the homologous recombination approach[43]. The microinjection mixture consisted of 300 mM of KCl, 20 mM of HEPES, 100 ng μl⁻¹ of trans-activating CRISPR RNA (cat. no. U-002005, Dharmacon), 50 ng μl⁻¹ of CRISPR RNA (crRNA) targeting *vha-6* or *elt-2*, 200 ng μl⁻¹ of DNA repair template for *vha-6* or *elt-2*, 0.25 μg μl⁻¹ of Cas9 protein (cat. no. CAS9PROT-250UG, Sigma), 200 ng μl⁻¹ *dpy-10* crRNA and 200 ng μl⁻¹ *dpy-10* repair template. To generate the homologous recombination DNA repair templates, two homologous arms (~1,000 bp each) corresponding to the 5' and 3' sides of the insertion site, and

the GFP/Degron-mNG tags were cloned in a vector and then amplified altogether. The plasmids were injected into the gonad of young adult hermaphrodite worms using the standard method. F1s with roller phenotype were singled on a new nematode growth medium (NGM) plate and allowed to produce sufficient offspring. The successful knockin events were screened by PCR genotyping from independent F1 transgenic animals' progeny that did not display roller phenotype and further confirmed by DNA sequencing. The crRNAs and cloning primers used to generate the two strains are listed in Supplementary Table 1.

For generation of the strains expressing *vha-6* and *elt-2* promoter-driven mCherry *TYL006 (vha-6p::mCherry; vha-6p::vha-6::gfp)* and *TYL007 (elt-2p::mCherry; elt-2::Degron::mNG)*, promoters of *vha-6* or *elt-2* were amplified with the following primers: 5'-TCGG TAAGTTGCTACTTCAG-3' and 5'-TTTTTATGGGTTTTGGTAGGTTTTAG-3' for *vha-6* promoter and 5'-ATTATATGAAAACTAATGAG-3' and 5'-TCT ATAATCTATTTTCTAGTTTCTATTTTATT-3' for *elt-2* promoter. The PCR products were then ligated in frame with mCherry in a pPD95.75 expression vector (Addgene, #184130). The constructs were then injected into the gonad of their corresponding *TYL003 (vha-6p::vha-6::gfp)* and *TYL004 (elt-2::Degron::mNG)* strains at 50 ng μl⁻¹ along with a coinjection marker pRF4 (*rol-6*) at 40 ng μl⁻¹.

All worm strains were routinely maintained at 20 °C (except for the *CB1370 [daf-2(e1370)]* strain, which was maintained at 15 °C; the *SM190 [pha-4(zu225);smg-1(cc546ts)]* strain, which was maintained at 25 °C) on NGM or high growth medium (NGM recipe modified as follows: 20 g l⁻¹ Bacto-peptone, 30 g l⁻¹ Bacto-agar and 4 ml l⁻¹ cholesterol (5 mg ml⁻¹ in ethanol); all other components same as NGM) plates, with *Escherichia coli* OP50 as the food source[93].

### RNAi

For RNAi experiments, the worms were fed with *E. coli* strains HT115(DE3) containing an empty vector L4440 or expressing double-strand RNAi. The RNAi clones were obtained from either the Ahringer or Vidal library and verified by sequencing or quantitative real-time polymerase chain reaction (qRT–PCR) before use. The *vha-6* RNAi clone from the Vidal library (11038-D9, *vha-6* RNAi_1) was used for all experiments related to *vha-6* RNAi unless otherwise indicated. The other two *vha-6* RNAi clones used were both from the Ahringer library with the accession codes: II-7F06 for *vha-6* RNAi_2, and II-7F04 for *vha-6* RNAi_3. The two *cbp-1* RNAi clones used were as described previously[49].

RNAi clones for *elt-4*, *elt-6*, *egl-27*, *daf-2* and *cpr-5* were constructed by PCR amplification of cDNAs from total RNA with the following primers: *elt-4*_RNAi_Fw: 5'-TAGATGCTTCTCATCGGAAACGG-3', *elt-4*_RNAi_Rv: 5'-CAGTTTCGAAATGCCAGGAGC-3'; *elt-6*_RNAi _ Fw: 5'-GATGCGCTCAGCTTCACAAG-3', *elt-6*_RNAi_Rv:5'-GAAAACGG CTGCTTGACTGG-3'; *egl-27*_RNAi_Fw: 5'-ACAAGAACGAGCTGA GCTTGAA-3', *egl-27*_RNAi_Rv: 5'-AAAGACCGTTTGCGTGATGC-3'; *daf-2*_RNAi_Fw: 5'-GCTCTCGGAACAACCACT GA-3', *daf-2*_RNAi_ Rv: 5'-GTCGCATCATTCACACGCTC-3'; *cpr-5*_RNAi_Fw: 5'-GCTGT GGTGATTCCTGGACA-3', *cpr-5*_RNAi_Rv: 5'-CCCATCCGAGGAT CTTGACG-3'. The PCR products were then ligated into the L4440 empty vector and transformed into *E. coli* HT115 competent cells.

For RNAi feeding, the RNAi bacteria were inoculated and cultured in lysogeny broth medium with 100 μg ml⁻¹ ampicillin overnight on a shaker at 37 °C. And then the bacteria were seeded onto RNAi plates (NGM containing 2 mM isopropyl β-ᴅ-thiogalactopyranoside and 25 mg ml⁻¹ carbenicillin) and allowed to form a dry bacterial lawn. The experiments with mixed RNAi were achieved by mixing bacterial cultures, normalized to their optical densities measured at optical density at 600 nm (OD_{600}) before seeding.

### Worm alignment-based imaging

For worm alignment-based imaging, the worms at the last larval stage (L4) were picked and transferred onto the RNAi bacteria-seeded plates and incubated at 20 °C to allow overnight egg laying. After 24 h for

worm development and egg laying, adult worms were removed from the plates. When the eggs were grown and developed into young adults, eight to ten worms were randomly picked and aligned after being placed in a drop of 10 mM tetramisole (cat. no. T1512, Sigma) shortly. Fluorescent photos were taken with the same exposure time for each condition using a Nikon SMZ1000 microscope. For aggregate quantification, after the AM140 and AM725 eggs reached L4 stage, the worms were washed off the plate and transferred onto RNAi plates and allowed to develop to the desired age. The worms were randomly picked and imaged after submerging in a drop of 10 mM tetramisole. The aggregates were counted for each worm on day 1, 5 and 8 of adulthood. The GFP intensity of worms was analysed by using the ImageJ/Fiji 1.53c software.

### Auxin treatment
Auxin treatment was performed by transferring worms to bacteria-seeded plates containing the natural auxin indole-3-acetic acid (IAA) (cat. no. A10556, Alfa Aesar), NAA (cat. no. HY-18570, MCE) or 5-Ph-IAA (cat. no. HY-134653, MCE), as described previously[42]. For IAA, a 400 mM stock solution in ethanol was prepared and stored at 4 °C for up to 1 month. Auxin was diluted into the NGM agar and cooled to about 50 °C before pouring plates. A fresh HT115 bacterial culture was highly concentrated and spread on plates. The plates were then left at room temperature for 1–2 days to allow bacterial lawn growth.

### CQ treatment
For CQ treatment, CQ (cat. no. C6628, Sigma) was dissolved in M9 buffer (6 g l$^{-1}$ Na$_2$HPO$_4$, 3 g l$^{-1}$ KH$_2$PO$_4$, 5 g l$^{-1}$ NaCl and 1 ml l$^{-1}$ 1 M MgSO$_4$ in distilled water) at 400 mM and used as the stock. CQ at a final concentration of 1 mM or 5 mM was added to the NGM just before pouring the plates. After RNAi bacteria seeding, synchronized worm eggs obtained by bleaching were then transferred onto the NGM plates and collected at L4/young adult stage for western blots. In *C. elegans*, certain mM levels of CQ are required to functionally inhibit the lysosomal activity, as described previously[94,95].

### Oregon Green-dextran 488, LysoSensor and LysoTracker staining
Oregon Green-dextran 488, LysoSensor and LysoTracker staining for *C. elegans* were carried out as described previously[17,19]. The worms were treated with control RNAi or RNAi targeting different v-ATPase subunits until L2/L3 stage. The worms were then soaked in 80 µl of S-basal buffer containing 5 mg ml$^{-1}$ Oregon Green-dextran 488 (cat. no. D7172, ThermoFisher) or 10 µM LysoSensor Green DND-189 (cat. no. L7535, ThermoFisher) and 10 µM LysoTracker Red DND-99 (cat. no. L7528, ThermoFisher). Staining was carried out for 2 h for Oregon Green-dextran 488 and 1 h for LysoSensor/LysoTracker, at 20 °C in the dark. For Oregon Green-dextran 488 staining, the worms were then washed two times in the S-basal buffer and immediately examined. For LysoSensor/LysoTracker staining, the worms were then transferred to NGM plates with fresh OP50 and allowed to recover at 20 °C for 1 h in the dark before examination using a ZEISS LSM 980 with Airyscan 2 confocal microscope.

### ELT-2::GFP and DAPI imaging and quantification
The staining and imaging of DAPI in worms was performed as described previously[96]. Briefly, *elt-2p::elt-2::gfp-flag* worms treated with control or *vha-6* RNAi were fixed with ethanol and stained with DAPI at a final concentration of 2 ng µl$^{-1}$. The worms were then mounted on 2% agarose pads and imaged at 63× using a ZEISS LSM 700 confocal microscope. A quantification of the DAPI signal was performed using ImageJ/Fiji 1.53c software as described[97], the image voxels were ranked by DAPI intensity within each nucleus and divided into four equal-volume bins. The percentage of total DAPI intensity in each of the bins was then quantified. Analyses were performed in at least 30 nuclei for each condition.

### DR in *C. elegans*
DR of *C. elegans* was achieved by feeding worms with serially diluted HT115 bacteria or no bacteria, as described previously[51,52]. Briefly, synchronized worm eggs obtained by bleaching were treated with HT115 control bacteria until the L4 stage. The worms were then transferred onto NGM plates seeded with HT115 bacteria carrying either empty vector or RNAi clones. One day later, the adult day 1 worms were transferred onto NGM plates containing 10 µg ml$^{-1}$ kanamycin (to prevent bacteria from further growing) and serially diluted HT115 bacteria concentrations (BCs) ranging from $1.2 × 10^{10}$ colony-forming unit (c.f.u.) ml$^{-1}$ to $1.2 × 10^6$ c.f.u. ml$^{-1}$ or no bacteria. The BC was obtained by measuring OD$_{600}$. The relation between OD$_{600}$ and BC was determined by colony formation assay (BC = OD$_{600}$/0.0121 × 10$^7$ c.f.u. ml$^{-1}$). For 35 mm plates, 100 µl of bacteria were added; for 90 mm plates, 800 µl of bacteria were added. At adult day 2, the worms were either collected for RNA extraction or continued to be maintained for lifespan analysis.

### Lifespan and paralysis analysis
Lifespan assays were conducted as described in the previous study[98]. Briefly, five to ten L4 hermaphrodite worms were randomly picked from maintenance plates and transferred onto plates seeded with the indicated RNAi bacteria. After 24 h for worm development and egg laying, the adult worms were removed from the plates. The synchronized larvae were raised at 20 °C until they developed into L4 worms. A total of 80–100 L4 worms were randomly picked and transferred onto RNAi plates seeded with HT115 *E. coli* carrying either empty vector or RNAi clones. The worms were transferred every 24 h until the day that no eggs were produced. After that, the animals were transferred once a week. Those escaped from the plates or had vulva explosions were censored from the assay. To remove potential confounding effects, lifespans were examined in a condition without 5-FU. To ensure reproducibility, all lifespans were examined in at least three biological replicates with 80–100 worms in each replicate. A paralysis analysis was manually scored after poking, at least 80 total worms were analysed for each condition. Statistical analyses and details of replication for all lifespan experiments conducted in the current study are provided in Supplementary Table 1.

### Temperature-sensitive inactivation of *pha-4*
SM190 *[pha-4(zu225);smg-1(cc546ts)]* double mutant worms were grown at 25 °C to inactivate *smg-1* and allow *pha-4* expression. *pha-4* was inactivated by shifting the double mutants to 15 °C, restoring *smg-1* activity, which results in degradation of the *pha-4(zu225)* allele, after the first day of adulthood, thus avoiding any developmental defects due to loss of *pha-4* during larval stages, as described previously[70]. All control worms were treated identically.

### Quantification of NUC-1::pHTomato intensity
To induce the expression of NUC-1::pHTomato, *XW19180 [hsp-16.2p::nuc-1::pHTomato]*, worms were incubated at 33 °C for 30 min and recovered at 20 °C for 24 h before examination using a ZEISS LSM 980 confocal microscope, as described previously[19]. The average intensity of pHTomato per lysosome in the hypodermis was quantified by Image J/Fiji (v1.47b).

### RNA extraction and RNA-seq analysis
For worm samples, the synchronized worm eggs obtained by bleaching were transferred onto RNAi plates and cultured for 2.5 days at 20 °C to allow developing to L4/young adult stage. The worms were washed off the plates with M9 buffer three times, and the worm pellets were snap frozen in liquid nitrogen. To extract total RNA, 1 ml of TriPure Isolation Reagent (cat. no. 11667165001, Roche) was pipetted to each worm sample. The cell membranes were ruptured by freezing with liquid nitrogen and thawing in a water bath (37 °C) quickly eight times. And then, the total RNAs were extracted using a column-based kit (cat. no.

740955.250, Macherey-Nagel). For cells, 1 ml of the TriPure Isolation Reagent (cat. no. 11667165001, Roche) was directly added to the cells, and then cell homogenate was transferred to a 1.5 ml Eppendorf tube followed by using the same kit to extract total RNA. RNA-seq was performed by Beijing Genomics Institute with the BGISEQ-500 platform. To analyse the RNA-seq results, FastQC (version 0.11.9) was used to verify the quality of the sequence data. Adaptor sequences, contamination as well as low quality (Phred score <20) reads were filtered out from the raw data. Then, qualified reads were mapped to the worm '*Caenorhabditis_elegans*.WBcel235.89' genome with STAR aligner version 2.6.0a and counted by htseq-count version 0.10.0 using the following flags: -f bam -r pos -s no -m union -t exon -I gene_id. Limma-Voom was used to calculate gene differential expression. The genes with a Benjamini–Hochberg adjusted *P* value < 0.05 and with either log$_2$ fold change (log$_2$FC) >1 or log$_2$FC <−1 were considered as significantly upregulated or downregulated. The genes with significantly upregulated (adjusted *P* value < 0.05, log$_2$FC > 1) expression in the *vha-6* RNAi condition and were then downregulated by more than 25% of the log$_2$FC after *elt-2* RNAi cotreatment, compared with the log$_2$FC of the *vha-6* RNAi condition, were considered as ELT-2-dependent. A functional clustering was performed with the Database for Annotation, Visualization and Integrated Discovery (DAVID) (v6.8)[99]. The heat maps were created using Morpheus (https://software.broadinstitute.org/morpheus).

### Binding motif enrichment analysis

The 760 upregulated genes upon *vha-6* RNAi but not upon *vha-16* or *vha-19* RNAi were extracted from the RNA-seq and used as the input dataset. To identify the motifs significantly enriched for the promoters of these genes, the motif enrichment analysis was performed with HOMER (v4.11)[100], using the findMotifs.pl script (with start: −2,000 bp; end: 2,000 bp). The most enriched de novo motif of the input genes was compared against a library of known motifs downloaded from the Cis-BP database (catalogue of inferred sequence binding preferences)[101], using PWMEnrich R package (version 4.31.0). The promoters of the input genes were also downloaded from the resource of the HOMER software. Based on these promoter sequences, the genomic distribution of the most enriched motif hit with a weight score >6.0 was calculated using pattern matching method of the regulatory sequence analysis tools web server (http://rsat.sb-roscoff.fr/matrix-scan-quick_form.cgi)[102].

### Protein extraction and western blots

The proteins were extracted with radio-immunoprecipitation assay buffer containing protease and phosphatase inhibitors as previously described[98]. The western blots were carried out with antibodies against GFP (cat. no. 2956, CST, 1:1,000, RRID:AB_1196615), β-amyloid 1–16 (6E10) (cat. no. 803001, BioLegend, 1:1,000, RRID:AB_2564653), tubulin (cat. no. T5168, Sigma, 1:2,000, RRID:AB_477579), H3K27Ac (Ab4729, abcam, 1:1,000, RRID:AB_2118291), H3K9Ac (cat. no. 06-942, 1:1000, Merck, RRID:AB_310308), H3K4Ac (cat. no. Ab176799, abcam, 1:1,000, RRID:AB_2891335) and histone 3 (cat. no. 9715, CST, 1:2,000, RRID:AB_331563). The antibody for worm CPL-1 (1:5,000) was a kind gift from Professor Xiaochen Wang (SUSTech), as described and validated previously[19]. The horseradish peroxidase (HRP)-labelled anti-rabbit (cat. no. 7074; CST, 1:5,000, RRID:AB_2099233) and anti-mouse (cat. no. 7076; CST; 1:5,000, RRID:AB_330924) secondary antibodies were applied.

### Quantitative RT–PCR and ChIP–qPCR

The worms were collected and the total RNA was extracted with the same method mentioned above as for RNA-seq. A total of 1,000 ng of RNA was used for cDNA synthesis using the reverse transcription kit (cat. no. 205314, Qiagen). A qRT–PCR was performed with the LightCycler 480 SYBR Green I Master kit (cat. no. 04887352001, Roche). The primers for *act-3* and *pmp-3* were used as reference genes. For Figs. 2a,

3e,f, 4d, 5b,c and 7a, the alternative housekeeping genes *rps-26* and *rpl-35* (ref. 103), were also used to double check the data, and similar results were acquired (as currently showed). A ChIP–qPCR was carried out as previously described[49]. Briefly, *elt-2::TY1::EGFP::3xFLAG* worms were fixed with 1% formaldehyde solution for 15 min and quenched by glycine. After a total of 15 min sonication, immunoprecipitations were performed using the anti-FLAG M2 beads (cat. no. A2220, Sigma) in radio-immunoprecipitation assay buffer. All primers for qRT–PCR or ChIP–qPCR are as indicated in Supplementary Table 1.

### Thrashing/movement analysis

The worms were randomly picked from the culture plates, and ~15 worms were used for the thrashing assay of each condition. In brief, a single worm was placed in a drop of M9 buffer on a glass slide and allowed to acclimatize to the environment for 30 s. One movement of the worm that swung its head to the same side was considered as one thrash, and the frequency of thrashes was counted for 30 s as previously described[93].

### Positive olfactory associative memory assays

The vector control or *vha-6* RNAi-treated wild-type GRU101 and GRU102 animals were trained and tested for intermediate-term memory at day 4 of adulthood as previously described[67,104]. Briefly, synchronized day 4 adult hermaphrodites were washed from high growth medium RNAi plates with M9 buffer, allowed to settle by gravity and washed again with M9 buffer. After washing, the animals were starved for 1 h in M9 buffer. For intermediate-term memory training, the worms were then transferred to 10 cm NGM conditioning plates (seeded with OP50 *E. coli* bacteria and with 12 µl 10% 2-butanone (Acros Organics) in ethanol streaked across the lid with a pipette tip for 1 h. After conditioning, the trained population of worms were transferred to 10 cm NGM plates with fresh OP50 bacteria for a 60 min interval before testing worms for intermediate-term memory performance by chemotaxis to 10% butanone previously described chemotaxis assay conditions[105]. Chemotaxis indices were calculated as follows: (Number of worms$_{butanone}$ − Number of worms$_{ethanol}$)/(total number of worms). The performance index is the change in chemotaxis index following training relative to the naive (untrained) chemotaxis index, which was determined using a subpopulation of animals. The calculation for performance index is: chemotaxis index$_{trained}$ − chemotaxis index$_{naive}$.

### Statistics and reproducibility

No statistical methods were used to predetermine sample sizes, but our sample sizes are similar to those reported in previous publications[49,69,98]. Except for the random allocation of *C. elegans* to experimental groups/treatments after large-scale synchronization, the experiments were not randomized. All experiments, except for the RNA-seq, were repeated at least twice and similar results were acquired. The investigators were not blinded to allocation during experiments and outcome assessment, except for the RNA-seq analyses (Figs. 1h, 3g and 5f), where data analysis was performed in a blind manner until the group-to-group comparison steps were reached. No samples were excluded, except for the lifespan assays, where worms that escaped or had vulva explosions were censored. Graphpad Prism 8 (v8.3.1) or JMP 18 (v18.0.1) was used to conduct the statistical analyses. The data distribution was assumed to be normal but this was not formally tested. Two-tailed and unpaired Student's *t*-test was used to determine the differences between two independent groups. For more than two groups of comparisons, an analysis of variance (ANOVA) followed by Tukey's honest significant difference test was performed. A one-way ANOVA was used for comparisons between different groups, and two-way ANOVA was used for examining the effect of two independent variables on a dependent variable, for example, age and strain. For survival analyses, the Kaplan–Meier method was performed, and the significance was calculated using the log-rank (Mantel–Cox) method.

**Reporting summary**

Further information on research design is available in the Nature Portfolio Reporting Summary linked to this article.

## Data availability

Sequencing data that support the findings of this study have been deposited in the Gene Expression Omnibus under accession codes GSE196021, GSE196022 and GSE296199. For all RNA-seq analyses, the reads were mapped against the Caenorhabditis_elegans.WBcel235.89 genome downloaded from Ensembl. The gene expression data for the *elt-2* overexpression worms were retrieved from the Gene Expression Omnibus (GSE69263). Source data are provided with this paper. All other relevant data and materials are available either in the article and supplementary tables or from the corresponding authors upon reasonable request.

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

## Acknowledgements

We thank X. Wang (SUSTech) for the kind gift of anti-CPL-1 antibody and worm strains, M. Sarov (MPI-CBG) for the constructs for generating the new worm strains, and the Caenorhabditis Genetics Center, M.-Q. Dong (NIBS), S. Mitani (TWMU) and W. Zhou (ZJU) for providing the *C. elegans* strains. We thank all members of J. Auwerx and K. Schoonjans laboratories for helpful discussions. We thank the Imaging platform at the Shanghai Key Laboratory of Metabolic Remodeling and Health, Fudan University for technical assistance. This work was supported by the European Research Council (ERC-AdG-787702 to J.A.), the Swiss National Science Foundation (SNSF 31003A_179435 and Sinergia CRSII5_202302 to J.A.), National Research Foundation of Korea (NRF 2017K1A1A2013124 to J.A.), Human Frontier Science Program (LT000731/2018-L to T.Y.L.), National Key Research and Development Program of China (2024YFA1803003 to T.Y.L.), National Natural Science Foundation of China (2021hwyq47 and 32470816 to T.Y.L. and 82300708 to Y.S.), Fudan University and Cao'ejiang Basic Research (24FCA06 to T.Y.L.), United Mitochondrial Disease Foundation (PF-19-0232 to A.W.G.), Amsterdam UMC Postdoc Career Bridging Grant (A.W.G.), Horizon-MSCA-PF-EF-2022 (A.W.G.), AGEM Talent Development Grant (A.W.G.), China Postdoctoral Science Foundation (2023TQ0080 to Y.S.), Shanghai Magnolia Talent Plan Pujiang Project (2023PJD004 to Y.S.), China Scholarship Council (201906050019 to X.L.), Whitehall Foundation (R.N.A.), Glenn Foundation for Medical Research (R.N.A.), Longevity Impetus Grant from Norn Group (R.H.H.) and European Molecular Biology Organization (ALTF 1161-2021 and ALTF 111-2021 to Y.J.L and Q.W., respectively).

## Author contributions

Conceptualization: T.Y.L., A.W.G. and J.A. Methodology: T.Y.L., A.W.G., R.Y., Y.S., X.L. L.C., T.-j.Z., Q.W., T.L. and R.H.H. Investigation: T.Y.L., A.W.G., R.Y., Y.S., Y.L., Y.J.L., R.N.A., K.M., R.B.L., W.W., A.Z., W.L., A.L. and Q.W. Funding acquisition: T.Y.L., Y.S., R.N.A., R.H.H. and J.A. Supervision: T.Y.L. and J.A. Writing—original draft: T.Y.L., A.W.G. and J.A. Writing—review and editing: T.Y.L., A.W.G., R.Y., Y.S. and J.A.

## Funding

## Competing interests

The authors declare no competing interests.

## Additional information

**Extended data** is available for this paper at https://doi.org/10.1038/s41556-025-01693-y.

**Correspondence and requests for materials** should be addressed to Terytty Yang Li or Johan Auwerx.

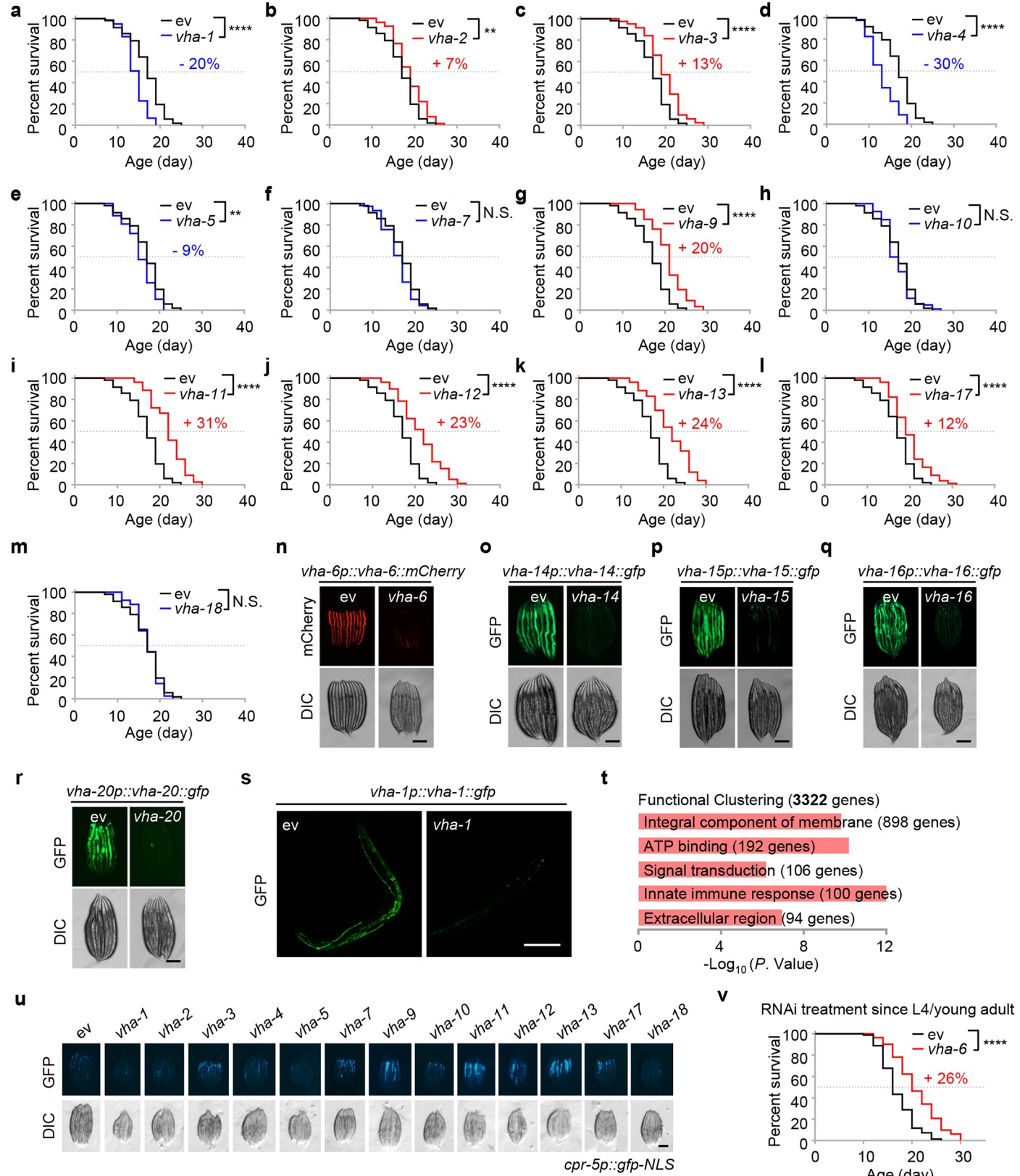

**Extended Data Fig. 1 | See next page for caption.**

**Extended Data Fig. 1 | Lifespan and gene expression changes in response to RNAi targeting different v-ATPase subunits in *C. elegans*. a-m**, Survival of worms treated with control (ev), or RNAi targeting different v-ATPase subunits. Each v-ATPase RNAi occupied 40%, except for *vha-11* and *vha-12* RNAi, which occupied 10%; control RNAi was used to supply to a final 100% of RNAi for all conditions. The percentages indicate the mean lifespan changes as compared to the control condition (****$P < 0.0001$; in (**b**), **$P = 0.0015$; in (**e**), **$P = 0.0056$; in (**f**), $P = 0.0580$ (N.S.); in (**h**), $P > 0.9999$ (N.S.); in (**m**), $P = 0.7280$ (N.S.)). **n-s**, Transgenic worm strains expressing mCherry tagged VHA-6 (**n**), and GFP-tagged VHA-14 (**o**), VHA-15 (**p**), VHA-16 (**q**), VHA-20 (**r**) and VHA-1 (**s**), were treated with the corresponding VHA RNAi and examined for fluorescence intensity. **t**, Functional clustering of the 3,322 differentially expressed genes (DEGs) that commonly up-regulated in in response to *vha-6*, *vha-16* and *vha-19* RNAi (25%). *P*. Value was derived from DAVID (one-sided Fisher's Exact test). **u**, GFP expression levels of *cpr-5p::gfp* worms treated with RNAi targeting different v-ATPase subunits. **v**, *vha-6* RNAi (100%) extended wild-type N2 worm lifespan by 26%, even when RNAi treatment started since the L4/young adult stage (****$P < 0.0001$). Scale bars, 0.3 mm. Statistical analysis was performed by log-rank test (N.S., not significant). Statistical data for lifespan can be found in Supplementary Table 1.

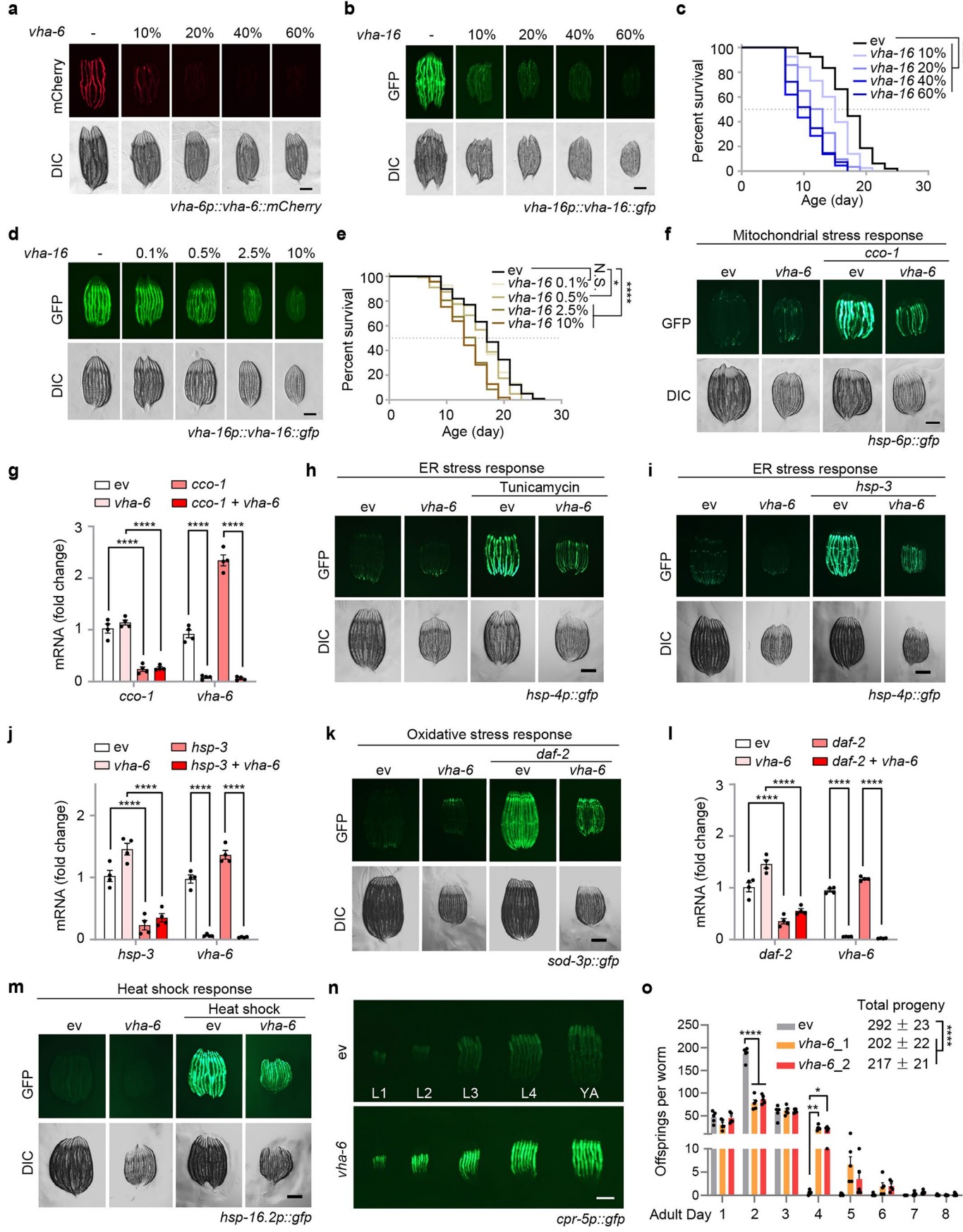

**Extended Data Fig. 2 | See next page for caption.**

**Extended Data Fig. 2 | Effect of *vha-6* and *vha-16* RNAi on gene expression, lifespan, stress responses and fertility. a**, mCherry-VHA-6 expression of worms treated with control or 10%-60% *vha-6* RNAi. **b,c**, GFP-VHA-16 expression (**b**) and survival (**c**) of worms treated with control or 10%-60% *vha-16* RNAi (****$P < 0.0001$). **d,e**, GFP-VHA-16 expression (**d**) and survival (**e**) of worms treated with control or 0.1%-10% *vha-16* RNAi (****$P < 0.0001$, *$P = 0.0127$ (ev VS *vha-16* 0.5%), $P = 0.3713$ (N.S., ev VS *vha-16* 0.1%)). **f**, RNAi of *vha-6* attenuates the mitochondrial stress response activation induced by *cco-1* RNAi in *hsp-6p::gfp* worms. RNAi targeting *cco-1* occupies 40%, *vha-6* RNAi occupies 20%. **g**, qRT-PCR analysis ($n = 4$ biologically independent samples) of worms as indicated in (**f**) (****$P < 0.0001$). **h,i**, RNAi of *vha-6* attenuates the ER stress response induced by tunicamycin (5 µg/mL) (**h**) or *hsp-3* RNAi (**i**) in *hsp-4p::gfp* worms. **j**, qRT-PCR analysis ($n = 4$ biologically independent samples) of worms as indicated in (**i**) (****$P < 0.0001$). **k**, RNAi of *vha-6* attenuates the oxidative stress response induced by *daf-2* RNAi in *sod-3p::gfp* worms. RNAi targeting *daf-2* occupies 40%,

*vha-6* RNAi occupies 20%. **l**, qRT-PCR analysis ($n = 4$ biologically independent samples) of worms as indicated in (**k**) (****$P < 0.0001$). **m**, RNAi of *vha-6* (20%) does not affect the heat shock response activation induced by heat shock (31 °C for 8 h). **n**, Different developmental stages of *vha-6* RNAi treated worms all displayed much higher *cpr-5p::gfp* induction as compared to that in worms treated with control RNAi, suggesting that the LySR response and body size can be decoupled. **o**, The rate of egg-laying ($n = 5$ plates with 3 worms/plate for each condition) at different days of adulthood in worms treated with control, *vha-6_1* or *vha-6_2* RNAi. RNAi (100%) treatment started since the L4/young adult stage. The total egg output per worm was also calculated (****$P < 0.0001$, **$P = 0.0018$ (Day 4, ev VS *vha-6_1*), *$P = 0.0277$ (Day 4, ev VS *vha-6_2*)). Scale bars, 0.3 mm. Error bars denote SEM. Statistical analysis was performed log-rank test (**c**, **e**) and ANOVA followed by Tukey post-hoc test (**g**, **j**, **l**, **o**) (N.S., not significant). Statistical data for lifespan can be found in Supplementary Table 1.

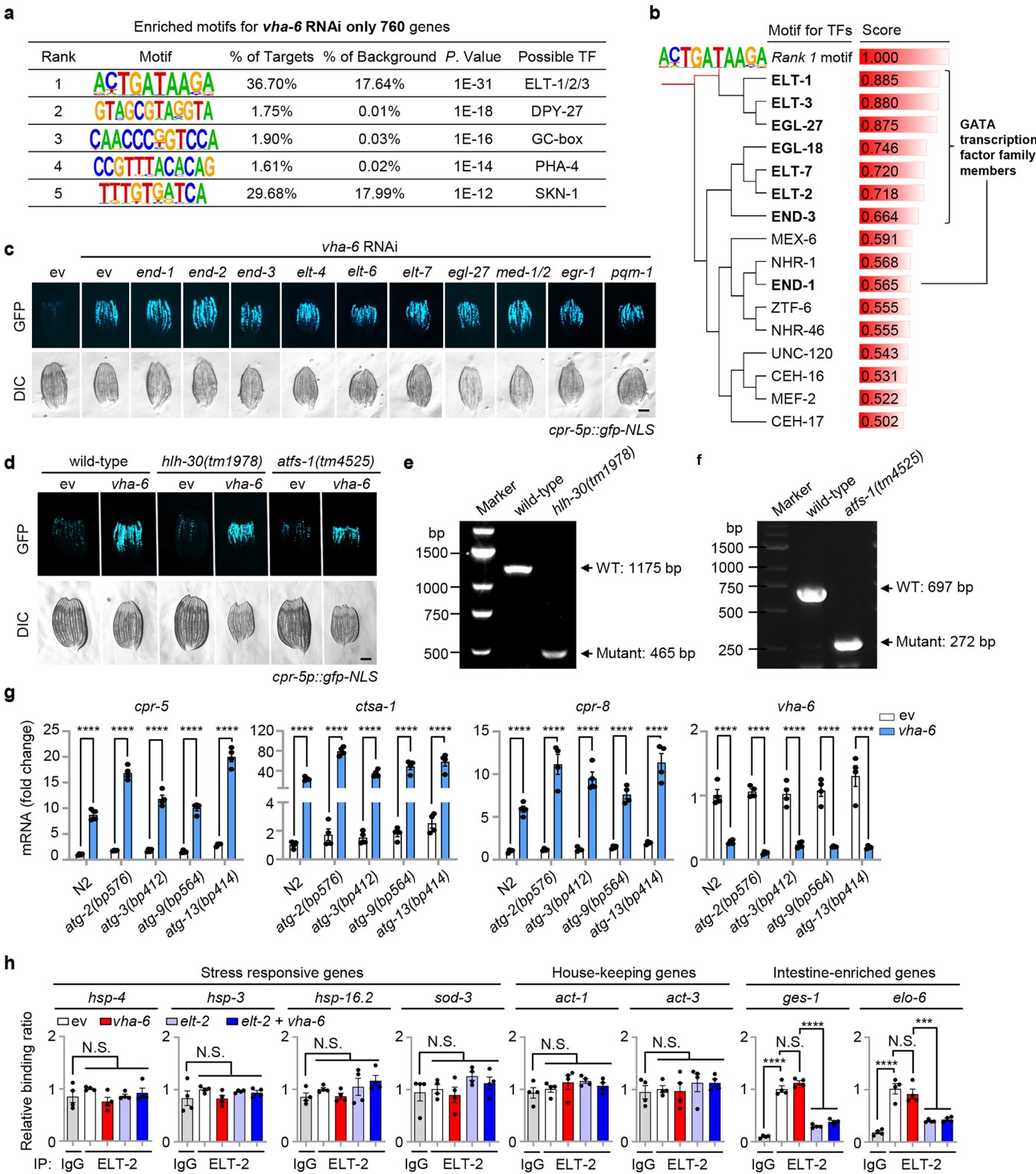

**Extended Data Fig. 3 | See next page for caption.**

**Extended Data Fig. 3 | Identification of determinates that regulate LySR activation. a**, The top five enriched motifs for promoters of the 760 up-regulated genes upon *vha-6* RNAi, but not upon *vha-16* or *vha-19* RNAi, by using Hypergeometric optimization of motif enrichment (HOMER) and ranked based on the *P* values. *P*. Value was derived from HOMER (one-sided hypergeometric test). **b**, The similarity of the Rank 1 motif as found in (**a**) to the putative binding motifs of all known transcription factors in *C. elegans*. All transcription factors with a similarity score above 0.500 were shown. The GATA transcription factor members were highlighted in bold. **c**, RNAi of other GATA transcription factor members, or *pqm-1*, did not affect the GFP expression of *cpr-5p::gfp* worms induced by *vha-6* RNAi. RNAi targeting *vha-6* occupied 40%, RNAi targeting GATA transcription factor members, or *pqm-1*, occupied 60%. **d**, GFP induction induced by *vha-6* RNAi in *cpr-5p::gfp* worms is not blocked by the knockout of *hlh-30* or *atfs-1*. **e**,**f**, Genotyping results for the wild-type, *hlh-30(tm1978)* (**e**) and *atfs-1(tm4525)* (**f**) strains in a *cpr-5p::gfp-NLS* background as indicated in (**d**). **g**, The indicated autophagy defective mutants have a normal induction of representative lysosomal proteases in response to *vha-6* RNAi. qRT-PCR analysis (*n* = 4 biologically independent samples) of wild-type (N2) or autophagy defect mutants treated with control or *vha-6* RNAi (****$P$ < 0.0001). **h**, ChIP-qPCR

(*n* = 4 biologically independent samples) analyses of the indicated genes in *elt-2p::elt-2::gfp-flag* worms treated with control or *vha-6* RNAi, and/or *elt-2* RNAi. ChIP was performed by using IgG control or anti-Flag M2 beads (****$P$ < 0.0001; for *hsp-4*, $P$ = 0.6641 (N.S., IgG VS ev), $P$ = 0.8960 (N.S., IgG VS *vha-6*), $P$ > 0.9999 (N.S., IgG VS *elt-2*), $P$ = 0.9688 (N.S., IgG VS *elt-2* + *vha-6*); for *hsp-3*, $P$ = 0.5232 (N.S., IgG VS ev), $P$ > 0.9999 (N.S., IgG VS *vha-6*), $P$ = 0.7291 (N.S., IgG VS *elt-2*), $P$ = 0.8456 (N.S., IgG VS *elt-2* + *vha-6*); for *hsp-16.2*, $P$ = 0.8197 (N.S., IgG VS ev), $P$ > 0.9999 (N.S., IgG VS *vha-6*), $P$ = 0.6324 (N.S., IgG VS *elt-2*), $P$ = 0.2148 (N.S., IgG VS *elt-2* + *vha-6*); for *sod-3*, $P$ = 0.9950 (N.S., IgG VS ev), $P$ = 0.9985 (N.S., IgG VS *vha-6*), $P$ = 0.4144 (N.S., IgG VS *elt-2*), $P$ = 0.8408 (N.S., IgG VS *elt-2* + *vha-6*); for *act-1*, $P$ = 0.9714 (N.S., IgG VS ev), $P$ = 0.5075 (N.S., IgG VS *vha-6*), $P$ = 0.3890 (N.S., IgG VS *elt-2*), $P$ = 0.8057 (N.S., IgG VS *elt-2* + *vha-6*); for *act-3*, $P$ = 0.9976 (N.S., IgG VS ev), $P$ > 0.9999 (N.S., IgG VS *vha-6*), $P$ = 0.8545 (N.S., IgG VS *elt-2*), $P$ = 0.8580 (N.S., IgG VS *elt-2* + *vha-6*); for *ges-1*, $P$ = 0.1858 (N.S., ev VS *vha-6*); for *elo-6*, $P$ = 0.7980 (N.S., ev VS *vha-6*), ***$P$ = 0.0003 (*vha-6* VS *elt-2*), ***$P$ = 0.0005 (*vha-6* VS *elt-2* + *vha-6*)). Scale bar, 0.3 mm. Error bars denote SEM. Statistical analysis was performed by two-tailed unpaired Student's *t*-test (**g**) or ANOVA followed by Tukey post-hoc test (**h**) (N.S., not significant).

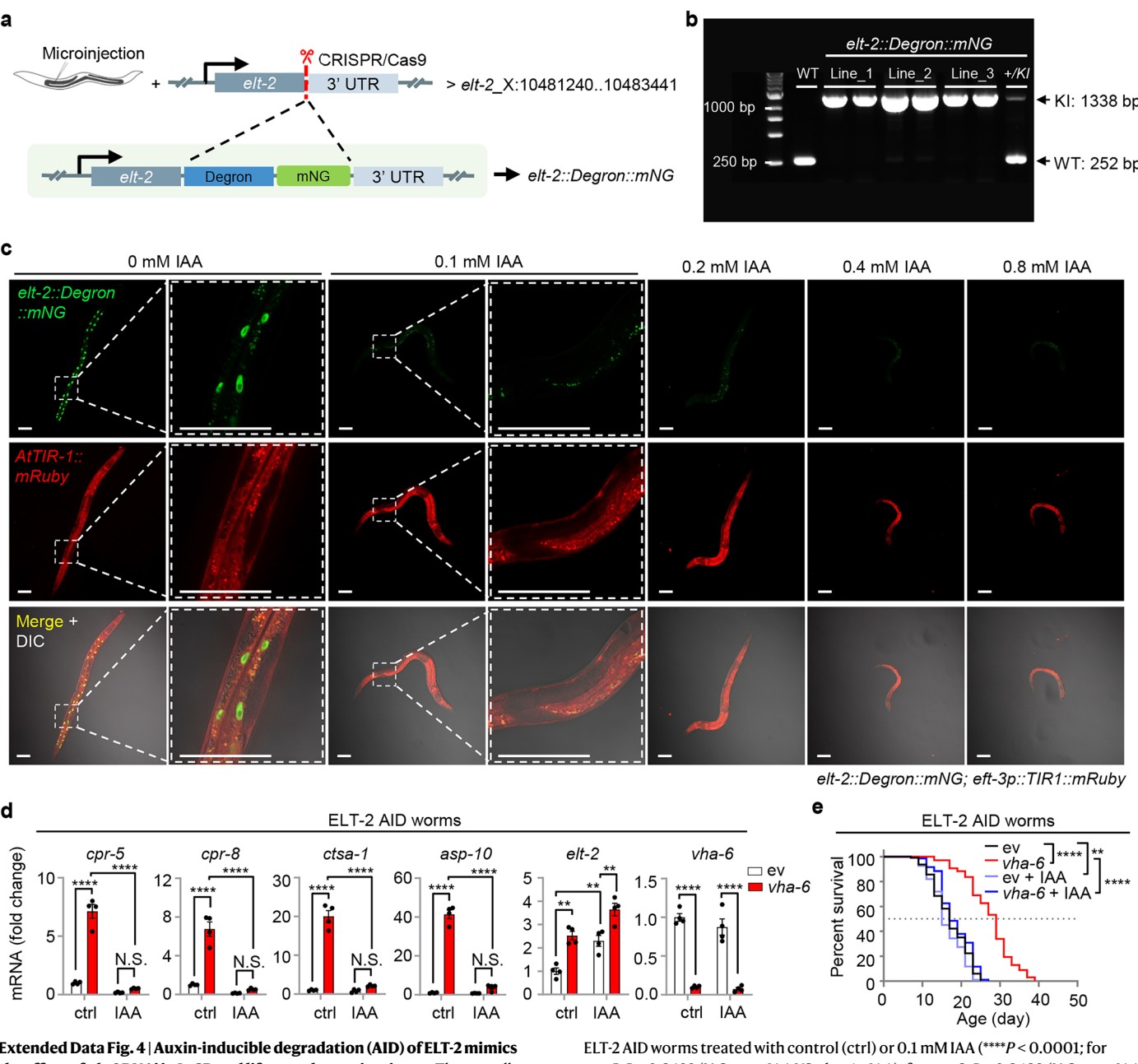

**Extended Data Fig. 4 | Auxin-inducible degradation (AID) of ELT-2 mimics the effect of *elt-2* RNAi in LySR and lifespan determination. a**, The overall design for CRISPR/Cas9-mediated *elt-2::Degron::mNeonGreen* knock-in with the endogenous tagging of a Degron-mNeonGreen tag immediately after the last amino acid codon of the *elt-2* gene at chromosome X. **b**, The genotyping results of the wild-type, homozygous and heterozygous *elt-2::Degron::mNeonGreen* knock-in strains. **c**, The natural auxin indole-3-acetic acid (IAA) treatment at 0.1-0.8 mM remarkably decreased the Degron-mNG-ELT-2 expression in the *elt-2::Degron::mNeonGreen; eft-3p::TIR1::mRuby* (ELT-2 AID) worms. Scale bars, 0.1 mm. **d**, qRT-PCR analysis (*n* = 4 biologically independent samples) of the

ELT-2 AID worms treated with control (ctrl) or 0.1 mM IAA (****$P$ < 0.0001; for *cpr-5*, $P$ = 0.8422 (N.S., ev + IAA VS *vha-6* + IAA); for *cpr-8*, $P$ = 0.8689 (N.S., ev + IAA VS *vha-6* + IAA); for *ctsa-1*, $P$ = 0.6185 (N.S., ev + IAA VS *vha-6* + IAA); for *asp-10*, $P$ = 0.3410 (N.S., ev + IAA VS *vha-6* + IAA); for *elt-2*, **$P$ = 0.0012 (ev VS *vha-6*), **$P$ = 0.0043 (ev VS ev + IAA), **$P$ = 0.0033 (ev + IAA VS *vha-6* + IAA)). **e**, Survival of ELT-2 AID worms treated with control or *vha-6* (20%) RNAi and/or 0.1 mM IAA (****$P$ < 0.0001, **$P$ = 0.0022 (ev VS ev + IAA)). Error bars denote SEM. Statistical analysis was performed by ANOVA followed by Tukey post-hoc test (**d**) or log-rank test (**e**) (N.S., not significant). Statistical data for lifespan can be found in Supplementary Table 1.

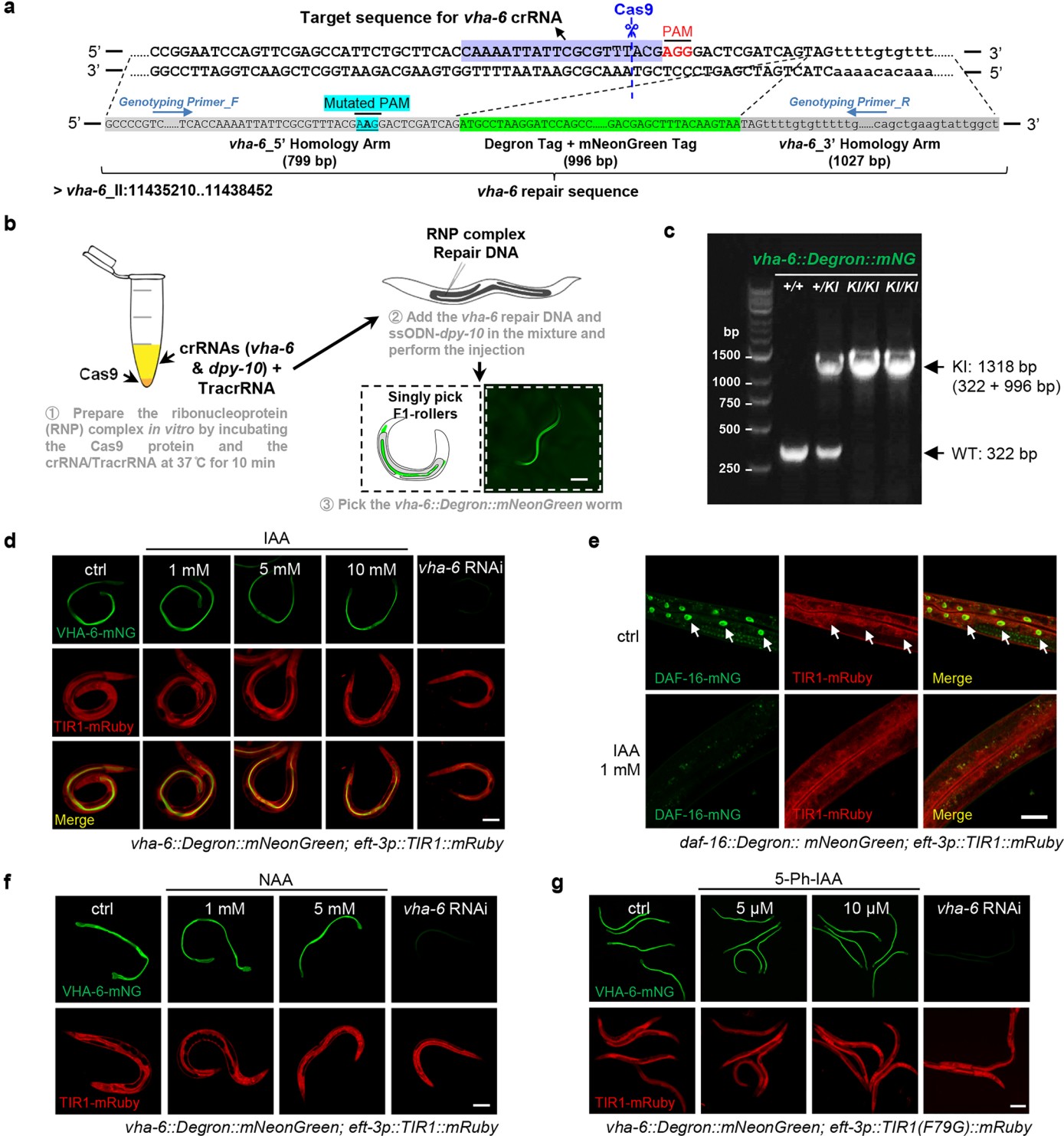

**Extended Data Fig. 5 | Attempts to degrade VHA-6 with the AID/AID2 system.**
**a**, The overall design for CRISPR/Cas9-mediated *Degron::mNeonGreen* knock-in
with the endogenous tagging of a degron-mNeonGreen tag immediately
after the last amino acid codon of *vha-6* gene at chromosome II. **b,c**, The
experimental flow (**b**) and genotyping result of the wild-type (+/+), heterozygous
(+/*KI*) and homozygous (*KI/KI*) *vha-6::Degron::mNeonGreen* strains (**c**).
**d**, The natural auxin indole-3-acetic acid (IAA) treatment at 1, 5 or 10 mM since
egg stage barely reduced the expression level of Degron-mNeoGreen tagged
VHA-6 in *vha-6::Degron::mNeonGreen; eft-3p::TIR1::mRuby* worms. *vha-6* RNAi

(20%) was used as a positive control. **e**, IAA at 1 mM remarkably decreased the
Degron-mNeoGreen-DAF-16 expression in *daf-16::Degron::mNeonGreen; eft-
3p::TIR1::mRuby* worms. **f**, 1-naphthaleneacetic acid (NAA) treatment at 1-10 mM
since egg stage barely reduced the expression level of degron-mNeoGreen
tagged VHA-6 in *vha-6::Degron::mNeonGreen; eft-3p::TIR1::mRuby* worms. **g**, The
IAA derivative phenyl-indole-3-acetic acid (5-Ph-IAA) treatment at 5 or 10 μM
since egg stage barely reduced the expression level of degron-mNeoGreen
tagged VHA-6 in *vha-6::Degron::mNeonGreen; eft-3p::TIR1(F79G)::mRuby* worms.
Scale bars, 0.2 mm.

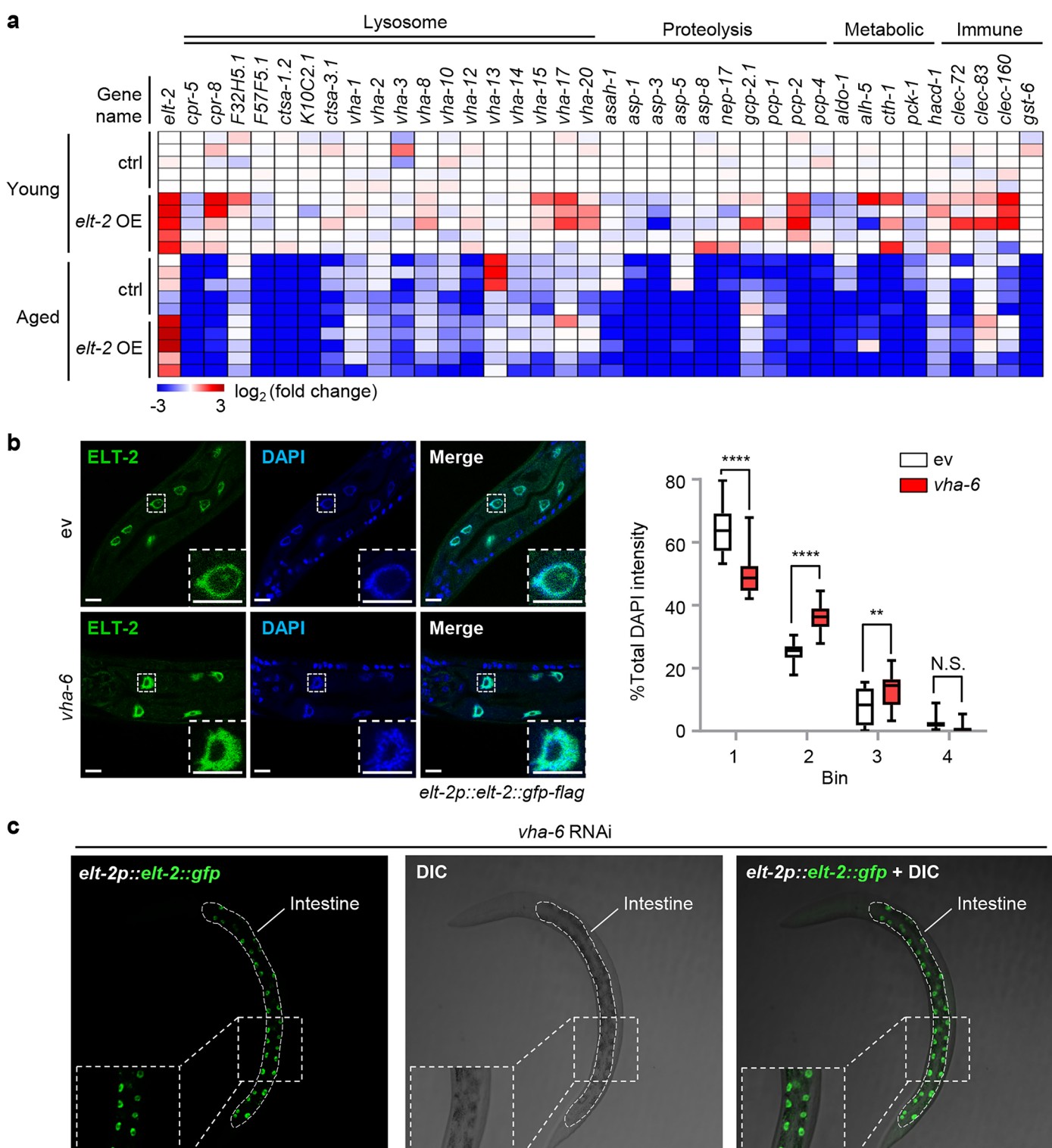

**Extended Data Fig. 6 | The impact of *elt-2* overexpression and analysis of ELT-2 localization. a**, Heat-map of the relative expression of representative LySR genes in control or *elt-2* overexpression (OE) worms at young (L4) or aged (Day 13) stage based on an extant RNA-seq dataset (GSE69263). The color represents gene expression differences in log₂(fold change, FC). **b**,**c**, Representative images of GFP-tagged ELT-2 and DAPI staining in *elt-2p::elt-2::gfp-flag* worms treated with control or *vha-6* RNAi. Scale bars, 10 μm for (**b**) and 0.1 mm for (**c**). The quantification panel indicates the distribution of DAPI or

ELT-2 signal in intestinal nuclei in *elt-2p::elt-2::gfp-flag* worms treated with control (ev) or *vha-6* RNAi. Image voxels were ranked by DAPI intensity within each nucleus and divided into 4 equal-volume bins. Percentage of total DAPI intensity in each of the bins was measured (*n* = 30 nuclei for each condition) (****$P$ < 0.0001, **$P$ = 0.0037 (Bin 3, ev VS *vha-6*), $P$ = 0.8625 (N.S., Bin 4, ev VS *vha-6*)). Boxes and lines represent interquartile range (IQR) and median, respectively; whiskers represent min to max. Statistical analysis was performed by ANOVA followed by Tukey post-hoc test (N.S., not significant).

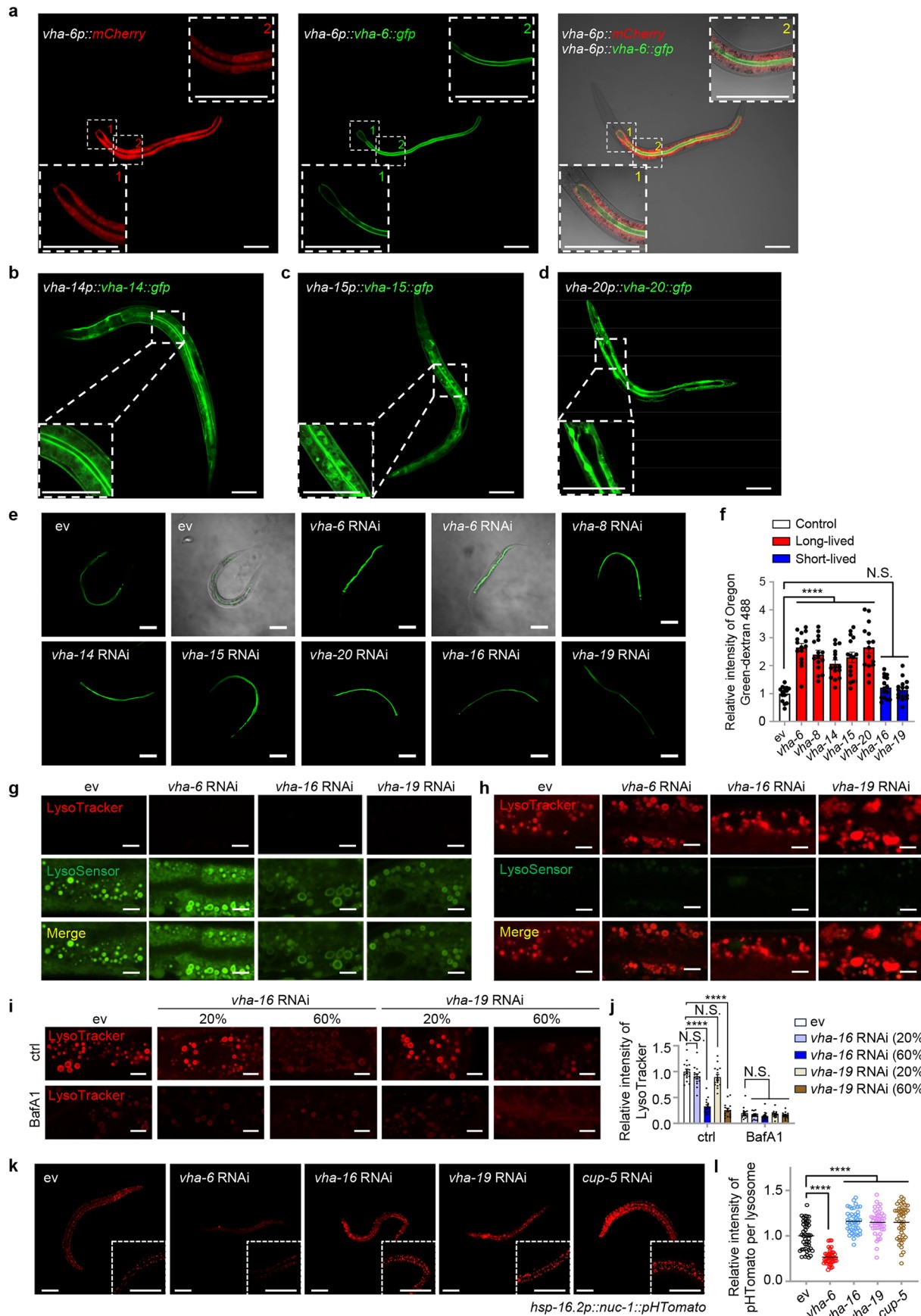

**Extended Data Fig. 7 | See next page for caption.**

**Extended Data Fig. 7 | Expression pattern of VHA-6 and VHA-16, and effect of different v-ATPase RNAi on intestinal pH. a-d**, Transgenic worm strains expressing *vha-6* promoter-driven mCherry and GFP-tagged VHA-6 (**a**), GFP-tagged VHA-14 (**b**), VHA-15 (**c**) and VHA-20 (**d**). Scale bars, 0.1 mm. **e,f**, The effect of different v-ATPase RNAi on intestinal lumen pH, as revealed by Oregon Green-dextran 488 staining (**e**). Scale bars, 0.1 mm. Worms treated with *vha-6*, *vha-8*, *vha-14*, *vha-15* and *vha-20* RNAi show brighter green fluorescence, indicating higher intestinal lumen pH. The relative intensity of Oregon Green-dextran 488 ($n = 15$ individual worms for each condition) is shown in (**f**). Each VHA RNAi occupies 20%, control RNAi was used to supply to a final 100% of RNAi for all conditions (****$P < 0.0001$, $P = 0.9637$ (N.S., ev VS *vha-16*), $P = 0.9980$ (N.S., ev VS *vha-19*)). **g,h**, The single staining of worms by LysoSensor Green (**g**) and LysoTracker Red (**h**) validates that the fluorescent signal from each channel is specific for the corresponding dye applied. Scale bars, 10 μm. **i,j**, LysoTracker Red staining of worms treated with control (ev), *vha-16* or *vha-19* RNAi. Each VHA RNAi occupies 20% or 60%, control RNAi was used to supply to a final 100% of RNAi for all conditions (**i**). When indicated, worms were pretreated with v-ATPase inhibitor Bafilomycin A1 (BafA1) (0.2 mM, 2 h) prior to the addition of LysoTracker Red. Scale bars, 10 μm. The relative average intensity of LysoTracker for each animal is shown in (**j**) (15 animals were scored for each condition) (****$P < 0.0001$, $P = 0.6753$ (N.S., in ctrl, ev VS *vha-16* 20%), $P = 0.4511$ (N.S., in ctrl, ev VS *vha-19* 20%), $P = 0.9963$ (N.S., in BafA1, ev VS *vha-16* 20%), $P = 0.9726$ (N.S., in BafA1, ev VS *vha-16* 60%), $P = 0.9999$ (N.S., in BafA1, ev VS *vha-19* 20%), $P = 0.9978$ (N.S., in BafA1, ev VS *vha-19* 60%)). **k,l**, Confocal fluorescence images indicating the acidification states of the lysosomes revealed by the pHTomato-tagged NUC-1 controlled by the heat-shock *hsp-16.2* promoter (**k**). The relative intensity of pHTomato per lysosome is shown in (**l**) ($n = 45$ independent worm images for each condition) (****$P < 0.0001$). Scale bars, 0.1 mm. Error bars denote SEM. Statistical analysis was performed by ANOVA followed by Tukey post-hoc test (N.S., not significant).

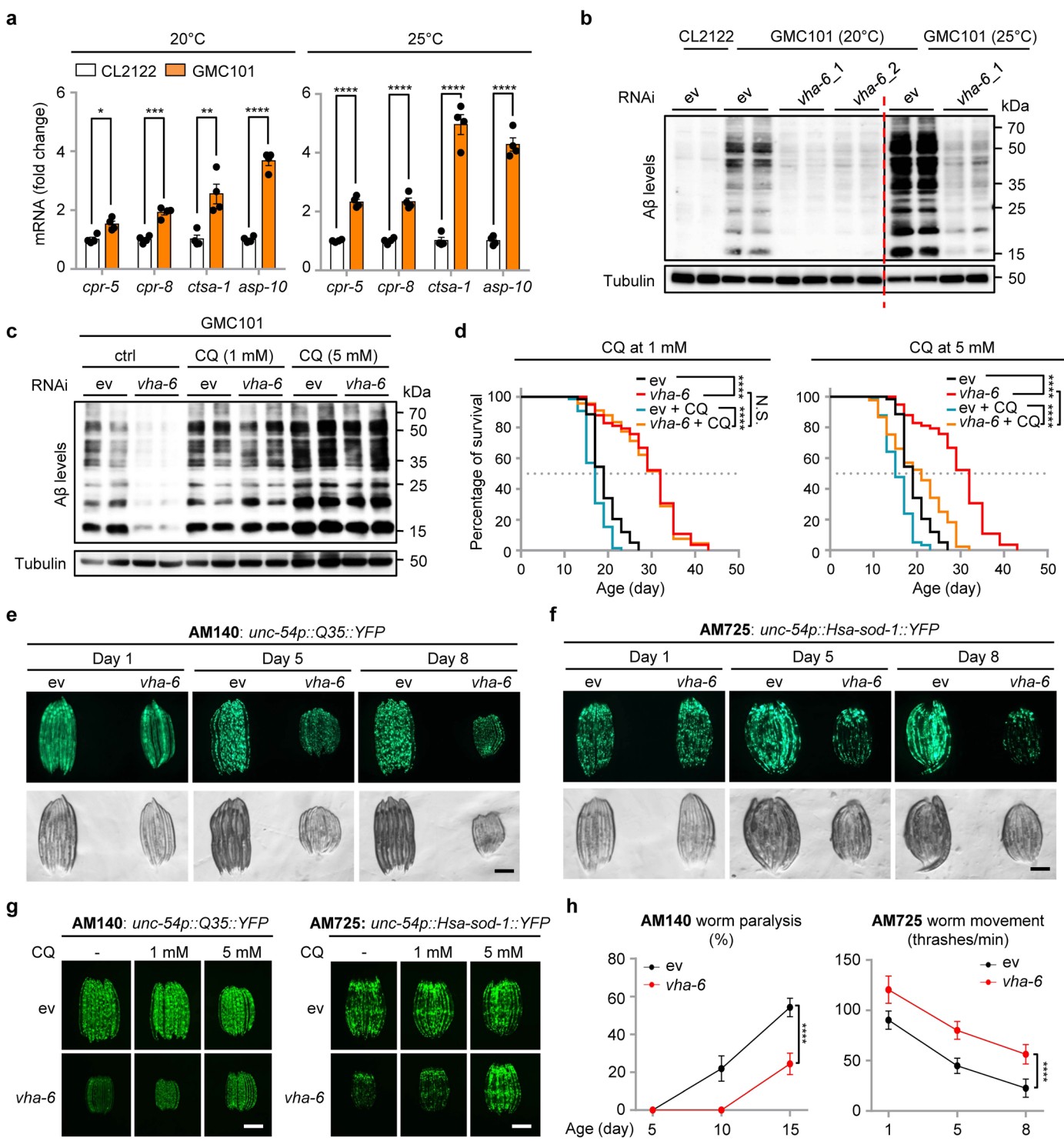

**Extended Data Fig. 8 | LySR activation enhances aggregation clearance and extends healthspan. a**, qRT-PCR analysis (*n* = 4 biologically independent samples) of CL2122 and GMC101 worms cultured at two different temperatures since Larval 4 (L4) stage (****P < 0.0001; for *cpr-5* at 20 °C, *P = 0.0174; for *cpr-8* at 20 °C, ***P = 0.0001; for *ctsa-1* at 20 °C, **P = 0.0054). **b**, Western blots of CL2122 or GMC101 worms treated with control (ev), *vha-6_1* or *vha-6_2* RNAi, cultured at two different temperatures since L4 stage. **c**, Chloroquine (CQ) treatment led to strong accumulation of amyloid-β aggregates and almost completely blunted *vha-6* RNAi induced amyloid-β aggregation clearance. Western blots of GMC101 worms treated with control or *vha-6* RNAi and treated with or without CQ at a final concentration of 1 mM or 5 mM. **d**, Survival of worms treated with control or *vha-6* RNAi, and treated with or without CQ at 1 mM (left) or 5 mM (right)

(****P < 0.0001, P = 0.8085 (N.S., *vha-6* VS *vha-6* + 1 mM CQ)). **e,f**, RNAi of *vha-6* (20%) reduces disease-causing protein aggregate formation in *unc-54p::Q35::YFP* (Huntington's disease polyQ model) (**e**) and *unc-54p::Hsa-sod-1::YFP* (ALS model) (**f**) worms. **g**, Worms expressing either *unc-54p::Q35::YFP* (polyQ model) (left) or *unc-54p::Hsa-sod-1::YFP* (ALS model) (right) were treated with control or *vha-6* (20%) RNAi and/or 1-5 mM CQ, analyzed at Day 5 of adulthood. **h**, Paralysis (*n* = 10 independent worm plates for each condition) (left), or movement (*n* = 12 individual worms for each condition) (right) of worms as indicated in (**e**) and (**f**), respectively (****P < 0.0001). Scale bars, 0.3 mm. Error bars denote SEM. Statistical analysis was performed by two-tailed unpaired Student's *t*-test (**a**), log-rank test (**d**), or ANOVA followed by Tukey post-hoc test (**h**) (N.S., not significant). Statistical data for lifespan can be found in Supplementary Table 1.

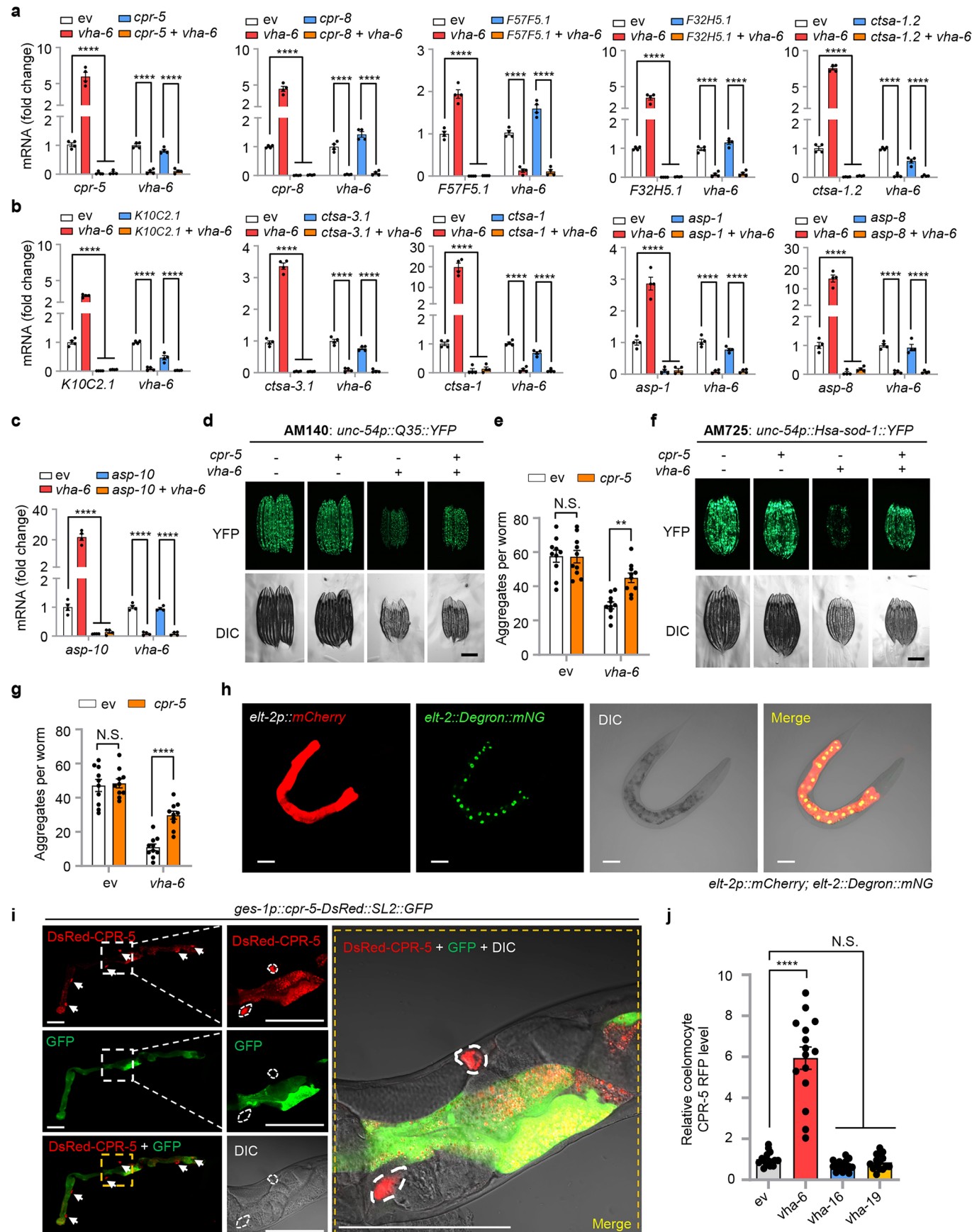

**Extended Data Fig. 9 | See next page for caption.**

**Extended Data Fig. 9 | Role of lysosomal proteases in LySR activation and *vha-6* RNAi-induced beneficial effects. a-c**, Knockdown validations with qRT-PCR analysis (*n* = 4 biologically independent samples) of worms treated with RNAi as indicated (****$P$ < 0.0001). **d-g**, Worms expressing either *unc-54p::Q35::YFP* (polyQ model) (**d** and **e**) or *unc-54p::Hsa-sod-1::YFP* (ALS model) (**f** and **g**) were treated with control, *cpr-5* and/or *vha-6* RNAi, and analyzed at Day 5 of adulthood (*n* = 10 individual worms for each condition). RNAi targeting *cpr-5* occupied 80%, *vha-6* RNAi occupied 20%. Control RNAi was used to supply to a final 100% of RNAi for all conditions (****$P$ < 0.0001; in (**e**), $P$ = 0.9999 (N.S., ev VS *cpr-5*), **$P$ = 0.0037 (*vha-6* VS *vha-6* + *cpr-5*); in (**g**), $P$ = 0.9999 (N.S., ev VS *cpr-5*)). Scale bars, 0.3 mm. **h**, Transgenic worm strains expressing *elt-2* promoter-driven mCherry and Degron-mNG-tagged ELT-2. Scale bars, 0.1 mm. **i**, The expression pattern of intestine-specific *ges-1* promoter-driven DsRed-CPR-5 and GFP in *ges-1p::cpr-5::DsRed;SL-2::GFP* worms. The CPR-5-DsRed is localized not only in the intestine but also in the six coelomocytes (white arrows and in dashed circles). The gene expression of *cpr-5* is indicated by polycistronic GFP, while the CPR-5 protein is visualized by its RFP fusion. Scale bar, 100 μm. **j**, The level of secreted CPR-5-RFP fusion is increased by *vha-6* RNAi, but not by *vha-16* or *vha-19* RNAi (*n* = 15 individual worms for each condition) (****$P$ < 0.0001, $P$ = 0.8769 (N.S., ev VS *vha-16*), $P$ = 0. 9843 (N.S., ev VS *vha-19*)). Error bars denote SEM. Statistical analysis was performed by ANOVA followed by Tukey post-hoc test (N.S., not significant).

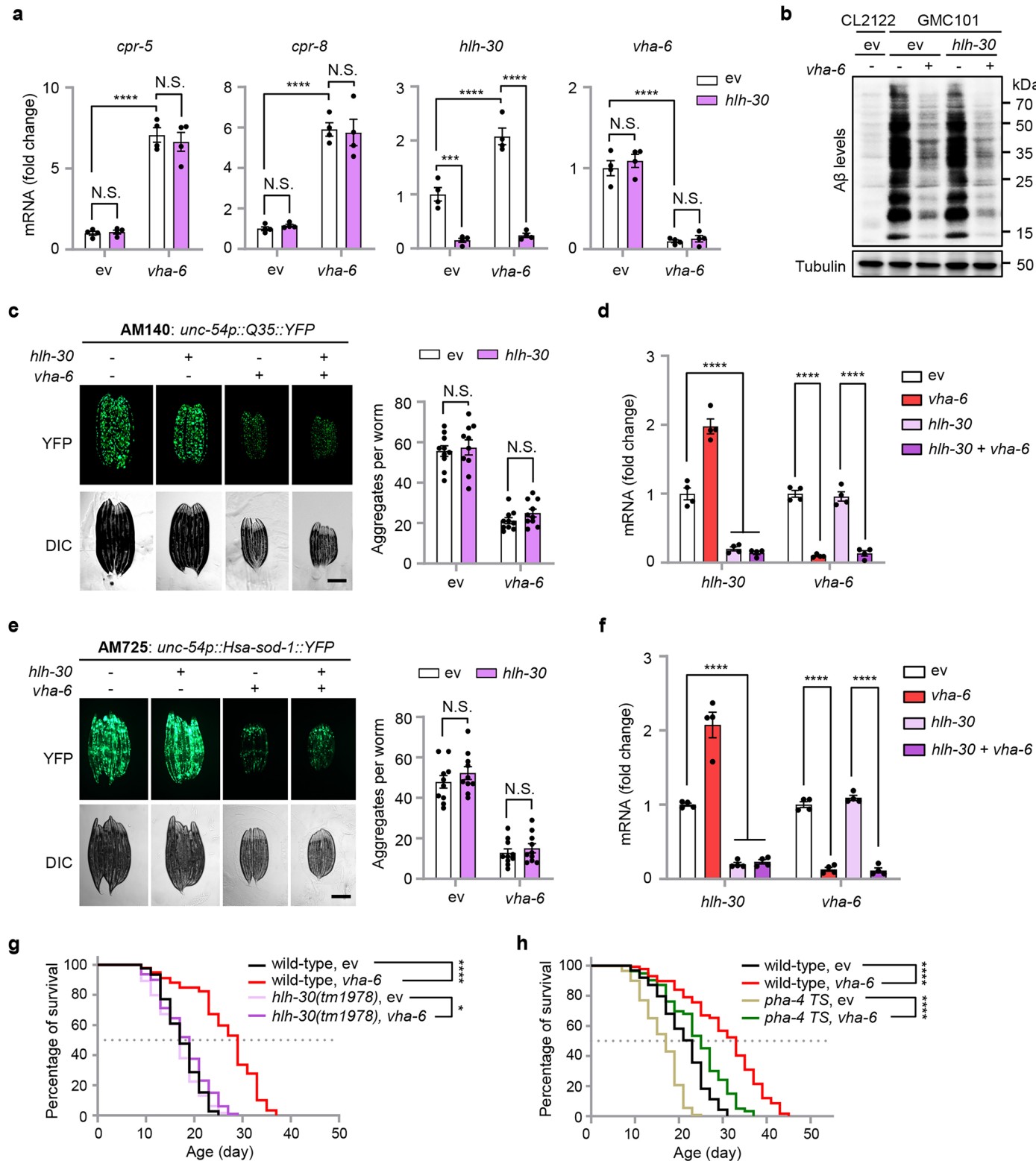

**Extended Data Fig. 10 | See next page for caption.**

**Extended Data Fig. 10 | Impacts of *hlh-30* and *pha-4* inactivation in *vha-6* RNAi-induced LySR activation and lifespan extension. a**, qRT-PCR analysis (*n* = 4 biologically independent samples) of GMC101 worms treated with control (ev), *hlh-30* and/or *vha-6* RNAi. RNAi targeting *hlh-30* occupied 80%, *vha-6* RNAi occupied 20%. Control RNAi was used to supply to a final 100% of RNAi for all conditions (****$P$ < 0.0001; for *cpr-5*, $P$ = 0.9991 (N.S., ev VS *hlh-30*), $P$ = 0.8633 (N.S., *vha-6* VS *vha-6* + *hlh-30*); for *cpr-8*, $P$ = 0.9912 (N.S., ev VS *hlh-30*), $P$ = 0.9908 (N.S., *vha-6* VS *vha-6* + *hlh-30*); for *hlh-30*, ***$P$ = 0.0004 (ev VS *hlh-30*); for *vha-6*, $P$ = 0.7724 (N.S., ev VS *hlh-30*), $P$ = 0.9869 (N.S., *vha-6* VS *vha-6* + *hlh-30*)). **b**, Western blots of CL2122 or GMC101 worms treated with control, *hlh-30* and/or *vha-6* RNAi. **c**, Worms expressing *unc-54p::Q35::YFP* (polyQ model) were treated with control, *hlh-30* and/or *vha-6* RNAi, and analyzed at Day 5 of adulthood (*n* = 10 individual worms for each condition) ($P$ = 0.9604 (N.S., ev VS *hlh-30*), $P$ = 0.7285 (N.S., *vha-6* VS *vha-6* + *hlh-30*)). Scale bars, 0.3 mm. **d**, qRT-PCR analysis (*n* = 4 biologically independent samples) of *unc-54p::Q35::YFP* worms treated with RNAi as indicated (****$P$ < 0.0001). **e**, Worms expressing *unc-54p::Hsa-sod-1::YFP*

(ALS model) were treated with control, *hlh-30* and/or *vha-6* RNAi, and analyzed at Day 5 of adulthood (*n* = 10 individual worms for each condition) ($P$ = 0.6603 (N.S., ev VS *hlh-30*), $P$ = 0.9307 (N.S., *vha-6* VS *vha-6* + *hlh-30*)). Scale bars, 0.3 mm. **f**, qRT-PCR analysis (*n* = 4 biologically independent samples) of *unc-54p::Hsa-sod-1::YFP* worms treated with RNAi as indicated (****$P$ < 0.0001). **g**, Survival of wild-type and *hlh-30(tm1978)* worms treated with control or *vha-6* (20%) RNAi (****$P$ < 0.0001, *$P$ = 0.0109 (*tm1978* ev VS *vha-6*)). **h**, Survival of wild-type and *pha-4(zu225);smg-1(cc546ts)* temprature sensivie (TS) worms treated with control or *vha-6* (20%) RNAi. Lifespan was conducted at 15°C after the first day of adulthood to inactivate *pha-4*. Note that similar impacts of *vha-6* RNAi on lifespan were found in wild-type and *pha-4* TS mutant worms, as compared to their own control RNAi conditions. Wild-type worms live longer than normal as the experiments were conducted at 15°C instead of 20°C (****$P$ < 0.0001). Error bars denote SEM. Statistical analysis was performed by ANOVA followed by Tukey post-hoc test (**a**, **c**-**f**), or log-rank test (**g**, **h**) (N.S., not significant). Statistical data for lifespan can be found in Supplementary Table 1.

# Reporting Summary

## Statistics

For all statistical analyses, confirm that the following items are present in the figure legend, table legend, main text, or Methods section.

| n/a | Confirmed | |
|---|---|---|
| ☐ | ☒ | The exact sample size (*n*) for each experimental group/condition, given as a discrete number and unit of measurement |
| ☐ | ☒ | A statement on whether measurements were taken from distinct samples or whether the same sample was measured repeatedly |
| ☐ | ☒ | The statistical test(s) used AND whether they are one- or two-sided *Only common tests should be described solely by name; describe more complex techniques in the Methods section.* |
| ☐ | ☒ | A description of all covariates tested |
| ☐ | ☒ | A description of any assumptions or corrections, such as tests of normality and adjustment for multiple comparisons |
| ☐ | ☒ | A full description of the statistical parameters including central tendency (e.g. means) or other basic estimates (e.g. regression coefficient) AND variation (e.g. standard deviation) or associated estimates of uncertainty (e.g. confidence intervals) |
| ☐ | ☒ | For null hypothesis testing, the test statistic (e.g. *F*, *t*, *r*) with confidence intervals, effect sizes, degrees of freedom and *P* value noted *Give P values as exact values whenever suitable.* |
| ☒ | ☐ | For Bayesian analysis, information on the choice of priors and Markov chain Monte Carlo settings |
| ☒ | ☐ | For hierarchical and complex designs, identification of the appropriate level for tests and full reporting of outcomes |
| ☐ | ☒ | Estimates of effect sizes (e.g. Cohen's *d*, Pearson's *r*), indicating how they were calculated |

*Our web collection on statistics for biologists contains articles on many of the points above.*

## Software and code

Policy information about availability of computer code

| Data collection | Microscopy pictures were acquired with Zeiss LSM 700 Upright or ZEISS LSM 980 with Airyscan 2 confocal microscope (Carl Zeiss AG); Victor X4 plate reader (Perkin Elmer) was used for all the assays requiring absorbance, luminescence or fluorescence quantifications; the qPCR reactions were performed using the Light-Cycler system (Roche Applied Science). RNA-seq analysis was performed using the R (version 3.6.3), FastQC (version 0.11.9) was used to verify the quality of the sequence data. Sequenced reads were mapped using STAR aligner (version 2.6.0a). Reads were counted using htseq-count (version 0.10.0). Differential expression of genes was calculated by using Limma-Voom method. |
|---|---|
| Data analysis | GraphPad Prism 8 for Mac OS X (GraphPad Software, Inc.; v8.3.1), JMP (https://www.jmp.com/en_ch/home.html; v18.0.1), Fiji (http://imagej.nih.gov/ij; version 1.47b). Heat-maps were generated by using Morpheus (https://software.broadinstitute.org/morpheus). Motif enrichment analysis was performed with HOMER (v4.11). Functional clustering in this study was performed by using the DAVID (Database for Annotation, Visualization and Integrated Discovery) database (https://david.ncifcrf.gov/home.jsp; v6.8). |

For manuscripts utilizing custom algorithms or software that are central to the research but not yet described in published literature, software must be made available to editors and reviewers. We strongly encourage code deposition in a community repository (e.g. GitHub). See the Nature Portfolio guidelines for submitting code & software for further information.

## Data

Policy information about availability of data

All manuscripts must include a data availability statement. This statement should provide the following information, where applicable:
- Accession codes, unique identifiers, or web links for publicly available datasets
- A description of any restrictions on data availability
- For clinical datasets or third party data, please ensure that the statement adheres to our policy

Sequencing data that support the findings of this study have been deposited in the Gene Expression Omnibus under accession codes: GSE196021, GSE196022 and GSE296199. For all RNA-seq analyses, reads were mapped against the Caenorhabditis_elegans.WBcel235.89 genome downloaded from Ensembl. Gene expression data for the elt-2 OE worms were retrieved from the Gene Expression Omnibus (GSE69263). Uncropped images for immunoblots and statistical data are available as source data. All other relevant data and materials are available either in the article and Supplementary Tables or from the corresponding authors upon reasonable request.

# Field-specific reporting

Please select the one below that is the best fit for your research. If you are not sure, read the appropriate sections before making your selection.

☒ Life sciences ☐ Behavioural & social sciences ☐ Ecological, evolutionary & environmental sciences

For a reference copy of the document with all sections, see nature.com/documents/nr-reporting-summary-flat.pdf

# Life sciences study design

All studies must disclose on these points even when the disclosure is negative.

| | |
|---|---|
| Sample size | No statistical methods were used to pre-determine sample sizes, but our sample sizes are similar to those reported in previous publications (Li, T. Y. et al., Nat Aging, 2021; Savini, M. et al., Nat Cell Biol, 2022; Houtkooper, R. H. et al. Nature, 2013). |
| Data exclusions | No data were excluded from the analysis, except for the C. elegans lifespan experiments (the reasons for censoring were the "exploded vulva" phenotype or worms that crawled off the plate). These reasons were pre-established before the beginning of the experiment (Li, T. Y. et al., Nat Aging, 2021; Savini, M. et al., Nat Cell Biol, 2022; Houtkooper, R. H. et al. Nature, 2013). |
| Replication | All experiments, except for the RNA-seq, were repeated at least twice and similar results were acquired. |
| Randomization | Except for the random allocation of C. elegans to experimental groups/treatments after large-scale synchronization, the experiments were not randomized. |
| Blinding | Investigators were not blinded to allocation during experiments and outcome assessment, except for the RNA-seq analyses (Figs. 1h, 3g and 5f), where data analysis was performed in a blind manner until the group-to-group comparison steps were reached. |

# Reporting for specific materials, systems and methods

We require information from authors about some types of materials, experimental systems and methods used in many studies. Here, indicate whether each material, system or method listed is relevant to your study. If you are not sure if a list item applies to your research, read the appropriate section before selecting a response.

### Materials & experimental systems

| n/a | Involved in the study |
|---|---|
| ☐ | ☒ Antibodies |
| ☐ | ☒ Eukaryotic cell lines |
| ☒ | ☐ Palaeontology and archaeology |
| ☐ | ☒ Animals and other organisms |
| ☒ | ☐ Human research participants |
| ☒ | ☐ Clinical data |
| ☒ | ☐ Dual use research of concern |

### Methods

| n/a | Involved in the study |
|---|---|
| ☒ | ☐ ChIP-seq |
| ☒ | ☐ Flow cytometry |
| ☒ | ☐ MRI-based neuroimaging |

## Antibodies

| | |
|---|---|
| Antibodies used | Western blots were carried out with antibodies against green fluorescent protein (GFP) (Cat. 2956, CST, 1:1000, RRID:AB_1196615), β-amyloid 1–16 (6E10) (Cat. 803001, BioLegend, 1:1000, RRID:AB_2564653), Tubulin (Cat. T5168, Sigma, 1:2000, RRID:AB_477579), H3K27Ac (Ab4729, abcam, 1:1000, RRID:AB_2118291), H3K9Ac (Cat. 06-942, 1:1000, Merck, RRID:AB_310308), H3K4Ac (Cat. Ab176799, abcam, 1:1000, RRID:AB_2891335), Histone 3 (Cat. 9715, CST, 1:2000, RRID:AB_331563). The antibody for worm CPL-1 |

(1:5000) was a kind gift from Prof. Xiaochen Wang (SUSTech), as described and validated previously (Sun, Y. et al. Elife, 2020). Horseradish peroxidase (HRP)-labeled anti-rabbit (Cat. 7074; CST, 1:5,000, RRID:AB_2099233) and anti-mouse (Cat. 7076; CST; 1:5,000, RRID:AB_330924) secondary antibodies were applied.

**Validation**

Anti-GFP (CST, Cat. 2956); Suitable for: WB, IHC; Reacts with: all species
https://www.cellsignal.com/products/primary-antibodies/gfp-d5-1-rabbit-mab/2956
Anti-β-amyloid 1–16 (6E10) (BioLegend, Cat. 803001); Suitable for: WB, ELISA, IHC-P; Reacts with: Human
https://www.biolegend.com/de-de/products/purified-anti-beta-amyloid-1-16-antibody-11228?GroupID=BLG15648
Anti-Tubulin (Sigma, Cat. T5168); Suitable for: WB, IF, RIA; Reacts with: Mouse, Chicken, Chlamydomonas, African green monkey, Human, Rat, Bovine, Sea urchin, Kangaroo rat
https://www.sigmaaldrich.cn/CN/zh/product/sigma/t5168
Anti-H3K27Ac (Abcam, Ab4729); Suitable for: ICC/IF, WB, IHC-P, ChIP, PepArr; Reacts with: Mouse, Rat, Cow, Human, Recombinant fragment
https://www.abcam.cn/products/primary-antibodies/histone-h3-acetyl-k27-antibody-chip-grade-ab4729.html
Anti-H3K9Ac (Merck, Cat. 06-942); Suitable for: WB, ChIP, DB, FC, ChIP-seq; Reacts with: Human, Mouse, Rat
https://www.merckmillipore.com/CN/zh/product/Anti-acetyl-Histone-H3-Lys9-Antibody,MM_NF-06-942
Anti-H3K4Ac (Abcam, Cat. Ab176799); Suitable for: WB, ICC/IF, DB, ChIP, PepArr, ChIP-seq; Reacts with: Mouse, Human
https://www.abcam.cn/products/primary-antibodies/histone-h3-acetyl-k4-antibody-epr16596-chip-grade-ab176799.html
Anti-Histone 3 (CST, Cat. 9715); Suitable for: WB; Reacts with: Human, Mouse, Rat, Monkey, Zebrafish, Bovine, Pig
https://www.cellsignal.com/products/primary-antibodies/histone-h3-antibody/9715
Anti-CPL-1; Suitable for: WB; Reacts with: C. elegans; Validated by the producer in their publication:
Sun, Y. et al. Lysosome activity is modulated by multiple longevity pathways and is important for lifespan extension in C. elegans. Elife 9:e55745, doi: 10.7554/eLife.55745 (2020).

# Eukaryotic cell lines

Policy information about cell lines

| Cell line source(s) | N.A. |
|---|---|
| Authentication | N.A. |
| Mycoplasma contamination | N.A. |
| Commonly misidentified lines (See ICLAC register) | N.A. |

# Animals and other organisms

Policy information about studies involving animals; ARRIVE guidelines recommended for reporting animal research

**Laboratory animals**

The N2 (Bristol) strain was employed as the wild-type strain. IA123 (ijIs10[cpr-5::GFP-NLS::lacZ + unc-76(+)]), CB1370 [daf-2(e1370)], CF1038 [daf-16(mu86)], VC222 [raga-1(ok386)], RB754 [aak-2(ok524)], DA465 [eat-2(ad465)], VC3201 [atfs-1(gk3094)], OP56 (gaEx290 [elt-2::TY1::EGFP::3xFLAG(92C12) + unc-119(+)]), CL2122 (dvIs15 [pPD30.38] unc-54(vector) + (pCL26) mtl-2::GFP]), GMC101 (dvIs100 [unc-54p::A-beta-1-42::unc-54 3'-UTR + mtl-2p::GFP]), AM140 (rmIs132 [unc-54p::Q35::YFP]), AM725 (rmIs290 [unc-54p::Hsa-sod-1(127X)::YFP]), DA2123 (adIs2122 [lgg-1p::GFP::lgg-1 + rol-6(su1006)]), GRU101(gnaIs1[myo-2p::yfp]), CA1200 (ieSi57 [eft-3p::TIR1::mRuby::unc-54 3'UTR + Cbr-unc-119(+)] II), JIN1375 [hlh-30(tm1978) IV], atfs-1(tm4525) V, SM190 [pha-4(zu225);smg-1(cc546ts)], GRU102 (gnaIs1[myo-2p::yfp + unc-119p::Aβ1-42]), HZ1683 [atg-2(bp576)], HZ1684 [atg-3(bp412)], HZ1687 [atg-9(bp564)] and HZ1688 [atg-13(bp414)] were provided by the Caenorhabditis Genetics Center (CGC, University of Minnesota) or the National Bioresource Project (NBRP). The XW19180 [hsp-16.2p::nuc-1::pHTomato] strain was a kind gift from Prof. Xiaochen Wang (SUSTech).The MQD2491[daf-16(hq389[daf-16::gfp::degron]) I; ieSi57[eft-3p::TIR1::mRuby::unc-54 3'UTR+Cbr-unc-119(+)] II; unc-119(ed3) III; daf-2(e1370ts) III] strain was a kind gift from Prof. Meng-Qiu Dong (NIBS). Hermaphrodite and L4/young adult worms were used for analyses throughout the study, unless otherwise specified in the figure legends or methods.

For generation of the strains with GFP-tag of vha-1, vha-14, vha-15, vha-16 and vha-20, the constructs (vha-1, Clone: 9473457628999774 E12; vha-14, Clone: 8859124759762056 C08; vha-15, Clone: 3304493055384826 B08; vha-16, 2491680425634929 G12; vha-20, Clone: 5745981749165295 F12) were obtained from Prof. Mihail Sarov, as part of the TransgeneOme project (https://transgeneome.mpi-cbg.de/). The constructs were injected at 10-60 ng/μL along with a co-injection marker pRF4 (rol-6) at 40 ng/μL to generate transgenic lines. Strains were made by the SunyBiotech Co.

A new UV-integrated N2 background cpr-5 reporter strain TYL001 (cpr-5p::gfp + rol-6) for optimal LySR activation detection was also constructed, which is available upon request. To construct this strain, the 1018 bp cpr-5 promoter was amplified with the following primers: 5'- GAATTGACATGCACTCCGGC-3' and 5'-AAGAATAGCGGAGAGCTTCC-3' and ligated in frame with eGFP in a pPD95.75 expression vector (Addgene, #184130). The construct was then injected at 50 ng/μL along with a co-injection marker pRF4 (rol-6) at 40 ng/μL. Extra-chromosomal arrays were integrated using UV irradiation and backcrossed two times to N2, non-roller worms were maintained afterwards.

For generation of the TYL002 (ges-1p::cpr-5-DsRed::SL2::GFP + rol-6) worm strain, the ges-1 promoter (amplified from the pJL3 plasmid, Addgene #184131), the cpr-5 protein coding sequence (amplified from worm total cDNA) and DsRed sequence (amplified from the pJL6 plasmid, Addgene #184134), were ligated into a pPD95.77_SL2 vector backbone (Addgene #184129), between SphI and XmaI restriction sites. The construct was then injected at 25 ng/μL along with a co-injection marker pRF4 (rol-6) at 40 ng/μL to generate transgenic lines.

For generation of the knock-in worm strains with endogenously GFP/Degron-mNG tagged VHA-6 [TYL003 (vha-6p::vha-6::gfp) and

TYL004 (vha-6::Degron::mNG)] and ELT-2 [TYL005 (elt-2::Degron::mNG)], the CRISPR/Cas9 engineering was performed by microinjection using the homologous recombination approach (Paix, A. et al. Genetics, 2015). The microinjection mixture consisted of 300 mM of KCl, 20 mM of HEPES, 100 ng/μL of tracrRNA (Cat. U-002005, Dharmacon), 50 ng/μL of crRNA targeting vha-6 or elt-2, 200 ng/μL of DNA repair template for vha-6 or elt-2, 0.25 μg/μL of Cas9 protein (Cat. CAS9PROT-250UG, Sigma), 200 ng/μL dpy-10 crRNA and 200 ng/μL dpy-10 repair template. To generate the homologous recombination DNA repair templates, two homologous arms (~1000 bp each) corresponding to the 5'- and 3'- sides of the insertion site and the GFP/Degron-mNG tags were cloned in a vector and then amplified altogether. Plasmids were injected into the gonad of young adult hermaphrodite worms using standard method. F1s with roller phenotype were singled on a new NGM plate and allowed to produce sufficient offspring. Successful knock-in events were screened by PCR genotyping from independent F1 transgenic animals' progeny that did not display roller phenotype, and further confirmed by DNA sequencing. The crRNAs and cloning primers used to generate the two strains are listed in Supplementary Table 6.

For generation of the strains expressing vha-6 and elt-2 promoter-driven mCherry TYL006 (vha-6p::mCherry; vha-6p::vha-6::gfp) and TYL007 (elt-2p::mCherry; elt-2::Degron::mNG)], promoters of vha-6 or elt-2 were amplified with the following primers: 5'-TCGGTAAGTTGCTACTTCAG-3' and 5'-TTTTTATGGGTTTTGGTAGGTTTTAG-3' for vha-6 promoter; 5'-ATTATATGAAAACTAATGAG-3' and 5'-TCTATAATCTATTTTCTAGTTTCTATTTTATT-3' for elt-2 promoter. The PCR products were then ligated in frame with mCherry in a pPD95.75 expression vector (Addgene, #184130). The constructs were then injected into the gonad of their corresponding TYL003 (vha-6p::vha-6::gfp) and TYL004 (elt-2::Degron::mNG) strains at 50 ng/μL along with a co-injection marker pRF4 (rol-6) at 40 ng/μL.

| Wild animals | No wild animals were used in the study. |
| Field-collected samples | No field collected samples were used in the study. |
| Ethics oversight | This study did not require an ethical approval. |

Note that full information on the approval of the study protocol must also be provided in the manuscript.

