## [Peer Review File · Nature Cell Biology]

A lysosomal surveillance response to stress extends healthspan

Corresponding Author: Professor Johan Auwerx

Version 0:

Decision Letter:

*Please delete the link to your author homepage if you wish to forward this email to co-authors.

Dear Professor Auwerx,

Thank you again for submitting your manuscript, "A lysosomal surveillance response (LySR) that boosts lysosomal activity and extends healthspan", to Nature Cell Biology. It has now been seen by 3 referees, who are experts in lysosomes (Referee #1); aging, longevity, metabolism, *C. elegans* (Referee #2); and aging, autophagy, *C. elegans* (Referee #3). As you will see from their comments (attached below), they found the work of potential interest but have raised substantial concerns, which in our view would need to be addressed with considerable revisions before we can consider publication in Nature Cell Biology.

As per our standard process, Nature Cell Biology editors discuss the referee reports in detail within the editorial team, including the chief editor, to identify key referee points that should be addressed with priority, and requests that are overruled as being beyond the scope of the current study. To guide the scope of the revisions, I have listed these points below. Our standard revision period is six months, and we are committed to providing a fair and constructive peer-review process, so please feel free to contact me if you would like to discuss any of the referee comments further or if you anticipate any issues addressing the reviews.

I should stress that the referees' concerns point to a premature dataset and their concerns would need to be addressed thoroughly experimentally; reconsideration of the study for this journal and re-engagement of referees will depend on the strength of these revisions. In particular, it would be essential to dedicate efforts to address the reviews as follows:

1- The reviewers had significant concerns with the approaches used and requested stronger and clearer imaging data, controls for longevity analyses, mutant studies to confirm RNAi studies, and more clarity when it comes to the analysis of V-ATPase function and lysosome acidification. We find these points essential to address experimentally:

Rev#1 points #3-4-5

Rev#2 points #1-2-3-5-6-7-8-11-12-13

Rev#3 paragraphs "The use of dilution of the RNAi-containing E. ..."; "The use of mixed RNAi .."; "Even though *h1h-30* is not.." "Replication for lifespan studies .."; "A more robust genetic probe.."

2- Importantly, the reviewers were not yet convinced by the mechanism – both the importance of LySR and the roles of key molecules, *ELT-2* in particular:

Rev#1 points #1-2 (the first point includes broad questions. We think out of these, "How and where is *elt-2* activated?" is important)

Rev#2 point #4

Rev#3 paragraphs "A trivial possibility for *elt-2*'s ability .." "elt-2 is involved in establishment .." "The evidence, similarly, that *DR..*"

3- All other referee concerns pertaining to strengthening existing data, minor points, providing controls, methodological details, clarifications and textual changes, should also be addressed. We suggest considering discussing more of the relevant literature if possible, such as <https://doi.org/10.7554/eLife.55745>; <https://doi.org/10.1073/pnas.2104832118>; <https://doi.org/10.1038/ncb2741>, to enrich the discussion.

4- Finally, please pay close attention to our guidelines on statistical and methodological reporting (listed below) as failure to do so may delay the reconsideration of the revised manuscript. In particular, please provide:

We would be happy to consider a revised manuscript that would satisfactorily address these points, unless a similar paper is published elsewhere or is accepted for publication in Nature Cell Biology in the meantime.

- ensure that it conforms to our format instructions and publication policies (see below and <https://www.nature.com/nature/for-authors>).
- provide a point-by-point rebuttal to the full referee reports verbatim, as provided at the end of this letter.
- provide the completed Reporting Summary (found here <https://www.nature.com/documents/nr-reporting-summary.pdf>). This is essential for reconsideration of the manuscript will be available to editors and referees in the event of peer review. For more information see <http://www.nature.com/authors/policies/availability.html> or contact me.

Nature Cell Biology is committed to improving transparency in authorship. As part of our efforts in this direction, we are now requesting that all authors identified as 'corresponding author' on published papers create and link their Open Researcher and Contributor Identifier (ORCID) with their account on the Manuscript Tracking System (MTS), prior to acceptance. ORCID helps the scientific community achieve unambiguous attribution of all scholarly contributions. You can create and link your ORCID from the home page of the MTS by clicking on 'Modify my Springer Nature account'. For more information please visit please visit www.springernature.com/orcid.

This journal strongly supports public availability of data. Please place the data used in your paper into a public data repository, or alternatively, present the data as Supplementary Information. If data can only be shared on request, please explain why in your Data Availability Statement, and also in the correspondence with your editor. Please note that for some data types, deposition in a public repository is mandatory - more information on our data deposition policies and available repositories appears below.

Link Redacted

We hope that you will find our referees' comments and editorial guidance helpful. Please do not hesitate to contact me if there is anything you would like to discuss. Many thanks again for considering NCB for your work.

Best wishes,

Melina

Melina Casadio, PhD
Senior Editor, Nature Cell Biology
ORCID ID: <https://orcid.org/0000-0003-2389-2243>

Reviewers' Comments:

Reviewer #1:

Remarks to the Author:

In this manuscript Yang Li et al compare the effects of silencing several subunits of the V-ATPase in *C. elegans*. Notably, they report that silencing some subunits, e.g. vha-6, vha-8, vha-14, extend the lifespan of the worms, while others like vha-16 and vha-19 shorten the lifespan. Transcriptomic analyses showed that while upwards of 3,000 transcripts are upregulated in all cases, 760 additional transcripts increase only when the lifespan-enhancing V-ATPase subunits are silenced.

The authors then proceeded to show that the altered lifespan is associated with and requires the induction and activation of the elf-2 transcription factor. Moreover, they found that the subunits that extend the lifespan of the worms are components of the V-ATPase located in the luminal membrane of intestinal cells, while the subunits that reduce lifespan when silenced are expressed more generally, possibly ubiquitously.

Silencing the apical subunits caused alkalinization of the intestinal lumen and triggered the transcription of genes involved in the innate immune response, lysosomal function, metabolic processes and proteolysis. The authors proceeded to analyze the lysosomal surveillance response (LySR) triggered by inactivation of the intestinal V-ATPase and attributed the extended lifespan to enhanced lysosomal activity.

The data are clear and compelling, and the reported enhancement of the worm lifespan is impressive. The involvement of elf-2 is equally convincing. The following important aspects remain unclear:

- 1) How and where is elf-2 activated? Some of the data imply that elf-2 and the LySR manifest themselves in multiple (possibly all?)

tissues. How is the alkalization of the intestinal lumen sensed remotely by other tissues? The authors explored the logical possibility that diminished acidification causes impaired transport of dipeptides and other solutes, possibly resulting in dietary restriction. However, dietary restriction was shown by the authors to account for a minute fraction of the response. It is essential to document how changes in the intestinal lumen pH alter gene expression in non-intestinal tissues, which ones are affected and, critically, how the signal is conveyed.

2) The contribution of the LySR to longevity was favored by the authors, who seem to have dismissed the contributions of the innate immune response and of metabolic alterations. Can these components be segregated and their contribution assessed independently? If not, is the selective focus on LySR justified at this stage?

3) For reasons that are not clear, the authors used three different probes to monitor acidification of the intestinal lumen and of lysosomes, leading to results that are internally inconsistent. In Extended data Fig 5 the authors clearly show that acridine orange accumulates in lysosomes even in vha-16 and vha-19 knockdowns. In fact, the lysosome stain more brightly in these cases than in the vha-6, vha-8 etc. knockdowns, and even more than in the control worms. In contrast, the authors used LysoSensor in Fig. 6 to claim that vha-16 and vha-19 knockdown dissipates lysosomal acidification. These findings are incompatible, as is the accumulation of LysoTracker in the intestine of vha-6, vha-8 etc. knockdowns.

4) Are vha-16 and vha-19 not required for intestinal V-ATPase function? If they are and the intestinal lumen pH is affected (as suggested by Fig 6), why is their response different from that of vha-6? And if, as claimed by the authors, LysoSensor is more responsive to pH than LysoTracker, why is the former not accumulated in the intestinal lumen in Fig. 6?

5) If vha-16 and vha-19 are required for lysosomal acidification, what accounts for LysoTracker accumulation in the lysosomes?

Reviewer #2:

Remarks to the Author:

Summary Description

The manuscript titled "A lysosomal surveillance response (LySR) that boosts lysosomal activity and extends healthspan" presents a novel longevity mechanism. The study reveals that silencing specific vacuolar H⁺-ATPase subunits improves health outcomes and extends *Caenorhabditis elegans* lifespan significantly. The longevity is attributed to the activation of a transcriptional response termed the Lysosomal Surveillance Response (LySR), which is characterized by induction of the cathepsin encoding gene *cpr-5*. The evidence also shows that the transcription factor ELT-2 governs lysosomal gene expression.

The work is novel and potentially impactful. In particular, the conclusions that vha-6 inactivation causes longevity is very well supported by evidence. By contrast, the assertion that the relevant common feature of the longevity-causing VHAs is reduced luminal function while preserving (or enhancing) lysosomal function is not well supported. Similarly, vha-6 longevity can be decoupled from improved proteostasis (via chloroquine treatment), and ELT-2 is more likely regulating basal expression of lysosomal genes rather than the activation of the LySR program upon vha-6 inactivation. Therefore, important aspects remain unclear or need further validation.

Major Concerns

1. Resolution of *C. elegans* Images:

The hypothesis that the common feature of the VHA proteins that trigger LysR is their luminal localization and function is intriguing. Unfortunately, *C. elegans* images presented to substantiate this hypothesis lack sufficient resolution. For instance, it's not possible to define VHA-6's localization in Ext. Fig. 1-N. The same issue affects Ext. Figs. 1 O to S and others throughout the manuscript.

Recommendation: Use higher magnification, confocal images to show the anatomical and subcellular expression pattern of all relevant VHA proteins, especially VHA-6. Present VHA-6 anatomical and subcellular localization as a Main Fig. 1 panel.

2. Developmental Stage and Fertility:

Another caveat of using low magnification images is that it's not possible to visualize the features that would allow the reader to judge the developmental stage of the presented worms. vha-6 RNAi-treated animals seem to have a penetrant 'small size' phenotype. However, the presented images prevent the reader from judging whether the vha-6 animals are small but the same age as the controls or not (e.g., do they have adult vulvae, do they contain eggs, etc.). Therefore, the authors may want to display higher-magnification images that show the developmental or reproductive stage of the assessed animals. Stage/age of the animals is critical because the signal of the fluorescent reporters (e.g., hsp-6P, hsp-4P, etc.) varies with age; hence, if the vha-6 animals were chronologically the same age but developmentally delayed, the dimmer stress-reporter signals would need to be reinterpreted.

Recommendation: Display higher magnification images showing the developmental or reproductive stage of the assessed animals to confirm their age and developmental stage. Assess and document total egg output and rate of egg-laying relative to controls.

In a similar line of thought, vha-6 RNAi seems to barely affect body size in Fig. 2E; even when, *cpr-5::GFP* is robustly activated. Does this suggest that the LysR response and the body size effect of vha-6 inactivation can be decoupled?

3. Controls for Survival Curves:

It is critical that the authors present the controls in Figs. 2 G-L, meaning showing the survival curves corresponding to worms fed control (L4440) only and pro-longevity RNAi (e.g., *daf-2*) only. Also, if needed, reassess the conclusions after pairwise comparing all four conditions present in each figure panel.

4. ELT-2 Regulation of VHA Genes:

The data suggest that ELT-2 would regulate the basal expression of the vha genes of interest. Therefore, the *elt-2* qPCR data needs to be presented in a way that enables the reader to see the full effect of ELT-2 on vha genes' expression. Hence, the authors may want to present the qPCR data using a logarithmic scale Y axis. If, as it seems, ELT-2 controls the basal expression of vha genes, in particular vha-6, then it is critical that the authors focus on the "delta" expression to define the extent to which ELT-2 activates gene expression upon vha-6 RNAi. If the data would not support the initial conclusion that ELT-2 activates vha-6-responsive genes, but instead it controls

their basal expression, the authors would need to reframe ELT-2's role across the manuscript from abstract to model figure and state that identifying the TF/s that activate the vha-6 RNAi-responsive genes is important and will be the focus of future work. Finally, if the activation of CPR-5 is a major contributor to the pro-longevity effect of ELT-2 activation, then overexpressing CPR-5 should rescue elt-2 RNAi shortened lifespan phenotype.

Recommendation: Present qPCR data using a logarithmic scale Y axis. Focus on "delta" expression to define the extent of ELT-2 activation upon vha-6 RNAi. If necessary, reframe ELT-2's role throughout the manuscript and state future focus on identifying TF/s activating vha-6 RNAi-responsive genes.

5. RNAi Penetration and Double RNAi Experiments:

The authors performed double RNAi to identify the transcription factors that activate the LySR. Because RNAi is not necessarily 100% penetrant, works worse when two genes are simultaneously inactivated, and some tissues (e.g., pharynx and neurons) are resistant to RNAi, the conclusions of the experiment presented in Fig. 3.c. need to be toned down.

6. Expression Patterns and Gut Luminal pH:

Extended Fig. 5A shows data critical to the central model presented in the manuscript; however, the images are insufficient to draw the conclusions. It is unclear whether the single worm shown is representative. If it is representative, then the conclusion that VHA-6 is predominantly or exclusively expressed in the luminal side of the intestinal cells might have been misled by the fact that the intestinal cells themselves are severely shrunk in the vha-6 RNAi animals, giving the impression that expression follows a thin line in the center of the intestine when in fact the line may be the whole intestinal

Recommendation: Present higher magnification images of the intestine of several worms in supplementary data and a representative high magnification image in a main figure panel.

7. Lysosensor and LysoTracker Staining:

Lysosensor-lysoTracker staining is notoriously fastidious in the worm. Furthermore, the signal is dim and easily bleached. Therefore, it would be important for the authors to document the single staining with Lysosensor and LysoTracker, in parallel with staining with both dyes simultaneously (as now shown). This will permit discrimination of autofluorescence and bleed-through from true Lysosensor and LysoTracker signals.

Recommendation: Document single staining with Lysosensor and LysoTracker in parallel with simultaneous staining to discriminate autofluorescence and bleed-through.

Revise model to reflect that pro-longevity vha genes affect both gut lumen and lysosomal acidification.

8. Orange Acridine Staining:

The authors interpret the data presented in Extended data Fig. 5 C as VHA-6, 8, 14, 15, and 20 mostly affecting the luminal pH; however, dimmer acridine orange signal is also evident in the lysosomal compartment upon RNAi against these genes. In particular, while vha-16 and 19 show intense red/orange lysosomal signal, the long-lived vha inactivations show much weaker signal. This observation asks for the authors to revise their model to reflect that the pro-longevity vha genes affect gut lumen AND lysosomal acidification.

9. Small Size Phenotype and CQ Treatment:

Do the data presented in Ext. Data Fig. 7 c-d suggest that CQ partially rescues the small size phenotype of vha-6 RNAi animals? For example, compare AM725 on vha-6 without CQ to 5mM CQ.

Recommendation: Discuss this observation.

10. Conclusion on Movement and Healthspan:

The following conclusion needs to be toned down: "...as well as movement, were increased and sustained throughout the life history,..." The authors only tested up to day 8 (middle age); hence, they can conclude that movement was sustained later in adult life but not throughout the life history. More importantly, the statement "Collectively, these results highlight that activation of LySR by vha-6 RNAi reduces protein aggregates and extends organismal healthspan." contradicts the observation that 1mM CQ rescues increased proteotoxicity without reducing lifespan. Thus, the authors may need to revise their main model and include all data to draw their conclusions and frame the discussion.

Recommendation: Conclude that movement is sustained later in adult life but not throughout the entire life. Revise the boosted proteostasis and lysosomal function model and discussion to reflect all data, including acridine orange and CQ rescue effects.

11. Role of HLH-30/TFEB:

As justified in Item #5, the conclusions related to the potential role of HLH-30/TFEB in this pathway need to be toned down/revise because RNAi does not ensure full body 100% reduction in the activity of this TF.

Recommendation: Tone down/revise conclusions on HLH-30/TFEB's role in the pathway or repeat experiments using hlh-30 mutant.

12. Intestinal Apical Membrane Localization:

P9. L19: Present high-resolution and higher magnification images to validate this conclusion or revise "specific intestinal apical membrane-localized".

13. Lysosome integrity data Interpretation:

Evidence in Ext. Data. Fig. 5C shows that inactivation of the long-lived vha genes also affects the pH of the lysosomes. Therefore, "but not the lysosomes" in P9. L21 and P9. L25 "Importantly, the beneficial effects of vha-6 RNAi strongly depend on the intact function of lysosomes" need to be revised to accurately reflect the data.

Suggestions

Based on Extended Data Fig. 6 C vs D, discuss that longevity seems to be possible even when proteostasis is compromised.

Methods Needing Clarification or Improvement

1. Worm Incubation Conditions:

Authors indicate not using 5-FU at the same time that they state that animals are incubated at 20°C and transferred once a week. These conditions need revision as at least two generations of worms would grow in a week in the absence of chemical sterilization or transfer.

2. Housekeeping Genes for qPCR:

act-3 and pmp-3 are poor housekeeping genes (see DOI: 10.1126/sciadv.adg0506).

Consider alternative housekeeping genes based on recent literature.

Typos and Grammar Corrections

Correct typographical and grammatical errors throughout the manuscript, including:

- o L. 23: evidences -> evidence

- o L. 25: decreases with aging -> decreases with age

- o P4. L15: in reaction to -> in relation to

- o P5. L 47: reorganizations -> reorganization

- o P7. L47: Treatment of -> Treatment with

- o P8. L1: induced reduction of AB aggregates -> induced reduction of Aβ aggregates

- o P8. L33: systematic -> systemic

- o P9.L21: appear

- o P9. L36: alterations

Conclusion

The manuscript presents novel findings on the role of lysosomal activity in aging and healthspan. Addressing the major and minor concerns outlined above will strengthen the manuscript and enhance its clarity and impact.

Reviewer #3:

Remarks to the Author:

This manuscript, by Auwerx and colleagues, puts forward the notion of a lysosome function surveillance pathway they term LySR which senses and protects lysosome function via a transcriptional response to threatened lysosome function. LySR activation is dependent upon the GATA transcription factor ELT-2 in *C. elegans* and is associated with proteostatic and lifespan benefit. The finding that knockdown of essential VHA genes extends lifespan had been made almost 20 years ago by the Ruvkun lab (appropriately cited by this study), and this study aims to extend those findings by adding mechanistic understanding of the pathways activated to promote lifespan extension. The findings are interesting and timely, however over-reliance on RNAi as a methodology and moreover large-scale use of questionable methodology including mixed RNAi (even though "quality controlled" with assessment of knockdown efficiency) lessen the rigor with which the conclusions are substantiated. As well the study is missing some critical longevity experiments, in their place using marker gene expression as a surrogate which could be inconclusive. So while interesting, the findings, although extensive, are not well substantiated by the methodologies used and fundamental concerns remain about the soundness of conclusions, especially when factors invoked in LySR such as *elt-2* and *cbp-1* have tremendous pleiotropy (establishment of intestinal identity and largescale remodeling of the epigenome, respectively).

Major:

The use of dilution of the RNAi-containing *E. coli* food source is an imprecise way to target genes by knockdown. From the analyses conducted, it appears as though there are not simply different dose responses to reduction in apparently deleterious RNAi like *vha-16* and *vha-19* versus the beneficial knockdowns exemplified by *vha-6*, but the authors may want to conclusively prove this by testing even lower concentrations of *vha-16* (as in Ext Data Fig. 2B-C).

The use of mixed RNAi has variable and unpredictable efficacy in *C. elegans*. Thus, critical conclusions such as those made in figure 3C, where TFs assessed for their orchestration of LySR, could represent false negative results. These data should be generated with true null, loss of function alleles, at least for *TFEB/hlh-30*, which would be expected to contribute to the defense against lysosomal insult based upon its known biology.

A trivial possibility for *elt-2*'s ability to reverse the *vha-6* RNAi phenotype is that mixed RNAi to both factors somehow alters efficiency of *vha-6* and/or *elt-2* RNAi (positively or negatively). In the data in supplementary materials showing knockdown efficiency, there are significant differences between factors alone versus mixing. For a critical result such as this (*elt-2* dependency), a questionable methodology such as mixed RNAi is simply not a rigorous proof. An orthogonal way of depleting either *vha-6* or *elt-2* should be used such as auxin induced degradation or similar so that mixed RNAi is not required.

Even though hih-30 is not required for cpr-5 expression or for the apparent proteostasis advantages of vha-6 RNAi, its necessity for lifespan extension should be tested using null mutants. Given that starvation and mTOR depend upon the FoxA ortholog pha-4 and mTOR activity is critically linked to the lysosome, the authors should also test dependence on TS pha-4 null mutants, both readily available

Negative control regions not predicted to associate with ELT-2 binding are missing from figure 3K.

Replication for lifespan studies is not indicated in the figure legends, nor are biological replicates indicated in a summary table in supplementary information, as is standard practice in the field. In order to rigorously support conclusions, lifespans should be repeated a minimum of 3 times with sufficient numbers of animals, and this information made available to the reader (e.g. in a supplementary table as needed).

elt-2 is involved in establishment of intestinal cell identity, and so a question arises as to whether loss of elt-2 non-specifically mitigates the ability to respond to intestinal insults such as vha-6 RNAi versus specific involvement in the LySR. More exhaustive characterization of specific binding to LySR genes versus control genes is needed to substantiate this claim.

The evidence, similarly, that DR "hijacks" the LySR machinery is poorly substantiated, especially given the degree of transcriptional signature similarity. Elt-2 dependency, which again could be about core intestinal cell identity rather than true mechanistic intersection, is a poor correlate. Lack of dependency on AMPK and daf-16 speak strongly against this conclusion. Additionally, the epistasis of elt-2 to DR appears partial and/or parallel at best.

Minor:

Although *C. elegans* labs will understand the jargon (feeding *E. coli* expressing double stranded RNA to conduct RNA interference), the use of the term "fed" to indicate the practice of RNAi, e.g. p.3 line 13, should be avoided.

A more robust genetic probe for lysosomal pH, e.g. pHluorin, could provide more reliable, parallel assessment of lysosomal pH versus dyes, the uptake of which can be compromised in states of loss of lysosomal function and intestinal integrity, though this concern is offset by the examination of lysosome activity with CPL-1 cleavage.

REFERENCES – are limited to a total of 70 for Articles, Resources, Technical Reports; and 40 for Letters. This includes references in the

main text and Methods combined. References must be numbered sequentially as they appear in the main text, tables and figure legends and Methods and must follow the precise style of Nature Cell Biology references. References only cited in the Methods should be numbered consecutively following the last reference cited in the main text. References only associated with Supplementary Information (e.g. in supplementary legends) do not count toward the total reference limit and do not need to be cited in numerical continuity with references in the main text. Only published papers can be cited, and each publication cited should be included in the numbered reference list, which should include the manuscript titles. Footnotes are not permitted.

Methods should be written concisely, but should contain all elements necessary to allow interpretation and replication of the results. As a guideline, Methods sections typically do not exceed 3,000 words. The Methods should be divided into subsections listing reagents and techniques. When citing previous methods, accurate references should be provided and any alterations should be noted. Information must be provided about: antibody dilutions, company names, catalogue numbers and clone numbers for monoclonal antibodies; sequences of RNAi and cDNA probes/primers or company names and catalogue numbers if reagents are commercial; cell line names, sources and information on cell line identity and authentication. Animal studies and experiments involving human subjects must be reported in detail, identifying the committees approving the protocols. For studies involving human subjects/samples, a statement must be included confirming that informed consent was obtained. Statistical analyses and information on the reproducibility of experimental results should be provided in a section titled "Statistics and Reproducibility".

All Nature Cell Biology manuscripts submitted on or after March 21 2016 must include a Data availability statement as a separate section after Methods but before references, under the heading "Data Availability". For Springer Nature policies on data availability see <http://www.nature.com/authors/policies/availability.html>; for more information on this particular policy see <http://www.nature.com/authors/policies/data/data-availability-statements-data-citations.pdf>. The Data availability statement should include:

- Accession codes for primary datasets (generated during the study under consideration and designated as "primary accessions") and secondary datasets (published datasets reanalysed during the study under consideration, designated as "referenced accessions"). For primary accessions data should be made public to coincide with publication of the manuscript. A list of data types for which submission to community-endorsed public repositories is mandated (including sequence, structure, microarray, deep sequencing data) can be found here <http://www.nature.com/authors/policies/availability.html#data>.
- Unique identifiers (accession codes, DOIs or other unique persistent identifier) and hyperlinks for datasets deposited in an approved repository, but for which data deposition is not mandated (see here for details <http://www.nature.com/sdata/data-policies/repositories>).
- At a minimum, please include a statement confirming that all relevant data are available from the authors, and/or are included with the manuscript (e.g. as source data or supplementary information), listing which data are included (e.g. by figure panels and data types) and mentioning any restrictions on availability.
- If a dataset has a Digital Object Identifier (DOI) as its unique identifier, we strongly encourage including this in the Reference list and citing the dataset in the Methods.

We recommend that you upload the step-by-step protocols used in this manuscript to [protocols.io](https://www.protocols.io). More details can be found at <https://www.protocols.io/help/publish-articles>.

All imaging data should be accompanied by scale bars, which should be defined in the legend. Cropped images of gels/blots are acceptable, but need to be accompanied by size markers, and to retain visible background signal within the linear range (i.e. should not be saturated). The boundaries of panels with low background have to be demarked with black lines. Splicing of panels should only be considered if unavoidable, and must be clearly marked on the figure, and noted in the legend with a statement on whether the samples were obtained and processed simultaneously. Quantitative comparisons between samples on different gels/blots are discouraged; if this is unavoidable, it should only be performed for samples derived from the same experiment with gels/blots were processed in parallel, which needs to be stated in the legend.

The total number of Supplementary Figures (not including the "unprocessed scans" Supplementary Figure) should not exceed the number of main display items (figures and/or tables (see our Guide to Authors and March 2012 editorial <http://www.nature.com/ncb/authors/submit/index.html#suppinfo>; <http://www.nature.com/ncb/journal/v14/n3/index.html#ed>). No restrictions apply to Supplementary Tables or Videos, but we advise authors to be selective in including supplemental data.

GUIDELINES FOR EXPERIMENTAL AND STATISTICAL REPORTING

REPORTING REQUIREMENTS – We are trying to improve the quality of methods and statistics reporting in our papers. To that end, we are now asking authors to complete a reporting summary that collects information on experimental design and reagents. The Reporting Summary can be found here <https://www.nature.com/documents/nr-reporting-summary.pdf> If you would like to reference the guidance text as you complete the template, please access these flattened versions at <http://www.nature.com/authors/policies/availability.html>.

Version 1:

Decision Letter:

*Please delete the link to your author homepage if you wish to forward this email to co-authors.

Dear Professor Auwerx,

Thank you for your patience during the peer review process. Your manuscript, "A lysosomal surveillance response (LySR) that boosts lysosomal activity and extends healthspan", has now been seen by all of our original referees, who are experts in lysosomes (Referee #1); aging, longevity, metabolism, *C. elegans* (Referee #2); and aging, autophagy, *C. elegans* (Referee #3). As you will see from their comments (attached below) they find this work to be improved and of continued interest, but Reviewers #1 and #3 have raised some important points. Although we are also still interested in this study, we believe that their concerns should be addressed before we can consider publication in Nature Cell Biology.

Nature Cell Biology editors discuss the referee reports in detail within the editorial team, including the chief editor, to identify key referee points that should be addressed with priority, and requests that are overruled as being beyond the scope of the current study. To guide the scope of the revisions, I have listed these points below. We are committed to providing a fair and constructive peer-review process, so please feel free to contact me if you would like to discuss any of the referee comments further.

In particular, it would be essential to:

(A) Investigate why there is LysoTracker accumulation in *vha-16* and *vha-19* mutants, if this is not a sign of acidification, per the suggestions from Reviewer #1.

(B) Further discuss the lack of a clear mechanism as to how ELT-2 senses VHA-6 loss as a limitation of the study, per Reviewer #1's concerns. At the same time, please address all other discussion points outlined by Reviewer #3. Per our previous editorial feedback, we would also still encourage you to discuss relevant studies like <https://doi.org/10.1073/pnas.2104832118> & <https://doi.org/10.1038/ncb2741>

(C) From an editorial perspective, we would also ask that you:

- Remove the legends from all figure files, and instead provide the figures as full-page files. Please also ensure that font size in figures is at least size 7, to ensure readability.
- We do not allow for models or summary schematics to be standalone figures in the main text. Please move Fig. 8 to the final panel of Figure 7, or make it an Extended Data Figure (keeping in mind that there can be, at most, 10 Ext Data Figures).
- Please add text labels defining the units (ex. z-score) for the color gradient keys in Fig 1k and Ext Data Fig 5a
- Please note that exact p-values should be provided in the figures or figure legends, rather than ranges (the exception here would be for $p < 0.0001$, ****, when the p-value is too small to calculate).
- Please provide scalebars for the magnified subpanels in Fig 6a-b and Ext Data Fig 4c, 5b-c, 6a-e, 6i, 9b (including the merge)
- Please update the source data file with values underlying Fig 1a-g, 2c, 2f-l, 3l, 4g, 5d-e, 7h, & Ext Data Fig 1a-m, 1v, 2c, 2e, 4e, 7d, 10e-f (beyond the median/summary data provided in Supp Table 4)
- Please show individual data points in Fig 1m

(D) All other referee concerns pertaining to methodological details, clarifications and textual changes, should also be addressed.

(E) Finally please pay close attention to our guidelines on statistical and methodological reporting (listed below) as failure to do so may delay the reconsideration of the revised manuscript. In particular please provide:

In contrast, although we agree with referee XXX that YYY [EDITOR INCLUDE POINTS TO OVERRULE] would provide valuable insights, we consider this point to be beyond the scope of the present study. Thus, addressing it experimentally will not be necessary for reconsideration of the manuscript at this journal.

We therefore invite you to take these points into account when revising the manuscript. In addition, when preparing the revision please:

- ensure that it conforms to our format instructions and publication policies (see below and www.nature.com/nature/authors/).

- provide a point-by-point rebuttal to the full referee reports verbatim, as provided at the end of this letter.

- provide the completed Editorial Policy Checklist (found here <https://www.nature.com/authors/policies/Policy.pdf>), and Reporting Summary (found here <https://www.nature.com/authors/policies/ReportingSummary.pdf>). This is essential for reconsideration of the manuscript and these documents will be available to editors and referees in the event of peer review. For more information see <http://www.nature.com/authors/policies/availability.html> or contact me.

Nature Cell Biology is committed to improving transparency in authorship. As part of our efforts in this direction, we are now requesting that all authors identified as 'corresponding author' on published papers create and link their Open Researcher and Contributor Identifier (ORCID) with their account on the Manuscript Tracking System (MTS), prior to acceptance. ORCID helps the scientific community achieve unambiguous attribution of all scholarly contributions. You can create and link your ORCID from the home page of the MTS by clicking on 'Modify my Springer Nature account'. For more information please visit <http://www.springernature.com/orcid>.

Link Redacted

We would like to receive the revision within four weeks. If submitted within this time period, reconsideration of the revised manuscript will not be affected by related studies published elsewhere, or accepted for publication in Nature Cell Biology in the meantime. We would be happy to consider a revision even after this timeframe, but in that case we will consider the published literature at the time of resubmission when assessing the file.

We hope that you will find our referees' comments, and editorial guidance helpful. Please do not hesitate to contact me if there is anything you would like to discuss.

Best regards,

George Inglis

George Inglis, PhD
Senior Editor
<https://www.nature.com/ncb/research-cross-journal-editorial-team> Research Cross-Journal Editorial Team
Nature Cell Biology

Reviewers' Comments:

Reviewer #1 (Remarks to the Author):

The response to my original comments and requests was rather disappointing. The authors indicate in their reply that secretion of a cathepsin-like protease is the likely mechanism whereby silencing the epithelial V-ATPase subunits extends healthspan, ostensibly by degrading protein aggregates that form spontaneously in peripheral tissues like muscle. Such aggregates are presumably intracellular (cytosolic?) and it is not at all clear how extracellular cathepsin would gain access to such aggregates. Endocytosis of the protease and autophagy of the aggregates is a conceivable mechanism, but autophagosomes would fuse with endogenous lysosomes that are replete with proteases, including cathepsins, at concentrations that are most likely much higher than what would be contributed by internalized extracellular cathepsin.

The above, in my view unlikely, hypothesis is preferred over the possible immune and metabolic changes that are also likely to occur based on the authors' own findings. Investigating the alternative possibilities was deemed by the authors to be beyond the scope of their studies. As such, the manuscript fails to provide a mechanism linking the epithelial cause with the effects in peripheral tissues. Secondly, the puzzling accumulation of LysoTracker in lysosomes that are putatively unable to acidify properly remains unexplained and, in my view, inconsistent with the conclusions reached. The authors correctly point out that "LysoTracker fluorescence is independent of pH". But this refers to the fluorescence of individual LysoTracker molecules as a function of pH. But the accumulation of LysoTracker in acidic compartment is a pH-dependent process: LysoTracker is a membrane-permeant weak base that enters lysosomes and accumulates there as a result of protonation that generates the charged impermeant/less permeant species that is unable to leave the lumen. This has been demonstrated in dozens, if not hundreds of publications in a variety of biological systems. It is thus difficult to account for the clear accumulation of LysoTracker in the lysosomes of vha-16 and -19 mutants in figure 6c, which seems indistinguishable from that of wildtype worms. If not acidification, what causes the accumulation of LysoTracker in the vha-16 and -19 mutants? Would LysoTracker accumulate in those lysosomes if the cells are pretreated with concanamycin or with ammonia to dissipate the pH gradient prior to addition of LysoTracker?

Reviewer #2 (Remarks to the Author):

The authors have addressed all major concerns.
I recommend publication in NCB.

Reviewer #3 (Remarks to the Author):

In this revised manuscript by Li, Auwerx and colleagues, the authors have gone to extensive lengths to offer forward more rigorous proof for the LySR and its effector machinery. Specifically, this reviewer appreciates the addition/conduct of mutant analyses (e.g. atfs-1 FoxA/pha-4, and TFEB/hlh-30) and, in spite of its inability to effectively degrade VHA-6, the attempts made at AID-based targeting. The confirmatory data associated with AID-based ELT-2 targeting very nicely substantiate some of the manuscript's key conclusions. I am also more convinced by:

- 1) The inclusion of summary lifespan data in tabular format
 - 2) The specificity of ELT-2 interaction with promoters associated with the LySR
 - 3) The inclusion of state-of-the-art and well accepted tools to quantify lysosomal acidification and function.
- The manuscript is now exceptionally well positioned to make a substantive advance on the field.

A few minor comments:

I think the great lengths the authors went to to attempt AID-based degradation of vha-6 should be included somewhere in the supplementary materials because it is tremendously instructive to the field on the utility of this system for different classes of cellular proteins.

The authors should probably make reference to supplementary table 4 when they introduce the first make reference to lifespan data in the manuscript (P3 line 12, results, in addition to the first mention in the legend of figure 1) and make some generic statement that statistical data and information on replication can be found in this table for all lifespans conducted in the manuscript.

Although some conclusions including the mechanistic ties to DR-based lifespan extension are weakened by the failure of pha-4, AMPK, and DAF-16 to impact LySR lifespan-extension, the authors have suitably added more discussion to address these potential inconsistencies.

Line 35 abstract Alzheimer's, Huntington's should probably be separated by a comma as they are in a 3-item list.

Line 36 abstract no comma needed after ELT-2.

There is a CBP-1 GOF mutant available. The authors could test its sufficiency with or without elt-2 OE to see if the two in combination are sufficient to induce LySR gene expression, although this is not strictly required.

The hlh-30 mutant suppression of vha-6-RNAI-based lifespan extension could also be explained by the pleiotropic nature of the LySR, in that part of it is executed by elt-2, but not all dimensions. Thus an alternative (perhaps semantic) way to comment on the suppression seen with hih-30 loss is that it executes part of the LySR response that is required for the longevity response (not simply downstream).

GUIDELINES FOR SUBMISSION OF NATURE CELL BIOLOGY ARTICLES

ARTICLE FORMAT

ABSTRACT – should not exceed 150 words and should be unreferenced. This paragraph is the most visible part of the paper and should briefly outline the background and rationale for the work, and accurately summarize the main results and conclusions. Key genes, proteins and organisms should be specified to ensure discoverability of the paper in online searches.

TEXT – the main text consists of the Introduction, Results, and Discussion sections and must not exceed 3500 words including the abstract. The Introduction should expand on the background relating to the work. The Results should be divided in subsections with subheadings, and should provide a concise and accurate description of the experimental findings. The Discussion should expand on the findings and their implications. All relevant primary literature should be cited, in particular when discussing the background and specific findings.

REFERENCES – are limited to a total of 70 in the main text and Methods combined,. They must be numbered sequentially as they appear in the main text, tables and figure legends and Methods and must follow the precise style of Nature Cell Biology references. References only cited in the Methods should be numbered consecutively following the last reference cited in the main text. References only associated with Supplementary Information (e.g. in supplementary legends) do not count toward the total reference limit and do not need to be cited in numerical continuity with references in the main text. Only published papers can be cited, and each publication cited should be included in the numbered reference list, which should include the manuscript titles. Footnotes are not permitted.

Methods should be written concisely, but should contain all elements necessary to allow interpretation and replication of the results. As a guideline, Methods sections typically do not exceed 3,000 words. The Methods should be divided into subsections listing reagents and techniques. When citing previous methods, accurate references should be provided and any alterations should be noted. Information must be provided about: antibody dilutions, company names, catalogue numbers and clone numbers for monoclonal antibodies; sequences of RNAi and cDNA probes/primers or company names and catalogue numbers if reagents are commercial; cell line names, sources and information on cell line identity and authentication. Animal studies and experiments involving human subjects must be reported in detail, identifying the committees approving the protocols. For studies involving human subjects/samples, a statement must be included confirming that informed consent was obtained. Statistical analyses and information on the reproducibility of experimental results should be provided in a section titled "Statistics and Reproducibility".

All Nature Cell Biology manuscripts submitted on or after March 21 2016, must include a Data availability statement as a separate section after Methods but before references, under the heading "Data Availability". For Springer Nature policies on data availability see <http://www.nature.com/authors/policies/availability.html>; for more information on this particular policy see <http://www.nature.com/authors/policies/data/data-availability-statements-data-citations.pdf>. The Data availability statement should include:

- Accession codes for primary datasets (generated during the study under consideration and designated as "primary accessions") and secondary datasets (published datasets reanalysed during the study under consideration, designated as "referenced accessions"). For primary accessions data should be made public to coincide with publication of the manuscript. A list of data types for which submission to community-endorsed public repositories is mandated (including sequence, structure, microarray, deep sequencing data) can be found here <http://www.nature.com/authors/policies/availability.html#data>.
- Unique identifiers (accession codes, DOIs or other unique persistent identifier) and hyperlinks for datasets deposited in an approved repository, but for which data deposition is not mandated (see here for details <http://www.nature.com/sdata/data-policies/repositories>).
- At a minimum, please include a statement confirming that all relevant data are available from the authors, and/or are included with the manuscript (e.g. as source data or supplementary information), listing which data are included (e.g. by figure panels and data types) and mentioning any restrictions on availability.
- If a dataset has a Digital Object Identifier (DOI) as its unique identifier, we strongly encourage including this in the Reference list and citing the dataset in the Methods.

We recommend that you upload the step-by-step protocols used in this manuscript to [protocols.io](https://www.protocols.io). More details can found at <https://www.protocols.io/help/publish-articles>.

DISPLAY ITEMS – main display items are limited to 6-8 main figures and/or main tables. For Supplementary Information see below.

FIGURES – Colour figure publication costs \$395 per colour figure. All panels of a multi-panel figure must be logically connected and arranged as they would appear in the final version. Unnecessary figures and figure panels should be avoided (e.g. data presented in small tables could be stated briefly in the text instead).

All imaging data should be accompanied by scale bars, which should be defined in the legend.

Cropped images of gels/blots are acceptable, but need to be accompanied by size markers, and to retain visible background signal within the linear range (i.e. should not be saturated). The boundaries of panels with low background have to be demarked with black lines. Splicing of panels should only be considered if unavoidable, and must be clearly marked on the figure, and noted in the legend with a statement on whether the samples were obtained and processed simultaneously. Quantitative comparisons between samples on different gels/blots are discouraged; if this is unavoidable, it has to be performed for samples derived from the same experiment with gels/blots were processed in parallel, which needs to be stated in the legend.

Regardless of format, all figures must be vector graphic compatible files, not supplied in a flattened raster/bitmap graphics format, but should be fully editable, allowing us to highlight/copy/paste all text and move individual parts of the figures (i.e. arrows, lines, x and y axes, graphs, tick marks, scale bars etc). The only parts of the figure that should be in pixel raster/bitmap format are photographic images or 3D rendered graphics/complex technical illustrations.

Unprocessed scans of all key data generated through electrophoretic separation techniques need to be presented in a supplementary figure that should be labeled and numbered as the final supplementary figure, and should be mentioned in every relevant figure legend. This figure does not count towards the total number of figures and is the only figure that can be displayed over multiple pages, but should be provided as a single file, in PDF or TIFF format. Data in this figure can be displayed in a relatively informal style, but size markers and the figures panels corresponding to the presented data must be indicated.

The total number of Supplementary Figures (not including the "unprocessed scans" Supplementary Figure) should not exceed the number of main display items (figures and/or tables (see our Guide to Authors and March 2012 editorial <http://www.nature.com/ncb/authors/submit/index.html#suppinfo>; <http://www.nature.com/ncb/journal/v14/n3/index.html#ed>). No restrictions apply to Supplementary Tables or Videos, but we advise authors to be selective in including supplemental data.

GUIDELINES FOR EXPERIMENTAL AND STATISTICAL REPORTING

REPORTING REQUIREMENTS – To improve the quality of methods and statistics reporting in our papers we have recently revised the reporting checklist we introduced in 2013. We are now asking all life sciences authors to complete two items: an Editorial Policy Checklist (found here <https://www.nature.com/authors/policies/Policy.pdf>) that verifies compliance with all required editorial policies and a Reporting Summary (found here <https://www.nature.com/authors/policies/ReportingSummary.pdf>) that collects information on experimental design and reagents. These documents are available to referees to aid the evaluation of the manuscript. Please note that these forms are dynamic 'smart pdfs' and must therefore be downloaded and completed in Adobe Reader. We will then flatten them for ease of use by the reviewers. If you would like to reference the guidance text as you complete the template, please access these flattened versions at <http://www.nature.com/authors/policies/availability.html>.

Version 2:

Decision Letter:

Our ref: NCB-A54524B

11th April 2025

Dear Dr. Auwerx,

Thank you for submitting your revised manuscript "A lysosomal surveillance response (LySR) that boosts lysosomal activity and extends healthspan" (NCB-A54524B). It has now been seen by the original referees and their comments are below. The reviewers find that the paper has improved in revision, and therefore we'll be happy in principle to publish it in Nature Cell Biology, pending minor revisions to satisfy the referees' final requests and to comply with our editorial and formatting guidelines.

Thank you again for your interest in Nature Cell Biology Please do not hesitate to contact me if you have any questions.

Sincerely,

Angela R Parrish, PhD
Locum Senior Editor
Nature Cell Biology

Reviewer #1 (Remarks to the Author):

The authors have now addressed my comments and incorporated caveats and clarifications in the iteration that have significantly improved the manuscript.

Version 3:

Decision Letter:

Dear Dr Auwerx,

I am pleased to inform you that your manuscript, "A lysosomal surveillance response to stress extends healthspan", has now been accepted for publication in Nature Cell Biology. Congratulations!

Over the next few weeks, your paper will be copyedited to ensure that it conforms to Nature Cell Biology style. Once your paper is

typeset, you will receive an email with a link to choose the appropriate publishing options for your paper and our Author Services team will be in touch regarding any additional information that may be required.

Please note that *Nature Cell Biology* is a Transformative Journal (TJ). Authors may publish their research with us through the traditional subscription access route or make their paper immediately open access through payment of an article-processing charge (APC). Authors will not be required to make a final decision about access to their article until it has been accepted. [Find out more about Transformative Journals](https://www.springernature.com/gp/open-research/transformative-journals)

If you have not already done so, we strongly recommend that you upload the step-by-step protocols used in this manuscript to protocols.io (<https://protocols.io>), an open online resource that allows researchers to share their detailed experimental know-how. All uploaded protocols are made freely available and are assigned DOIs for ease of citation. Protocols and Nature Portfolio journal papers in which they are used can be linked to one another, and this link is clearly and prominently visible in the online versions of both. Authors who performed the specific experiments can act as primary authors for the Protocol as they will be best placed to share the methodology details, but the Corresponding Author of the present research paper should be included as one of the authors. By uploading your Protocols onto protocols.io, you are enabling researchers to more readily reproduce or adapt the methodology you use, as well as increasing the visibility of your protocols and papers. You can also establish a dedicated workspace to collect your lab Protocols. Further information can be found at <https://www.protocols.io/help/publish-articles>.

Nature Cell Biology encourages authors presenting evidence for cell, biological, molecular, and genetic interactions to consider communicating these findings using Biofactoid (<https://biofactoid.org/>). This tool helps users share a searchable representation of interactions (e.g. binding, gene expression, post-translational modification) between genes, gene products, or chemicals. Information added to Biofactoid, with author attribution, is shared on social media and public databases, such as Pathway Commons, where it can be discovered and analyzed in the context of a large and growing corpus of knowledge.

With kind regards,

Angela R Parrish, PhD
Locum Senior Editor
Nature Cell Biology

** Visit the Springer Nature Editorial and Publishing website at http://editorial-jobs.springernature.com?utm_source=eJP_NCB_email&utm_medium=eJP_NCB_email&utm_campaign=eJP_NCB for more information about our career opportunities. If you have any questions please click [here](mailto:editorial.publishing.jobs@springernature.com).
